# Microglial replacement in a Sandhoff disease mouse model reveals myeloid-derived β-hexosaminidase is necessary for neuronal health

Kate I. Tsourmas[1,2], Claire A. Butler [1,2], Nellie E. Kwang [1,2], Zachary R. Sloane[1,2], Koby J. G. Dykman[1,2], Ghassan O. Maloof[1,2], Biswa P. Choudhury[3], Mousumi Paulchakrabarti[3], Christiana A. Prekopa[1,2], Emily Z. Tabaie [1,2], Robert P. Krattli [4], Sanad M. El-Khatib [4], Vivek Swarup [1,2], Munjal M. Acharya [4,5], Lindsay A. Hohsfield[1,2] & Kim N. Green [1,2] ✉

Lysosomal storage disorders (LSDs) are a large disease class involving lysosomal dysfunction, often resulting in neurodegeneration. Sandhoff disease (SD) is an LSD caused by a deficiency in the β subunit of the β-hexosaminidase enzyme (*Hexb*). Although *Hexb* expression in the brain is specific to microglia, SD primarily affects neurons. To investigate how a microglial gene is involved in neuronal homeostasis, here we show that β-hexosaminidase is secreted by microglia and integrated into the lysosomal compartment of neurons. To assess therapeutic relevance, we treat the *Hexb*[-/-] SD mouse model with bone marrow transplant and colony stimulating factor 1 receptor inhibition, which broadly replaces *Hexb*[-/-] microglia with *Hexb*-sufficient cells. Microglial replacement reverses apoptotic gene signatures, improves behavior, restores β-hexosaminidase enzymatic activity and *Hexb* expression, prevents substrate buildup, and normalizes neuronal lysosomal phenotypes, underscoring the critical role of myeloid-derived β-hexosaminidase in maintaining neuronal health and establishing microglial replacement as a potential LSD therapy.

Lysosomal storage disorders (LSDs) are a class of genetic diseases caused by deficiencies in lysosomal enzymes or membrane proteins, resulting in the accumulation of excess substrate that causes lysosomal dysfunction and/or cell death[1]. Though individual LSDs are rare, LSDs have an overall frequency of approximately 1 in 5000 live births[2]. The majority of LSDs are characterized by a progressive neurodegenerative phenotype with an infantile or early childhood onset. One such disorder is Sandhoff disease (SD), which is caused by a complete loss of the β-Hexosaminidase enzyme (Hexβ)[3,4]. Hexβ is a dimeric enzyme composed of either an alpha (HEXA) and beta (HEXB) subunit or two beta subunits, with the respective subunits encoded by the *Hexa* and *Hexb* genes[5]. In SD, a disruption to the *Hexb* gene results in the complete loss of functioning Hexβ enzyme and the inability to break down its substrates, namely GM2 ganglioside glycolipids and other glycolipids/glycoproteins, which accumulate in the central nervous system (CNS) and peripheral organs[6,7]. While some studies have linked dysregulated glycolipid metabolism to the pathogenesis of neurological disorders, it remains unclear how glycolipid accumulation causes

[1]Department of Neurobiology and Behavior; University of California, Irvine, CA, USA. [2]Institute for Memory Impairments and Neurological Disorders; University of California, Irvine, CA, USA. [3]GlycoAnalytics Core, Glycobiology Research and Training Center, School of Medicine, University of California, San Diego, CA, USA. [4]Department of Anatomy and Neurobiology; University of California, Irvine, CA, USA. [5]Department of Radiation Oncology; University of California, Irvine, CA, USA. ✉e-mail: kngreen@uci.edu

neurodegeneration[8,9]. In humans and *Hexb*[-/-] mice, which closely recapitulate features of SD, the disease manifests as a rapidly progressive neurodegenerative disorder characterized by extensive neuroinflammation followed by mass neuronal apoptosis, severe motor and developmental impairments, and death by age four in humans and 18–20 weeks in mice[3,10,11]. At present, there is no available curative or disease-modifying treatment for SD.

Bone marrow transplant (BMT) has been shown to be an effective treatment for several LSDs and has been investigated as a potential treatment strategy for SD[12,13]. However, in human patients with SD and the closely related gangliosidosis Tay-Sach's disease (TSD), BMT has been ineffective and repeatedly failed to meaningfully extend lifespan[10,14,15]. BMT has shown limited efficacy in *Hexb*[-/-] mice, reducing microglial activation and partially prolonging life; however, it ultimately failed to normalize lifespan or correct CNS pathology[11,16]. *Hexb*[-/-] BMT-treated mice exhibit a reduction of GM2 ganglioside burden, but only in peripheral organs. This insufficiency may be linked to a failure to replace microglia, the primary myeloid cells of the CNS, using traditional BMT. Although studies have shown that BMT can reduce enzyme substrate accumulation in peripheral organs of many LSDs, including SD, the CNS has been notoriously difficult to correct in many cases[12]. This is likely attributable to the relatively higher rates of myeloid cell replacement in these organs compared to the nominal replacement rate of CNS myeloid cells with bone-marrow derived cells, ranging from < 10 to ~ 30%[17–21]. Microglia are heavily implicated in the development of SD, and *Hexb* expression in the CNS has been reported highly specific to microglia[11,22–26]. However, it remains unclear how deficits in a myeloid cell gene (i.e., loss of *Hexb*) result in the primarily neuronal pathology and cell death observed in the SD CNS. In this study, we sought to develop an approach that would allow us to better understand the relationship between myeloid *Hexb* expression and neuronal pathology in SD while improving upon the shortcomings of BMT and other treatment modalities with incomplete efficacy.

We and others have previously identified a means to reliably replace the microglial population with bone marrow-derived myeloid cells (BMDMs) via pharmacological inhibition of the colony-stimulating factor 1 receptor (CSF1R) combined with BMT[19,27–29]. This approach results in the broad and brain-wide replacement of microglia with BMDMs, achieving 70–99% replacement. Using this paradigm, we recently identified that BMDM cells enter the CNS via a specialized leptomeningeal structure underneath the hippocampus known as the velum interpositum[30], and then spread out to fill the brain. Busulfan-based BMT + CSF1R inhibitor (CSF1Ri) approaches have recently been utilized to therapeutically replace microglia in other mouse models of neurodegenerative disease, including progranulin deficiency, experimental autoimmune encephalomyelitis, Alzheimer's disease, and Prosaposin deficiency, all with promising results[31–34]. Here, we employ a BMT + CSF1Ri-based microglial replacement strategy in the *Hexb*[-/-] mouse model of SD and demonstrate that delivery of *Hexb*-expressing cells via myeloid cell replacement rescues neuron-related molecular and functional outcomes. In neurons, we observe reversed expression of apoptosis-associated genes, resolution of glycolipid/glycoprotein storage, clearance of accumulated lysosomal components, and reduction of vacuolization following microglial replacement with combined BMT + CSF1Ri treatment. Subsequent cell culture experiments reveal that microglia secrete enzymatically active Hexβ protein and that neurons take up extracellular Hexβ protein and integrate it into the lysosomal compartment, an exchange that may be mediated by the mannose-6-phosphate receptor. These experiments not only provide evidence for a promising treatment strategy for SD and other CNS LSDs but also indicate that myeloid-derived Hexβ may be essential for neuronal health and lysosomal function.

## Results

### Spatial transcriptomics reveals Hexb-associated genetic changes

To explore the molecular underpinnings of SD, we performed multiplex single-cell resolution in situ RNA analysis by spatial molecular imaging[35] on *Hexb*[-/-] mouse brains. Previous studies have shown that *Hexb*[-/-] mice faithfully recapitulate features of human SD, including neuroinflammation/microglial activation, GM2 ganglioside accumulation, and severe motor decline[7] (Fig. 1a). For this experiment, wildtype control and *Hexb*[-/-] mice (*n* = 10-11 per group; *n* = 3 selected from each group for transcriptomic experiments) were sacrificed at 16 weeks, a humane endpoint at which *Hexb*[-/-] mice present severe motor phenotypes. Fixed brains were sectioned sagittally at 10 μm, imaged with rRNA, Histone, DAPI, and GFAP markers for cell segmentation, and analyzed for 1000 genes using the Nanostring CosMx Spatial Molecular Imager platform (316 total FOVs, ~ 52 FOVs per brain section) (Fig. 1a, b; examples of cell segmentation in Supplementary Fig. 1a). An imaging-based spatial transcriptomic approach is advantageous in its ability to identify brain regions more affected by disease, while also offering a high percentage of cell capture (~ 90%), including much higher rates of myeloid cell capture (~ 99%), and relative reduction in sampling bias in comparison to single-cell RNA sequencing (RNA-seq) approaches.

With this approach, we captured 196,533 cells with a mean transcript count of ~ 800 transcripts per cell. Unsupervised cell clustering identified 39 transcriptionally distinct clusters (Fig. 1c and Supplementary Fig. 1b). Clusters were annotated with a combination of automated and manual approaches: (1) label annotations from the Allen Brain Atlas single-cell RNA-seq reference dataset (for cortex and hippocampus) were projected onto our spatial transcriptomics dataset[36], and (2) cluster identities were further refined via manual annotation based on gene expression of known marker genes and location in XY space (Supplementary Figs. S2 and S3). We identified 14 clusters of excitatory neurons, five clusters of inhibitory neurons, six astrocyte clusters, two myeloid clusters, four oligodendrocyte clusters, one oligodendrocyte precursor (OPC) cluster, three vasculature-associated clusters, two endothelial clusters, and two uncategorized (other) clusters. Projecting cell subclusters in XY space shows clear separation between anatomical regions and cortical layers (Fig. 1d and Supplementary Fig. 1c).

Analysis of the distribution of cell counts within each cluster by genotype revealed a robust change in myeloid cell populations in *Hexb*[-/-] mice (Fig. 1e). Both Myeloid subclusters share a significant number of genes associated with myeloid cells; however, we observe notable differences between the two subclusters in regard to cell type identity and activation state. The Myeloid 1 subcluster includes several top-enriched genes indicative of a homeostatic microglial signature (e.g., *Csf1r*, *Hexb*, *Pr2y12*, *Cx3cr1*, *Tmem119*, *Sall1*), whereas top-enriched genes in the Myeloid 2 subcluster include genes associated with antigen presentation and cell activation or disease-associated markers (e.g., *H2-Aa*, *Cd74*, *H2-Ab1*, *Lyz1/2*, *Ptprc*, *Ctss*, *Itgax*, *Axl*, *Apoe*, and *Cst7*)[37–42]. Based on these genes, it appears that the Myeloid 1 subcluster represents microglial cells and the Myeloid 2 subcluster represents either infiltrating macrophages/monocytes, dendritic cells, disease-associated microglia, or a combination of the three[43–46]. *Hexb*[-/-] mice exhibited a high proportion of cells in the Myeloid 2 subcluster (88.6%) compared to wild type (WT) mice, indicating a near-exclusive presence of this cell type in the *Hexb*[-/-] genotype. Interestingly, infiltrating monocytes/macrophages have previously been reported in small quantities in *Hexb*[-/-] brains[47], indicating the increase in Myeloid 2 *Hexb*[-/-] brains may indicate the presence of monocytes/macrophages, though the presence of dendritic cells or disease-associated microglia cannot be ruled out. Plotting of the Myeloid 2 subcluster in XY space indicates that the cells are localized to the thalamus and throughout

 

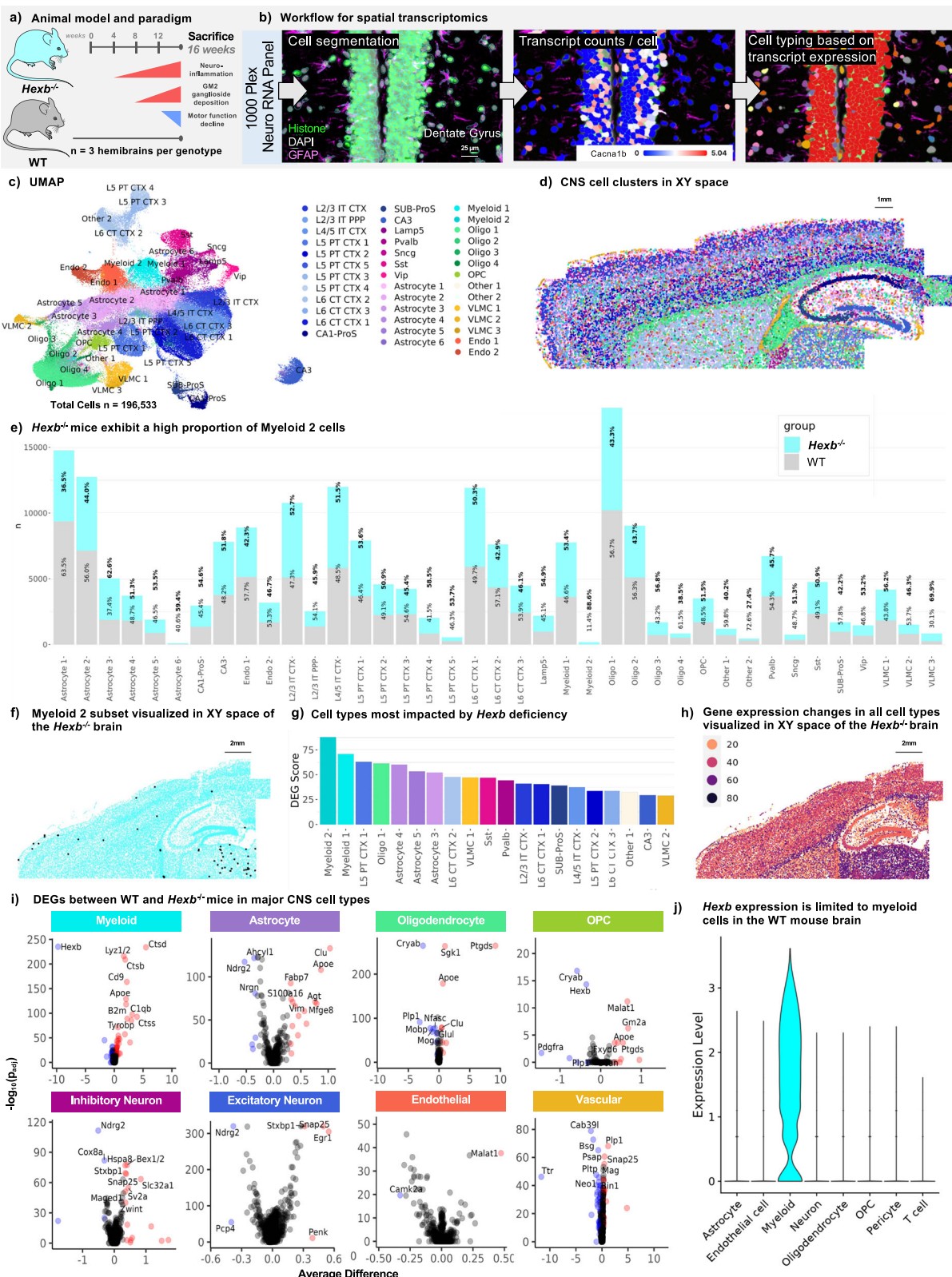

the cortex (Fig. 1f). It should be noted that there were no major differences in the overall counts of astrocytes, oligodendrocytes, or myeloid cells (Fig. 1e and Supplementary Fig. 1d). Surprisingly, we also observed no notable reductions in neuronal subcluster counts or overall neuronal counts in the *Hexb⁻/⁻* brain in comparison to WT, indicating a lack of overt neuronal loss by 16 weeks in SD mice (Supplementary Fig. 1d).

Next, we performed differential gene expression (DGE) analysis on all cell clusters to assess gene expression changes associated with loss of *Hexb* (Supplementary Fig. 4). Each cluster was then assigned a differentially expressed gene (DEG) score (Fig. 1g and Supplementary Fig. 3d). DEG score measures the magnitude of gene expression changes between two groups within each cellular subcluster; DEG scores were calculated for each subcluster by summing the absolute

**Fig. 1 | Spatial transcriptomic analysis of the SD mouse brain identifies disease-associated gene expression signatures. a** Timeline of symptom progression in *Hexb⁻/⁻* Sandhoff disease model mice up to the point of sacrifice at 16 weeks (*n* = 3/genotype, *Hexb⁻/⁻* and wildtype (WT) control). Microglial/myeloid activation begins at ~ 4 weeks, accumulation of GM2 ganglioside glycolipid can be detected ~ 8 weeks, and motor deterioration begins ~ 12 weeks. Mouse image adapted from Servier Medical Art, https://smart.servier.com/smart_image/mouse/. **b** Experimental workflow for targeted 1000-plex single-cell spatial transcriptomics. Fields of view (FOVs) were selected in the cortex, corpus callosum, hippocampus, and upper regions of caudate and thalamus of each sagittal section, then imaged with DNA, rRNA, Histone, and GFAP markers for cell segmentation. Transcript counts for each gene were acquired per cell. **c** Uniform Manifold Approximation and Projection (UMAP) of 196,533 cells across 6 brains. Clustering at 1.0 resolution yielded 39 clusters, which were annotated with a combination of automated and manual approaches with reference to Allen Brain Atlas single-cell RNA-seq cell types, gene

expression, and anatomical location in space. **d** 39 clusters plotted in XY space in WT brain. **e** Bar graph of proportions of cell counts by subcluster per genotype. **f** Myeloid 2 subcluster (black) overlaid above representative *Hexb⁻/⁻* brain plotted in XY space. **g** Descending bar graph of the top 20 subclusters with the highest differentially expressed gene (DEG) scores. Differential gene expression analysis per cell type between groups was performed on scaled expression data using Model-based Analysis of Single-cell Transcriptomics (MAST) to calculate the average difference, defined as the difference in log-scaled average expression between the two groups for each broad cell type. Following differential gene expression analysis, the DEG score was calculated per subcluster by summing the absolute value of the $\log_2$ fold change values for all DEGs between *Hexb⁻/⁻* and WT control with a $p_{adj}$ value below 0.05. **h** Projection of subclusters colored by DEG score in XY space in representative *Hexb⁻/⁻* brain. **i** Volcano plots of DEGs identified between *Hexb⁻/⁻* and WT control for each broad cell type. **j** Violin plot of *Hexb* transcript counts in cell types demonstrating myeloid-specific expression.

---

$\log_2$ fold change values of all genes with significant ($p_{adj} < 0.05$) differential expression patterns between *Hexb⁻/⁻* and WT. This metric allowed us to assess broad alterations in cellular subtypes caused by *Hexb* insufficiency (Fig. 1g). Myeloid 1 and 2 subclusters had the highest DEG scores, indicating that these cell populations are the most impacted by *Hexb* deficiency. To visualize DEGs across major CNS cell types, we next combined our subclusters into broad cell types (i.e., Astrocyte clusters 1–6 were placed in the Astrocyte broad cell type category). DGE analysis of all myeloid subclusters revealed, as expected, a marked downregulation of *Hexb*, accompanied by an upregulation in several key inflammatory genes: cathepsins (*Ctsb, Ctsd, Ctss*), immune activation genes (*B2m, Tyrobp*), and macrophage-associated genes (*Lyz1/2, C1qb*)[48–52] (Fig. 1i). DGE analysis of oligodendrocytes identified several genes associated with CNS inflammatory/stress response (*Ptgds, Sgk, Cryab*) and demyelination (*Mog, Mobp, Plp1*)[53–56] (Fig. 1i); oligodendrocyte expression of *Ptgds*, in particular, has been shown to induce neuronal apoptosis[57]. Plotting astrocyte DEGs, we found a downregulation in the homeostatic astrocyte gene *Ndrg2* and upregulation of markers associated with astrocyte activation and neurotoxicity (*Clu, Apoe, Fabp7, S100a6, Vim*)[58–66].

Neuronal DEGs and DEG scores indicated major gene expression alterations in response to the loss of *Hexb*. Plotting top DEGs revealed that inhibitory neurons exhibit a litany of DEGs associated with perturbed neurotransmission and apoptotic processes (*Sv2a, Slc32a1, Bex1/2, Zwint, Maged1*) and cellular stress/metabolic processes (*Hsp8a, Cox8a*); *Hsp8a* is also a key regulator of lysosome activity and autophagy[67–74]. Excitatory neurons also exhibited alterations in genes associated with apoptosis. We observe a strong upregulation in early growth response 1 (*Egr1*), a gene previously implicated in orchestrating neuronal apoptosis and modulating expression of stress-responsive transcription factor EB (TFEB), a master regulator of lysosomal biogenesis and autophagy; *Egr1* has also been shown to be upregulated under conditions of lysosomal dysfunction[75–77]. We also note a downregulation of Purkinje cell protein 4 (*Pcp4*), a gene that is decreased in various neurodegenerative diseases and also linked to apoptosis[78]. Both excitatory and inhibitory neurons also exhibited an upregulation of genes associated with development and synaptic function (*Snap25, Stxbp1*)[79,80]. Although we did not observe gross changes in neuronal counts at this stage of disease in *Hexb⁻/⁻* mice, these neuronal gene expression changes are notable in their indication of broad neuronal dysregulation and the initiation of apoptotic processes. The selected endpoint thus may have captured the state of the SD CNS shortly preceding overt neuronal loss, as *Hexb⁻/⁻* mice generally survive to 18–20 weeks and disease progresses rapidly.

We next plotted all subcluster DEG scores in XY space to visualize broad gene expression changes spatially and identify region-specific vulnerabilities (Fig. 1h). The thalamus and corpus callosum were populated by cells with the highest DEG scores. Cells throughout the cortex had higher DEG scores than those of the hippocampus and

caudoputamen. These region-specific effects align well with previous results from human SD patients and mice, which report white matter neurodegeneration, thalamic hyperintensities/hyperdensities, and cortical atrophy with relative sparing of the caudate[81–83]. Notably, many of the observed gene expression changes between *Hexb⁻/⁻* and WT mice also closely aligned with DEGs previously identified in datasets derived from human SD and TSD patients, including *Ptgds, Vim, Apoe, Clu, Ctsb, Nrgn*, and *Mbp*[84]. Our DGE analysis provides evidence that CNS cell types of various lineages are affected by SD. Upregulation of genes associated with reactivity in glial cells may contribute to the apoptotic signatures detected in *Hexb⁻/⁻* neurons. Understanding how differing cell types interact and contribute to neurodegeneration in SD is of great interest to understanding disease pathogenesis and uncovering potential therapeutic opportunities.

Finally, we assessed *Hexb* expression levels in various cell subtypes in WT animals. Interestingly, despite the established role of Hexβ in maintaining neuronal health, we detected *Hexb* transcripts exclusively in myeloid cells of WT animals (Fig. 1j). Very few transcripts were detected in other cell types, including astrocytes, endothelial cells, neurons, oligodendrocytes, OPCs, pericytes, or T cells. Our identification of myeloid-specific *Hexb* expression is in agreement with previous reports of transcript and protein expression patterns, which show specific expression of *Hexb* in microglia in the CNS[25,26]. These data collectively identify the myeloid population as a particularly significant cell type in the SD brain, highlighting it as a promising target for therapeutic intervention.

### BMT + CSF1Ri treatment leads to functional rescue

Given the high potential of a myeloid cell-based therapeutic target for SD, we next sought to replace *Hexb* deficient microglia in *Hexb⁻/⁻* mice with *Hexb* sufficient cells from WT donors and assess the viability of microglial replacement as a treatment for SD. Pre-pathological (4–6 weeks of age) *Hexb⁻/⁻* mice and WT controls were treated with BMT by total body irradiation and subsequent retro-orbital injection of bone marrow cells (Fig. 2a). Bone marrow cells were isolated from sex-matched CAG-EGFP mice, allowing for visual tracking of donor cells based on GFP expression[85]. We hypothesized that successful engraftment of CAG-EGFP donor cells in the brain would allow for normalization of *Hexb* expression. Chimerism analysis showed that BMT resulted in an average blood (granulocyte) and bone marrow (hematopoietic stem cell; HSC) chimerism rate of ~ 95–99%, with no notable differences between genotypes or treatment paradigms (Supplementary Fig. 5a–c). Following BMT, one group of mice was then placed on a control diet (WT BMT *n* = 10, *Hexb⁻/⁻* BMT *n* = 11). Another group underwent a 2-week post-irradiation recovery period before being treated with the CSF1R inhibitor diet PLX5622 at a dose of 1200 ppm for 7 days to induce widespread microglial depletion. The inhibitor was then withdrawn and the group returned to a control diet, which we have previously show results in efficient replacement of microglia with

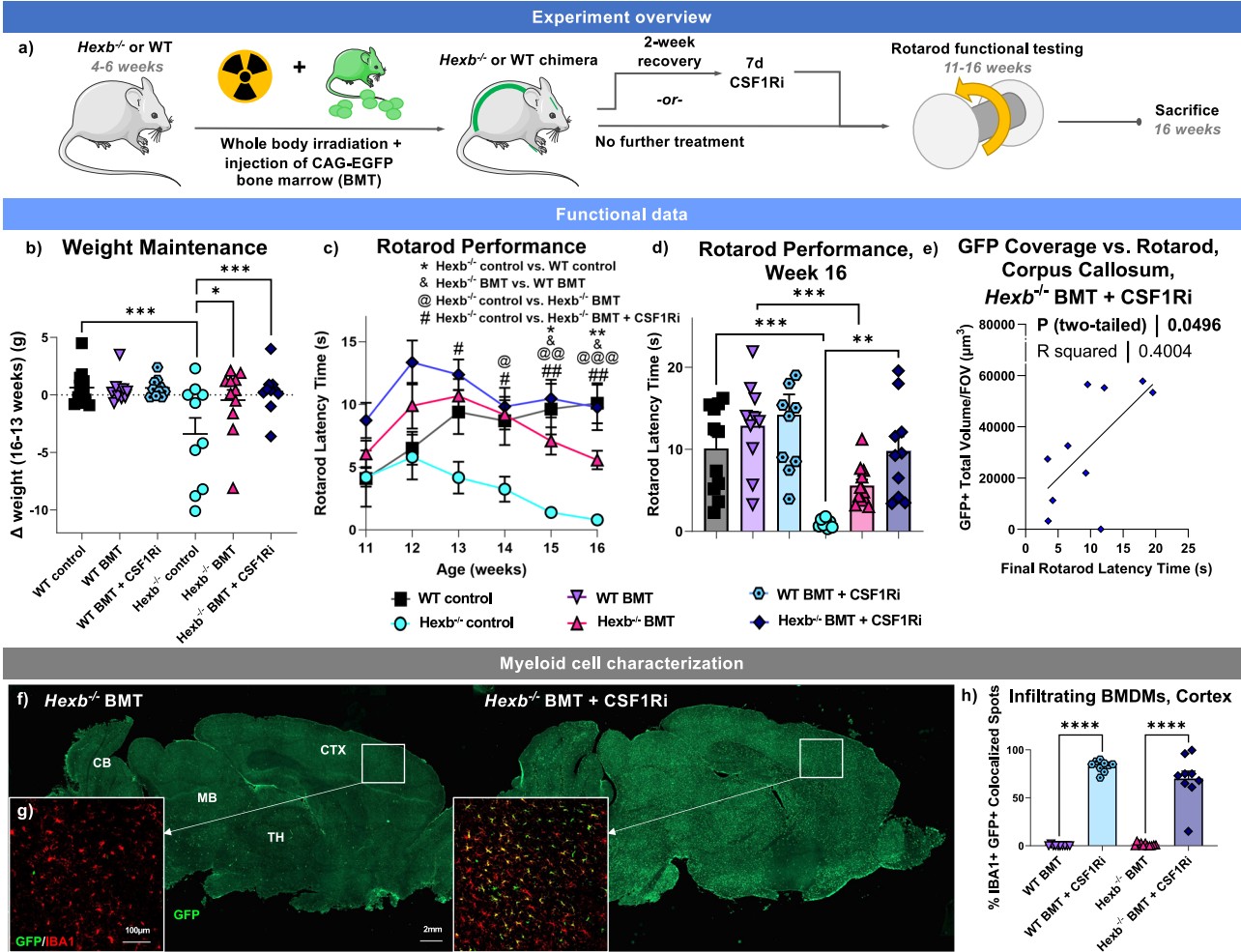

**Fig. 2 | Microglial replacement in SD leads to functional rescue. a** Schematic of the treatment paradigm. WT and *Hexb⁻/⁻* mice were split into 3 groups: untreated control, bone marrow transplant (BMT), and BMT plus colony-stimulating factor 1 inhibitor treatment (BMT + CSF1Ri). Mice underwent Rotarod testing and were sacrificed at 16 weeks. Mouse image adapted from Servier Medical Art, https://smart.servier.com/smart_image/mouse/. **b** Categorical scatter plot of weight change between weeks 13–16. *p*-values: WT vs. *Hexb⁻/⁻*, 0.0004; *Hexb⁻/⁻* vs. *Hexb⁻/⁻* BMT, 0.0225; *Hexb⁻/⁻* BMT vs. *Hexb⁻/⁻* BMT + CSF1Ri, 0.0041. **c** Line graph displaying average Rotarod latency-to-fall time from weeks 11–16. Groups compared by repeated measures ANOVA with Tukey's post-hoc testing from week 13. Symbols indicate significance between *Hexb⁻/⁻* and WT (*; w15 *p* = 0.0204, w16 *p* = 0.0016), *Hexb⁻/⁻* BMT and WT BMT (&; w15 *p* = 0.0409, w16 *p* = 0.0201), *Hexb⁻/⁻* BMT and *Hexb⁻/⁻* (@; w14 *p* = 0.0119, w15 *p* = 0.0032, w16 *p* = 0.0007), and *Hexb⁻/⁻* BMT + CSF1Ri and *Hexb⁻/⁻* (#; w13 *p* = 0.0252, w14 *p* = 0.0230, w15 *p* = 0.0016, w16 *p* = 0.0071). **d** Scattered bar plot of week 16 Rotarod performance. *p*-values: WT vs. *Hexb⁻/⁻*, 0.0004; WT BMT vs. *Hexb⁻/⁻* BMT, 0.0022; *Hexb⁻/⁻* vs. *Hexb⁻/⁻* BMT + CSF1Ri,

0.0023. **e** Scatterplot with correlation analysis between week 16 Rotarod performance and green fluorescent protein (GFP, green)⁺ staining volume in upper corpus callosum in BMT + CSF1Ri-treated *Hexb⁻/⁻* mice with line of best fit (*p* = 0.0456). **f** GFP (green) in representative 10x sagittal brain images from *Hexb⁻/⁻* BMT and *Hexb⁻/⁻* BMT + CSF1Ri mice. CTX, cortex; MB, midbrain; CB, cerebellum; MB, midbrain; TH, thalamus. **g** Representative confocal cortex images showing GFP (green) and IBA1 (red) colocalization (yellow). **h** Bar graph of quantification of percentage of IBA1⁺ cells with colocalized GFP⁺ per FOV in cortex images from all BMT-treated groups, indicating the ratio of myeloid cells with bone marrow-derived myeloid cell (BMDM) identity. Source data are provided as a Source Data file. Two-way ANOVA with Sidak's post-hoc test. *p*-values < 0.0001. Data represented as mean ± SEM (*n* = 9 WT BMT + CSF1Ri; *n* = 10 WT BMT, *Hexb⁻/⁻*, *Hexb⁻/⁻* BMT + CSF1Ri; *n* = 11 WT, *Hexb⁻/⁻* BMT); groups compared by two-way ANOVA with Tukey's post-hoc test to examine biologically relevant interactions unless otherwise noted; *p < 0.05, **p < 0.01, ***p < 0.001, ****p < 0.0001).

BMDMs following head irradiation[19] (WT BMT + CSF1Ri *n* = 9, *Hexb⁻/⁻* BMT + CSF1Ri *n* = 10). Untreated mice were also included to serve as controls (*Hexb⁻/⁻* control *n* = 10, WT control *n* = 11).

To assess the efficacy of these treatment strategies on functional readouts of disease progression, mice were weighed every other day and motor function was assessed on a weekly basis using the accelerating Rotarod task (Fig. 2b–e). We observe that *Hexb⁻/⁻* mice exhibit a significant loss of weight between 13 and 16 weeks of age compared to WT mice (Fig. 2b). Both *Hexb⁻/⁻* BMT and *Hexb⁻/⁻* BMT + CSF1Ri mice lost significantly less weight by week 16 compared to *Hexb⁻/⁻* control mice. On the accelerating Rotarod task, *Hexb⁻/⁻* control mice showed a steady decline in motor performance, and all *Hexb⁻/⁻* control mice were unable to stay on the Rotarod for any amount of time by week 16 (Fig. 2b, c).

Both *Hexb⁻/⁻* BMT and *Hexb⁻/⁻* BMT + CSF1Ri mice significantly outperformed *Hexb⁻/⁻* controls on the Rotarod task (Fig. 2c, d). However, *Hexb⁻/⁻* BMT mice displayed progressively declining performance over the course of testing and had significantly shorter latency-to-fall times than WT BMT control mice in weeks 15 and 16. By contrast, the *Hexb⁻/⁻* BMT + CSF1Ri group had stable performance in later weeks and had greater mean differences from *Hexb⁻/⁻* controls than *Hexb⁻/⁻* BMT mice (Fig. 2c, d). *Hexb⁻/⁻* BMT + CSF1Ri mice also did not significantly differ in latency-to-fall time in comparison to WT BMT + CSF1Ri controls at any time point. In addition, four *Hexb⁻/⁻* BMT mice died prematurely or required humane euthanasia at or before week 16, in comparison to three mice in the *Hexb⁻/⁻* control group and only one mouse in the *Hexb⁻/⁻* BMT + CSF1Ri group. Overall, these data suggest that BMT +

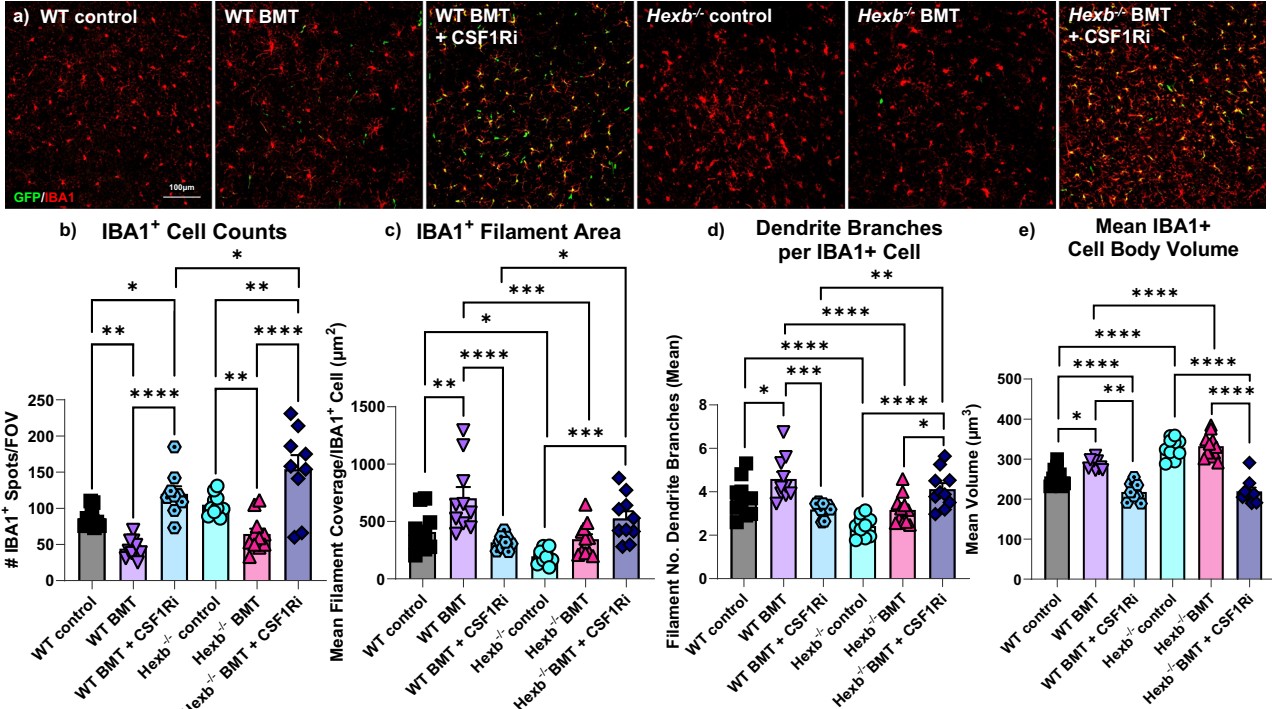

**Fig. 3 | Microglial replacement normalizes myeloid cell morphology.**
**a** Representative confocal images of cortex from all groups immunolabeled for GFP (green) and myeloid cell marker IBA1 (red). **b**–**e** Bar graphs of quantification of cortex images of (**b**) number of IBA1+ cells per FOV (p-values: WT vs. WT BMT, 0.0070; WT vs. WT BMT + CSF1Ri, 0.0419; WT BMT vs. WT BMT + CSF1Ri, < 0.0001; WT BMT + CSF1Ri vs *Hexb*-/- BMT + CSF1Ri, 0.0190; *Hexb*-/- vs *Hexb*-/- BMT, 0.0081; *Hexb*-/- vs *Hexb*-/- BMT + CSF1Ri, 0.0020; *Hexb*-/- BMT vs *Hexb*-/- BMT + CSF1Ri, < 0.0001), **c** mean area covered by filaments of individual IBA1+ cells in FOV (p-values: WT vs. WT BMT, 0.0016; WT vs. *Hexb*-/-, 0.0108; WT BMT vs. WT BMT + CSF1Ri, < 0.0001; WT BMT + CSF1Ri vs *Hexb*-/- BMT + CSF1Ri, 0.0141; WT BMT vs *Hexb*-/- BMT, < 0.0001; *Hexb*-/- vs *Hexb*-/- BMT + CSF1Ri, 0.0004), **d** mean number of branches per individual IBA1+ cell in FOV, (p-values: WT vs. WT BMT, 0.0231; WT vs.

*Hexb*-/-, 0.0004; WT BMT vs. WT BMT + CSF1Ri, 0.0004; WT BMT vs *Hexb*-/- BMT, < 0.0001; WT BMT + CSF1Ri vs *Hexb*-/- BMT + CSF1Ri, < 0.0001; *Hexb*-/- vs *Hexb*-/- BMT + CSF1Ri, < 0.0001; *Hexb*-/- BMT vs *Hexb*-/- BMT + CSF1Ri, 0.0132), and (**e**) mean cell body volume excluding filaments per IBA1+ cell in FOV (p-values: WT vs. WT BMT, 0.0207; WT vs. *Hexb*-/-, < 0.0001; WT vs· WT BMT + CSF1Ri, 0.0043; WT BMT vs. WT BMT + CSF1Ri, <0.0001; WT BMT vs *Hexb*-/- BMT, <0.0001; *Hexb*-/- vs *Hexb*-/- BMT + CSF1Ri, < 0.0001; *Hexb*-/- BMT vs *Hexb*-/- BMT + CSF1Ri, < 0.0001). Source data are provided as a Source Data file. Data are represented as mean ± SEM (n = 9 WT BMT + CSF1Ri; n = 10 WT BMT, *Hexb*-/-, and *Hexb*-/- BMT + CSF1Ri; n = 11 WT, *Hexb*-/- BMT); groups compared by two-way ANOVA with Tukey's post-hoc test to examine biologically relevant interactions unless otherwise noted; *p < 0.05, **p < 0.01, ***p < 0.001, ****p < 0.0001).

CSF1Ri leads to functional rescue, as seen by preservation of motor function and weight normalization in *Hexb*-/- mice.

## BMT + CSF1Ri causes extensive peripheral myeloid cell infiltration

We next assessed the efficacy of our treatment strategies in inducing BMDM infiltration into the CNS by staining for GFP and IBA1, a marker for myeloid cells. BMT alone led to limited GFP+ cell deposition throughout the parenchyma, where the BMT + CSF1Ri group exhibited a broad influx of GFP+ cells throughout the parenchyma (Fig. 2f–h). Myeloid cell chimerism in the cortex, identified based on colocalized GFP and IBA1 staining, averaged ~70–90% in BMT + CSF1Ri mice in comparison to near-zero colocalization after BMT alone (Fig. 2h). These observations are consistent with previous reports, in which BMT alone led to minimal myeloid cell replacement in the brain parenchyma aside from perivascular and meningeal spaces, contrasting the significant infiltration induced by BMT + CSF1Ri[19,86,87]. Overall, these data demonstrate highly efficient and significant replacement of microglia with donor-derived BMDMs following BMT + CSF1Ri. There was no significant difference in replacement rates between any of the *Hexb*-/- and WT groups, indicating that loss of *Hexb* does not affect rates of myeloid cell replacement.

To assess whether BMDM engraftment levels in different brain regions coincided with improved motor performance in *Hexb*-/- BMT + CSF1Ri mice, we performed correlation analyses between the final (week 16) Rotarod latency-to-fall time and GFP+ cell coverage in

the cortex, cerebellum, forebrain, and white matter/corpus callosum. We detected a significant positive correlation between week 16 Rotarod performance and GFP+ coverage in the upper corpus callosum (Fig. 2e). This finding complements the region-specific vulnerability identified in the corpus callosum by spatial transcriptomic DGE analysis, as well as previous reports of white matter-specific neurodegeneration in the human SD CNS[82,88]. There was no significant correlation between final week Rotarod performance and total GFP+ cell coverage in the cortex, cerebellum, or entire forebrain (Supplementary Fig. 5d–f). These data suggest that moderate overall BMDM infiltration is sufficient to improve motor performance, and that the presence of BMDMs in white matter regions is of particular importance. Overall, there is a clear relationship between the broad replacement of microglia with BMDMs and the significant functional rescue observed in the *Hexb*-/- BMT + CSF1Ri group in comparison to *Hexb*-/- controls, which BMT alone was not sufficient to produce.

## Infiltrating cells demonstrate microglia-like phenotypes

Detailed profiling of myeloid cell morphology revealed several changes induced by loss of *Hexb*, which were effectively reversed following microglial replacement. Staining for IBA1, a myeloid cell marker, revealed various morphological differences in *Hexb*-/- cells consistent with microglial activation, including a greater average cell count (Fig. 3b), decreased process (filament) length (Fig. 3c), decreased number of branches per cell (Fig. 3d), and increased cell body volume (Fig. 3e). The *Hexb*-/- BMT group only significantly differed from

controls in terms of cell count, with an overall loss of cells. However, the WT BMT group also demonstrated a significant loss of total IBA1+ cells, indicative of an irradiation-induced effect. By contrast, myeloid cells in the *Hexb*-/- BMT + CSF1Ri group had significantly longer processes, more branches, and a lower average cell body volume than *Hexb*-/- controls. These data suggest that infiltrating BMDMs induced by BMT + CSF1Ri appear less ameboid/activated compared to microglia in *Hexb*-/- control brains.

We then sought to determine the cellular identity of the infiltrating bone marrow-derived population via flow cytometry, immunohistochemistry, and spatial transcriptomics. It has been previously shown that HSCs have the highest capacity to engraft the brain following myeloablative conditioning and CSF1Ri treatment[33]; thus, we sought to build upon these results by profiling donor-derived populations once engrafted in the brain for an extended period. To determine the cellular identity of our engrafted BMDM GFP + cells and examine whether they maintain HSC marker expression, WT mice were lethally irradiated and transplanted with CAG-EGFP donor bone marrow, then treated with CSF1Ri (7 d PLX5622 1200ppm) after a four-week recovery period to deplete microglia and induce peripheral cell infiltration. Following inhibitor withdrawal, a six-week recovery period was allowed to achieve full cell engraftment[19,89] and provide a direct age-matched point of comparison to our *Hexb*-/- BMT + CSF1Ri mice (Supplementary Fig. 6a).

Next, we conducted flow cytometry experiments with two panels tailored to identify different myeloid populations: one panel for progenitor populations, and one panel to subdivide differentiated myeloid populations. With the progenitor identification panel (DAPI, CD3, CD19, CD45, Cd11b, Sca-1, cKit, CD34), we were able to identify that ~95% of GFP + cells were CD45hiCd11b+Sca-1-cKit-, indicating that very few cells maintain HSC/progenitor markers and are consistent with either a monocyte, macrophage, or monocytic myeloid-derived suppressor cell (M-MDSC) identity[90] (Supplementary Fig. 6e, f). We also unexpectedly identified a small population of GFP+CD45hiCd11b- cells that were also negative for Sca-1, cKit, and CD34, indicating the presence of a small number of non-myeloid immune cells in the BMT + CSF1Ri brain. Percentages of CMPs/GMPs, HSCs, and hematopoietic progenitors were nominal (<1%). To further parse the identity of these non-progenitor cells, we used a second panel targeted for the identification of differentiated myeloid cells (DAPI, CD3, CD19, CD45, Cd11b, Ly6C, Ly6G, CCR2, CD16/32) (Supplementary Fig. 6g). We found that within the GFP+CD45hiCd11b+ population, again a majority of cells shared a similar profile, with 95% being CD45hiCd11b+Ly6C-Ly6G-CCR2-, consistent with a macrophage or non-classical/Ly6Clo monocyte identity (Supplementary Fig. 6h). Interestingly, a previous study has shown that Ly6Clo is more consistent with a non-inflammatory/resident macrophage phenotype than an inflammatory phenotype[91]. We also observed nominal percentages (<1-2%) of classical monocytes/M-MDSCs, CCR2- MDSCs/ transitional monocytes, and polymorphonuclear myeloid-derived suppressor cells (PMN-MDSCs)/neutrophils[92].

Next, we used CX3CR1-GFP/CCR2-RFP reporter mice to further explore the identify of bone marrow-derived engrafted cells in BMT + CSF1Ri treated mice. In this, we transplanted an additional cohort of mice with bone marrow from CX3CR1-GFP/CCR2-RFP reporter mice[93,94] (Supplementary Fig. 6i) and show high engraftment of bone marrow-derived CX3CR1+ cells with no/minimal CCR2 cell engraftment (Supplementary Fig. 6j). We observe that ~80% of myeloid cells in the brain (i.e., IBA1+ cells) express CX3CR1 (Supplementary Fig. 6k), indicating that the majority of myeloid cells in the BMT + CSF1Ri brain resemble tissue macrophage-like cells rather than infiltrating CCR2 monocyte-like cells.

To further refine the identity of the donor-derived cells, we performed an additional CosMx single-cell spatial transcriptomic analysis on WT control mice transplanted with CAG-EGFP bone marrow at age 4–6 weeks, in which one group was given no further treatment (WT BMT) and one group was treated with CSF1Ri for 7 days between 6 and 8 weeks (WT BMT + CSF1Ri). Both mice were sacrificed at 16 weeks (Supplementary Fig. 7a). For this experiment, we used a custom probe set containing a *Gfp* probe for definitive identification of the bone marrow-derived cells. Here, we show that there is a distinct myeloid cell subcluster which expresses *Gfp* (Myeloid 2) along with a small number of cells in an endothelial cell cluster (Supplementary Fig. 7b, c). We also show that the *Gfp*+ myeloid population is specific to the BMT + CSF1Ri group (Supplementary Fig. 7d), validating their identity as the population of GFP+ cells present in the brains of BMT + CSF1Ri treated mice. In comparison to microglia in WT controls (i.e., Myeloid 1, which express canonical microglial genes, including *Sall1*, *P2ry12*, *Siglech*, and *Tmem119*), *Gfp*+ engrafted cells (i.e., Myeloid 2) are enriched in several genes that we identified as monocyte signature genes[19], including *Apobec1*, *Lyz2*, *Mrc1*, *Lilrb4*, and *Msr1*, as well as genes expressed by macrophages and disease-associated myeloid cells (*Mrc1*, *Msr1*, *Cd68*, *Trem2*, *Tyrobp*, *C1qa/b/c*, *Ctss*, *Apoe*, *Cx3cr1*, *Itgam* [CD11b], *Ptprc* [CD45], including genes associated with lysosomal/phagocytic activity (e.g., *Lyz1/Lyz2*, *Ctsb*, *Ctsd*, *Ctsz*) and macrophage immune modulation and tissue repair (e.g., *Tgfbr1*, *Tgfb1*) (Supplementary Fig. 7e). Of note, previous studies have shown that non-parenchymal macrophages express *Mrc1* and *Msr1*[95,96]. Taken together, these data provide evidence that bone marrow-derived engrafted cells in BMT + CSF1Ri treated mice are of a macrophage/non-classical monocyte identity.

## Microglial replacement reverses genetic changes

Having confirmed that microglial replacement via BMT and CSF1Ri leads to functional rescue in *Hexb*-/- mice, we next utilized spatial transcriptomic analysis to examine whether the delivery of *Hexb*-sufficient myeloid cells to the CNS can reverse the SD-associated gene expression changes observed in *Hexb*-deficient mice. Three brains from the WT control group and each *Hexb*-/- group (*Hexb*-/- control, *Hexb*-/- BMT, *Hexb*-/- BMT + CSF1Ri) were sagittally sectioned at 10 μm and imaged as described previously (632 total FOVs, ~ 53 FOVs per brain section) (Fig. 4a). Here, 389,585 cells were captured with a mean transcript count of ~ 800 transcripts per cell. Unsupervised cell clustering identified 38 transcriptionally distinct subclusters, and clusters and cell types were annotated as described in Fig. 1 (Fig. 4b and Supplementary Fig. 8a).

We first compared *Hexb* transcript expression in myeloid cells in order to assess whether donor BM cells that engrafted the brains indeed expressed *Hexb*. We detected *Hexb* transcripts in both microglia and monocyte/macrophage populations in the WT and *Hexb*-/- BMT + CSF1Ri mice, confirming the presence of *Hexb* transcripts in donor-derived cells (Fig. 4c). Assessing other broad cell types, we detected minimal *Hexb* transcripts in OPCs, oligodendrocytes, astrocytes, neurons, endothelial cells, T cells, and pericytes, as observed previously. These data reinforce a myeloid-specific *Hexb* expression pattern and identifies monocytes/macrophages as a cell type that can express *Hexb* within the CNS.

Analysis of cell cluster proportions revealed that the second-largest myeloid subcluster, identified as monocytes/macrophages (Mono/mac) by expression of canonical marker genes (high *Lyz1/2*, *H2-Aa*, *Cd74*; low *Tmem119*)[37–39,97], was drastically expanded in the *Hexb*-/- BMT + CSF1Ri group. Plotting the Mono/mac subcluster in XY space in the *Hexb*-/- BMT + CSF1Ri brain showed numerous cells scattered throughout the parenchyma (Fig. 4e), a spatial pattern consistent with the location and distribution of BMDMs as indicated by GFP staining. Aside from the Mono/mac subcluster and the vascular broad cell type, the number of cells within the broad cell types (Supplementary Fig. 8d) and cellular subclusters (Fig. 8b) were largely consistent between groups. These data indicate that BMT + CSF1Ri treatment induces the infiltration of cells that express *Hexb* in the CNS.

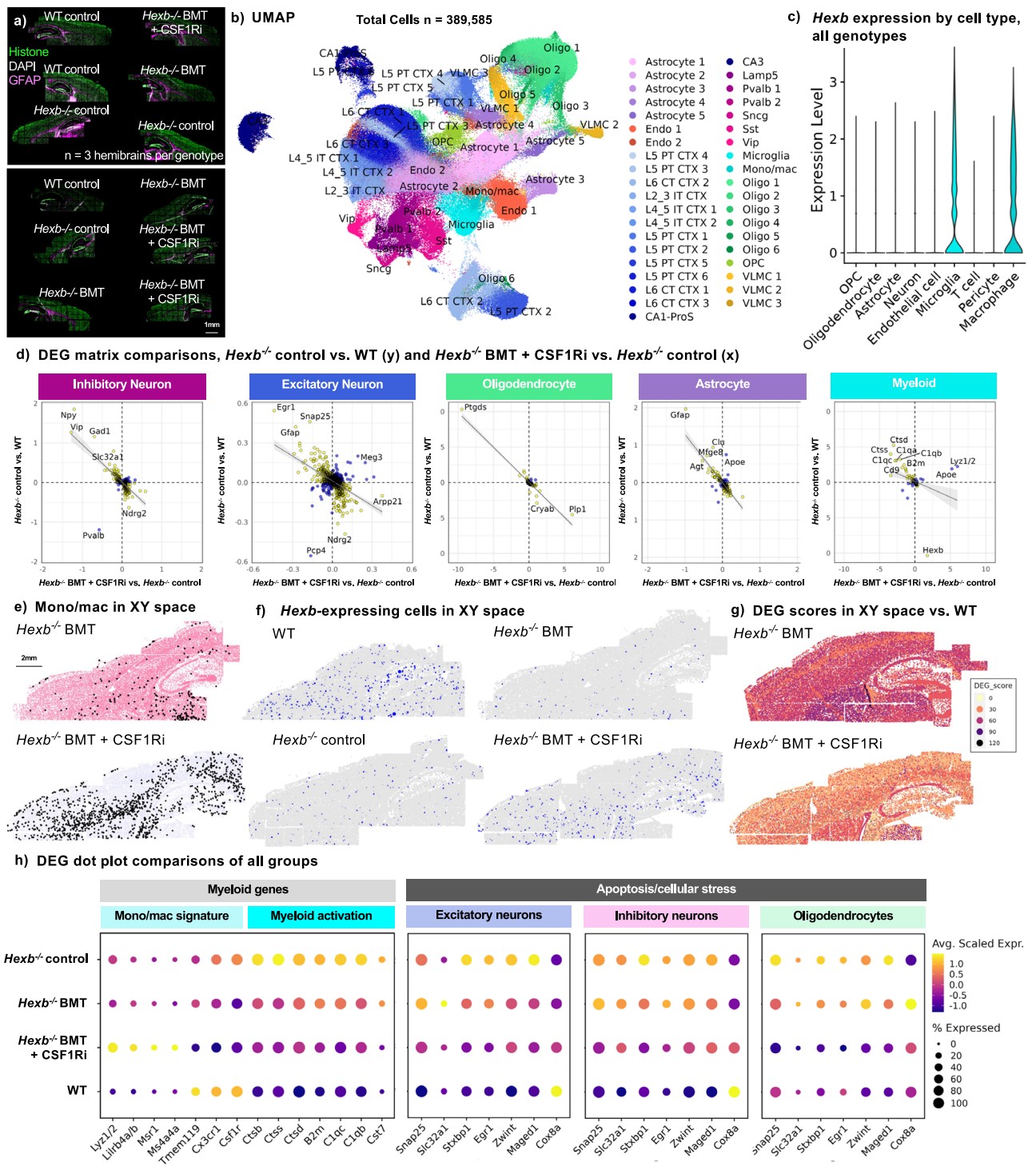

To assess whether gene expression changes associated with loss of *Hexb* were reversed with microglial replacement, we performed DGE analysis on all subclusters and broad cell types (Supplementary Figs. 10–12) between experimental group pairs. DGE analyses revealed shifts in gene expression between *Hexb⁻/⁻* BMT + CSF1Ri and *Hexb⁻/⁻* control mice in many broad cell types (Supplementary Fig. 10a). To evaluate whether the specific genes altered by Hexb deficiency (either upregulated or downregulated in *Hexb⁻/⁻* control versus WT mice) were rescued or reversed by BMT and CSF1Ri treatment (comparing *Hexb⁻/⁻* BMT + CSF1Ri with *Hexb⁻/⁻* control mice), we generated comparison matrices to assess expression differences in these two pairs (Fig. 4d). We were especially interested in whether certain disease-associated

genes would display reversed directionality, i.e., whether genes that were downregulated in *Hexb⁻/⁻* control mice vs. WT mice would be upregulated in *Hexb⁻/⁻* BMT + CSF1Ri vs. *Hexb⁻/⁻* control mice and vice versa. Comparison matrices revealed that many DEGs between *Hexb⁻/⁻* control and WT mice were significantly changed in the opposite direction between *Hexb⁻/⁻* BMT + CSF1Ri and *Hexb⁻/⁻* control mice. In neurons, DEGs with reversed directionality included genes associated with apoptosis and lysosomal dysfunction (*Hspa8, Egr1, Npy, Sgk*), neurodevelopment and synaptic function (*Snap25, Arpp21, Slc32a1, Gad1*), and immunomodulation (*Vip*)[54,68,73,77,79,98–101]. In oligodendrocytes, DEGs involved in apoptosis, cellular stress, and myelination (i.e., *Ptgds, Cryab, and Plp1*)[55–57] that had previously been identified

**Fig. 4 | Spatial transcriptomic analysis reveals reversal of disease-associated genetic changes following microglial replacement in SD mice. a** Image of WT, *Hexb*⁻/⁻, bone marrow transplant (BMT)-treated *Hexb*⁻/⁻, and BMT + colony-stimulating factor 1 receptor inhibitor (CSF1Ri)-treated *Hexb*⁻/⁻ groups (*n* = 3/group) imaged for cell segmentation markers histone (green), DAPI (gray), and GFAP (magenta). **b** Uniform Manifold Approximation and Projection (UMAP) of 389,585 cells across 12 brains. **c** Violin plot of *Hexb* transcript counts in broad cell types demonstrating myeloid-specific *Hexb* expression. **d** Comparison matrix scatterplot of average difference in significant (p$_{adj}$ < 0.05) differentially expressed genes (DEGs) in inhibitory neurons, excitatory neurons, oligodendrocytes, astrocytes, and myeloid cells between *Hexb*⁻/⁻ vs. WT and *Hexb*⁻/⁻ BMT + CSF1Ri vs. *Hexb*⁻/⁻. The plot shows inversely correlated genes (yellow), directly correlated genes (blue), a linear regression line, and error bands (gray) representing a 95% confidence interval. Differential gene expression analysis per cell type between groups was performed on scaled expression data using Model-based Analysis of Single-cell Transcriptomics (MAST) to calculate average difference, defined as the difference in log-scaled average expression between two groups for each broad cell type.

**e** The monocyte/macrophage subcluster (black) in *Hexb*⁻/⁻ BMT and *Hexb*⁻/⁻ BMT + CSF1Ri brains plotted in XY space. **f** *Hexb*-expressing cells (blue) plotted in XY space in WT, *Hexb*⁻/⁻, *Hexb*⁻/⁻ BMT, and *Hexb*⁻/⁻ BMT + CSF1Ri brains. Points sized by *Hexb* expression: cells with 0 transcripts not plotted, cells with 1 transcript plotted at point size=0.001, cells with 2 transcripts plotted at point size = 0.15, and cells with 3 + transcripts plotted at point size = 0.3. **g** Subclusters in XY space colored by DEG score calculated in comparison to WT controls in representative *Hexb*⁻/⁻ BMT and *Hexb*⁻/⁻ BMT + CSF1Ri brains. DEG score was calculated using the DEGs from treatment condition pairs in each subcluster by summing the absolute value of the log₂ fold change values for all significant (p$_{adj}$< 0.05) DEGs identified between WT control and *Hexb*⁻/⁻ BMT or *Hexb*⁻/⁻ BMT + CSF1Ri. Cell clusters with higher DEG scores colored with darker shades to visually represent the degree of difference in XY space. **h** Dot plot representing pseudo-bulked expression values across groups in genes related to monocyte/macrophage identity, myeloid cell activation, and apoptosis and/or cellular stress in excitatory neurons, inhibitory neurons, and oligodendrocytes.

between *Hexb*⁻/⁻ control and WT mice demonstrated a reversal of expression directionality in *Hexb*⁻/⁻ BMT + CSF1Ri versus *Hexb*⁻/⁻ control mice. In astrocytes, upregulated disease-associated DEGs *Clu*, *Mfge8*, and *Agt* in *Hexb*⁻/⁻ control mice versus WT were downregulated in *Hexb*⁻/⁻ BMT + CSF1Ri mice in comparison to *Hexb*⁻/⁻ controls. Finally, numerous myeloid subcluster genes that were upregulated in *Hexb*⁻/⁻ control versus WT mice (*Ctsd*, *Ctss*, *C1qa*, *C1qc*, *B2m*, *Cd9*) were then downregulated between *Hexb*⁻/⁻ BMT + CSF1Ri and *Hexb*⁻/⁻ control mice following replacement of microglia with BMDMs, indicating that BMT + CSF1Ri reverses disease-associated myeloid cell changes[46,49–52]. Myeloid cells in the *Hexb*⁻/⁻ BMT + CSF1Ri group also had significantly elevated *Hexb* expression compared to *Hexb*⁻/⁻ control mice. The reversal of directionality in disease-associated gene signatures observed with microglial replacement demonstrates the efficacy of this strategy in addressing SD-related phenotypes at the transcript level in glial cells and neurons.

We next sought to compare the efficacy of BMT + CSF1Ri treatment over BMT alone in reversing transcriptomic changes caused by *Hexb* deficiency. In contrast to the reversal in transcriptional changes observed in the BMT + CSF1Ri group, fewer broad cell type DEGs were reversed in directionality and/or were reversed to a lesser degree in terms of log₂fold change or adjusted *p* value in *Hexb*⁻/⁻ BMT mice in comparison to *Hexb*⁻/⁻ controls (Supplementary Fig. 10b). Plotting *Hexb*-expressing cells in XY space showed, predictably, high levels of expression in WT controls with minimal/background *Hexb* expression in *Hexb*⁻/⁻ controls, which appeared unchanged in *Hexb*⁻/⁻ BMT mice (Fig. 4f). By contrast, the *Hexb*⁻/⁻ BMT + CSF1Ri mice demonstrated a restoration of *Hexb*-expressing cells which mirrored the spatial localization of the Mono/mac subcluster. DGE analysis and DEG score calculation revealed higher DEG scores and greater overall deviation from WTs in *Hexb*⁻/⁻ BMT mouse brains; plotting DEG scores in XY space revealed higher overall DEG scores throughout the brain in *Hexb*⁻/⁻ BMT mice than *Hexb*⁻/⁻ BMT + CSF1Ri mice when each group was compared to WT controls (Fig. 4g). By performing a pseudo-bulk analysis in each broad cell type for all four groups (Supplementary Figs. 10b, 11e), we confirmed a BMDM signature in the myeloid cell population of the *Hexb*⁻/⁻ BMT + CSF1Ri group only (upregulation of monocyte/macrophage genes *Lyz1/2*, *Lilrb4a/b*, *Msr1*, *Ms4a4a*; downregulation of microglial homeostatic genes *Tmem119*, *Cx3cr1*, *Csf1r*) (Fig. 4h). Myeloid activation genes were reduced in both *Hexb*⁻/⁻ BMT groups in comparison to *Hexb*⁻/⁻ controls, though to a greater extent in *Hexb*⁻/⁻ BMT + CSF1Ri mice. Pseudobulk analysis also revealed that genes associated with apoptosis and cellular stress pathways demonstrated reversed directionality in the *Hexb*⁻/⁻ BMT + CSF1Ri group versus *Hexb*⁻/⁻ controls in excitatory neurons, inhibitory neurons, and oligodendrocytes. These genes were largely unchanged in *Hexb*⁻/⁻ BMT groups in comparison to *Hexb*⁻/⁻ controls, demonstrating a failure of BMT alone

to reverse genetic indicators of apoptotic processes. These data demonstrate similarity between WT mice and *Hexb*⁻/⁻ BMT + CSF1Ri and greater divergence from WT mice in *Hexb*⁻/⁻ BMT mice. Overall, BMT is not sufficient to reverse the majority of gene expression changes associated with the loss of *Hexb*. These findings underscore the importance of CSF1Ri-based microglial replacement in the correction of disease-associated gene expression changes in neurons, myeloid cells, oligodendrocytes, and astrocytes in *Hexb*⁻/⁻ mice.

In sum, spatial transcriptomic analysis reveals that several disease-associated gene signatures in *Hexb*⁻/⁻ mice can be reversed with microglial replacement following BMT + CSF1Ri treatment. Numerous DEGs identified between *Hexb*⁻/⁻ and WT mice were subsequently reversed in directionality between *Hexb*⁻/⁻ BMT + CSF1Ri-treated mice and *Hexb*⁻/⁻ controls, including genes related to apoptosis, myelination/demyelination, cellular stress response, inflammatory response, and endo-lysosomal function. We also observed a restoration of *Hexb* expression with the introduction of BMDMs to the *Hexb*⁻/⁻ CNS. The ability of microglial replacement to correct SD-associated changes at the molecular level further underscores the potential of this strategy to treat disease.

**Proteomic analysis shows reversal of disease-associated changes**
To further understand the effects of *Hexb* loss and microglial replacement, we performed spatial proteomic analysis using the CosMx Spatial Molecular Imager (Fig. 5a). We utilized a multi-plex 67-protein mouse neuroscience panel on four 10μm sagittal brain sections from WT control mice and all *Hexb*⁻/⁻ groups (*Hexb*⁻/⁻ control, *Hexb*⁻/⁻ BMT, *Hexb*⁻/⁻ BMT + CSF1Ri). This technique allows for detailed analysis based on protein markers while maintaining the original structure of the tissue. The panel contains markers relevant to inflammation, lysosomal function, and neurodegenerative disease. A total of 1,199,879 cells were identified and imaged for expression of protein markers. Cell segmentation was automated based on DAPI, histone, and GFAP markers, with clear separation even in densely packed regions such as the dentate gyrus (Fig. 5a). Cells were sorted into subtypes based on marker expression, and plotting in XY space demonstrated accurate identification (Fig. 5b). We identified seven neuronal subsets as well as astrocytes, neuroepithelial cells, microglia, vascular cells, and oligodendrocytes.

To assess how loss of *Hexb* affects expression of various proteins, especially those associated with lysosomal-endosomal function in the murine brain, we next performed differential protein expression (DPE) analysis between all groups in all cellular subsets (Supplementary Fig. 12b–f). We were particularly interested in differentially expressed proteins (DEPs) in neurons and myeloid cells after identifying disease-associated gene expression signatures in these cell types, and whether protein expression changes between *Hexb*⁻/⁻ and WT mice were

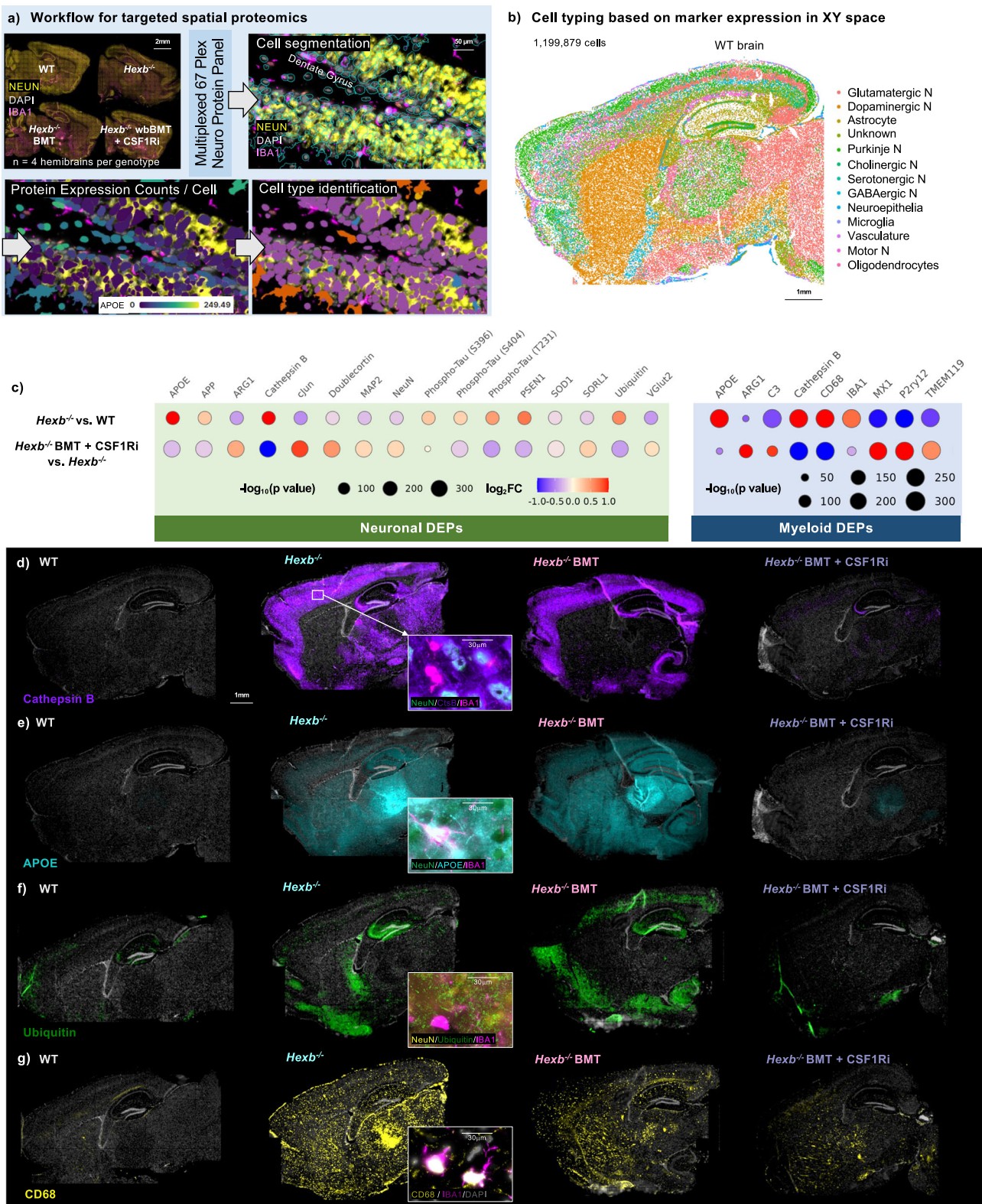

reversed between *Hexb*[-/-] BMT + CSF1Ri in comparison to *Hexb*[-/-] controls (Fig. 5c). Neurons and microglia/myeloid cells from *Hexb*[-/-] mice both had significantly higher expression of several proteins associated with dysregulation of the endosomal-lysosomal system compared to WT cells, including APOE and cathepsin B[102–105]. Cathepsin B protein was prominent in NeuN[+] neurons specifically, and expression was visible throughout the cortex, subiculum, dentate gyrus, pyramidal neurons of the hippocampus, and white matter striations of the

thalamus (Fig. 5d). APOE protein was widespread and did not appear to colocalize with any particular cell type (Fig. 5e). Neurons from *Hexb*[-/-] mice also exhibited elevated expression of several proteins associated with neurodegenerative diseases and/or lysosomal dysfunction in comparison to WT controls, such as amyloid precursor protein (APP), several species of phosphorylated tau, presenilin 1 (PSEN1), and ubiquitin[102,106–110]. Proteins associated with normal neuronal health and development, such as c-Jun, doublecortin, and MAP2, were

**Fig. 5 | Spatial proteomic analysis identifies disease-associated protein expression patterns in the SD mouse brain which are reversed with microglial replacement. a** Workflow for targeted 67-plex single-cell spatial proteomics. Fields-of-view (FOVs) are imaged with cell segmentation markers GFAP, NEUN, RPS6, and IBA1. Protein abundance is determined by quantification of fluorescently labeled oligos bound to proteins within each cell. Cell types are identified using the CELESTA algorithm, which classifies cells based using expression of marker proteins. **b** Cell types plotted in XY space in a representative WT control brain. 1,199,876 cells were captured across the four groups (WT control, *Hexb*−/− control, BMT-treated *Hexb*−/−, and BMT + CSF1Ri-treated *Hexb*−/− [*n* = 4/group]). CELESTA cell classification yielded 13 cell types, which were plotted in space to confirm accurate identification. **c** Bubble plots of differentially expressed proteins (DEPs) of interest between pairs *Hexb*−/− control vs. WT control, and BMT + CSF1Ri-treated *Hexb*−/− vs. *Hexb*−/− control in neurons and myeloid cells. Differential protein expression analysis per cell type between groups was performed on scaled expression data using Model-based Analysis of Single-cell Transcriptomics (MAST) to calculate the average difference, defined as the difference in log-scaled average expression between the two groups for each cell type. Dots are sized by *p*-value (-log$_{10}$ *p*-value) and colored by log$_2$ fold change (red indicating increased expression, blue indicating decreased expression) of each DEP. **d–g** Representative whole brain images chosen from an *n* = 4 per group of WT control, *Hexb*−/− control, BMT-treated *Hexb*−/−, and BMT + CSF1Ri-treated *Hexb*−/− brains and expanded insets showing cellular marker colocalization of proteins (**d**) Cathepsin B (purple), colocalization with NeuN$^+$ neurons (green) and not IBA1$^+$ myeloid cells (magenta); **e** Apolipoprotein E (APOE, cyan), colocalization with both NeuN$^+$ neurons (green) and IBA1$^+$ myeloid cells (magenta); **f** Ubiquitin (green), colocalization with NeuN$^+$ neurons (yellow) and not IBA1$^+$ myeloid cells (magenta); **g** CD68 (yellow), colocalization with IBA1$^+$ myeloid cells (magenta) with DAPI (gray) illustrating the rescue of pathological and lysosomal phenotypes by combined BMT and CSF1Ri treatment.

downregulated in *Hexb*−/− mice versus WT controls; when dysregulated, many of these proteins have also been associated with apoptosis[111–113]. Microglia/myeloid cells in *Hexb*−/− mice exhibited significantly elevated expression of CD68, a lysosomal marker linked to microglial/myeloid cell activation[114]. In line with myeloid cell activation, we observe colocalization of CD68 with IBA1$^+$ myeloid cells in *Hexb*−/− brains, including in the pia mater layer of the meninges (Fig. 5g). We also observed elevated APOE and CD68 deposition in the thalamus of *Hexb*−/− mouse brains. These data are in agreement with the region-specific effects identified by spatial transcriptomic analysis and data from human SD patients[81–83]. Microglia/myeloid cells in the *Hexb*−/− brain also showed marked reductions in homeostatic microglial proteins P2RY12 and TMEM119 in comparison to cells from WT control mice[97,115]. These findings provide further insight into the various myeloid and neuronal cell disruptions when myeloid *Hexb* expression is lost, manifesting as lysosomal abnormalities, neuronal dysregulation, and polarization of microglia from homeostatic to activated phenotypes.

We were next interested in whether microglial replacement via BMT + CSF1Ri led to reversal of protein expression changes associated with loss of *Hexb*. Indeed, all DEPs identified in neurons and myeloid cells between *Hexb*−/− control mice and WT controls exhibited reversed directionality and significant differences in expression between *Hexb*−/− BMT + CSF1Ri and *Hexb*−/− controls (Fig. 5c). Visually, expression of notable DEPs cathepsin B, APOE, ubiquitin, and CD68 was partially reduced in *Hexb*−/− BMT-treated mice (Fig. 5d–g). In the *Hexb*−/− BMT group, the cathepsin B phenotype was only corrected in white matter striations in the thalamus (Fig. 5d). By contrast, the overexpression of these proteins was completely or near-completely eliminated in the *Hexb*−/− BMT + CSF1Ri group. These data complement the findings from spatial transcriptomic analysis and demonstrate the BMT + CSF1Ri-induced microglial replacement can correct disease-associated protein expression patterns relevant to myeloid activation, lysosomal abnormalities, and neurodegenerative pathways. Furthermore, these data indicate that BMT + CSF1Ri improves upon the partial reductions in disease-associated protein expression achieved by BMT alone.

## BMT + CSF1Ri rescues SD CNS pathologies

Previous studies have shown that BMT prolongs lifespan and slows functional deterioration in *Hexb*−/− mice, but fails to prevent disease pathology, especially in neurons (i.e., brain glycolipid storage)[11,16,116]. Having identified a reversal of disease-associated gene signatures and protein expression patterns in mice treated with BMT + CSF1Ri, we next sought to investigate the efficacy of combined BMT and CSF1Ri treatment in ameliorating CNS pathological changes in *Hexb*−/− mice. We assessed GM2 ganglioside content, the hallmark pathological manifestation of SD, using MALDI mass spectrometry on fresh-frozen whole brain homogenate samples (Fig. 6a–c). This analysis demonstrated, as expected, marked accumulation of GM2 ganglioside in untreated *Hexb*−/− control mice (Fig. 6c). BMT alone did not significantly

reduce GM2 ganglioside content in comparison to controls; however, BMT + CSF1Ri significantly reduced GM2 burden (Fig. 6c). We then performed Periodic Acid Schiff (PAS) staining, a histological detection method for glycolipids/glycoproteins, to evaluate accumulation of additional Hexβ substrates. In line with prior reports[117], *Hexb*−/− and BMT-treated *Hexb*−/− mice exhibit numerous PAS$^+$ deposits throughout the brain parenchyma, which are consistent with the shape and size of neurons, and absent in WT animals (Fig. 7a–c). We observe a significant reduction in PAS$^+$ staining in BMT-treated compared to control *Hexb*−/− mice, indicating that BMT does partially reduce glycolipid storage, but does not resolve this pathology (Fig. 7c). Notably, PAS$^+$ deposits were undetectable in BMT + CSF1Ri-treated *Hexb*−/− mice. Taken together with the GM2 ganglioside data, these findings indicate that replacement of *Hexb*-deficient microglia with *Hexb*-sufficient BMDMs can partially to fully rescue the pathological accumulation of glycolipids in the murine SD brain.

Next, we assessed the effects of *Hexb* deficiency, BMT, and microglial replacement on lysosomal alterations in neurons by co-staining for LAMP1, a marker for lysosomes and autophagic organelles, and NeuN, a marker for neurons. Immunostaining revealed extensive LAMP1$^+$ accumulation (Fig. 7d) that colocalized with neurons (Fig. 7e–g) in *Hexb*−/− mouse brains, indicative of a disruption in the endosomal-lysosomal system in murine SD; in line with this, previous studies have shown that LAMP1 is degraded by Hexβ[37–41]. Here, we show that BMT alone did not significantly reduce LAMP1$^+$ staining in the *Hexb*−/− brain. However, BMT + CSF1Ri treatment led to a drastic and significant reduction in LAMP1$^+$ staining (Fig. 7g). These findings provide evidence that microglial replacement can resolve abnormal lysosomal phenotypes within neurons in *Hexb*−/− mice.

Having identified that several inhibitory neuronal subsets are affected by *Hexb* deficiency during spatial transcriptomics analysis, we next screened for morphological abnormalities and cell loss in parvalbumin (PV) neurons, a marker for a subtype of inhibitory neurons. We did not observe a significant loss in the number of cortical NeuN$^+$ or PV$^+$ cells in *Hexb*−/− mice, but we did detect neuronal abnormalities: PV$^+$ neurons in the *Hexb*−/− control group demonstrated unusual puncta within the cell body, indicative of vacuolization (Fig. 7h, i). These abundant vacuoles were consistent with previous reports in SD and have previously been identified as enlarged, dysfunctional lysosomes[23,118,119]. Vacuoles were also present in the *Hexb*−/− BMT group but significantly reduced in the *Hexb*−/− BMT + CSF1Ri group, which did not differ from WT BMT + CSF1Ri controls (Fig. 7j). Finally, we stained for GFAP to assess any changes in astrocyte activation (Fig. 7k). We detected a significant increase in total GFAP + staining volume in *Hexb*−/− mice versus WT controls, indicative of astrocyte activation/astrogliosis, that was relatively elevated in some animals following BMT (Fig. 7l). BMT + CSF1Ri, however, significantly reduced GFAP + staining volume in *Hexb*−/− mice in comparison to BMT alone. Altogether, we demonstrate correction of several CNS pathologies and

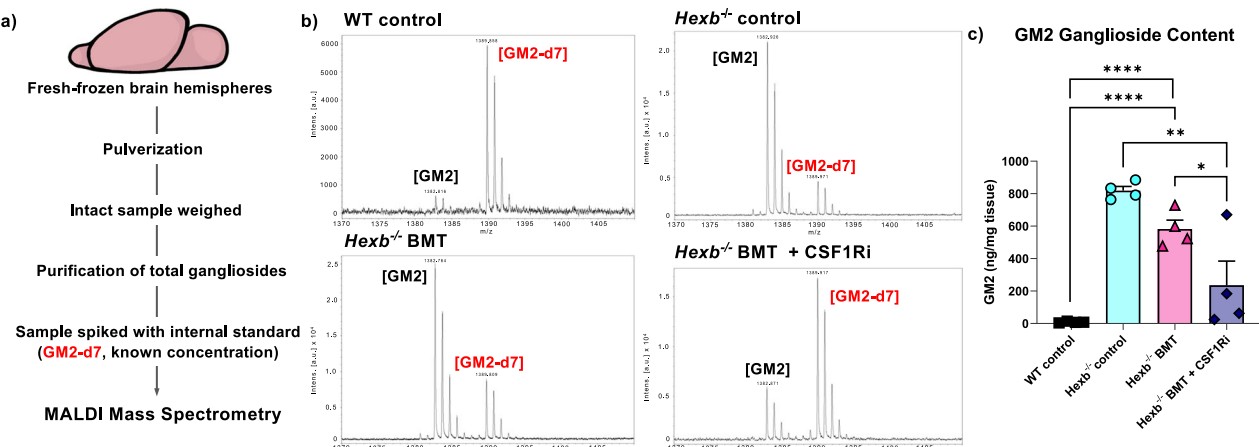

**Fig. 6 | MALDI mass spectrometry measurements show clearance of GM2 ganglioside pathology in brain extracts with microglial replacement.**
**a** Schematic of the MALDI mass spectrometry protocol. $n = 4$ per group.
**b** Representative mass spectrums displaying the ion relative abundance versus mass-to-charge ratio (m/z) of GM2 ganglioside (first peak) in relation to a known concentration of internal standard (GM2-d7, second peak) from homogenized fresh frozen brain samples from wildtype (WT), $Hexb^{-/-}$, bone marrow transplant (BMT)-treated $Hexb^{-/-}$, and BMT + colony-stimulating factor 1 receptor inhibitor (CSF1Ri)-treated $Hexb^{-/-}$ mice. **c** Bar graph of quantification of GM2 ganglioside content measured in ng of GM2 per total mg of tissue sample from WT, $Hexb^{-/-}$, BMT-treated $Hexb^{-/-}$, and BMT + CSF1Ri-treated $Hexb^{-/-}$ mice. Source data are provided as a Source Data file. Groups compared by one-way ANOVA with Tukey's post-hoc test. $p$-values are as follows: WT vs. $Hexb^{-/-}$, $< 0.0001$; WT vs. $Hexb^{-/-}$ BMT, 0.0014; $Hexb^{-/-}$ vs $Hexb^{-/-}$ BMT + CSF1Ri, 0.0012; $Hexb^{-/-}$ BMT vs $Hexb^{-/-}$ BMT + CSF1Ri, 0.0440.

abnormalities with microglial replacement in the SD CNS, reiterating the therapeutic potential of this treatment paradigm over traditional BMT approaches and suggesting that infiltrating $Hexb$-sufficient BM-derived myeloid cells can improve neuronal pathologies and astrocyte activation in SD.

### BMT alleviates peripheral SD pathologies

As SD is not limited to the CNS, we next profiled the consequences of our treatments outside of the CNS to assess the total-body efficacy of $Hexb$-sufficient myeloid cell replacement. We first performed histological analysis of the liver, an organ which exhibits high accumulation of GM2 gangliosides and other glycolipids in SD. Staining for GFP identified prominent deposition of donor bone marrow-derived GFP⁺ cells in the livers of all BMT groups (Fig. 8a, b). There were no significant differences in GFP⁺ cell counts between BMT alone and BMT + CSF1Ri in either WT or $Hexb^{-/-}$ mice. Having shown that liver myeloid cells were replaced following BMT treatment, we next stained for LAMP1 to assess endosomal-lysosomal abnormalities. Here, we observed a significant increase in LAMP1⁺ staining in the livers of $Hexb^{-/-}$ control mice compared to WT mice, as in the CNS, which was abolished in both the $Hexb^{-/-}$ BMT and $Hexb^{-/-}$ BMT + CSF1Ri groups (Fig. 8c, d). Finally, we performed a PAS stain and found PAS⁺ deposits throughout the liver parenchyma in $Hexb^{-/-}$ control animals, which were eliminated in both BMT and BMT + CSF1Ri groups (Fig. 8e, f). Together, these findings indicate that BMT alone is sufficient to improve pathological hallmarks in the SD liver, in alignment with previous reports[116].

In addition to immunohistochemical analysis of the liver, we also collected blood plasma to assess the levels of neurofilament light chain (NfL), a well-established biomarker of neurodegeneration that correlates with axonal damage[120]. Previous studies have shown that NfL is increased in human SD patients[121]. Here, we demonstrate that $Hexb^{-/-}$ control mice display significantly higher concentrations of plasma NfL than WT mice, signifying axonal damage (Fig. 8g). Notably, NfL was significantly reduced in both $Hexb^{-/-}$ BMT and $Hexb^{-/-}$ BMT + CSF1Ri-treated mice compared to $Hexb^{-/-}$ mice, indicative of a reduction in axonal damage in both treatment contexts. Piccolo multi-chemistry analysis of plasma also demonstrated a significant alteration in several circulating lipids/enzymes. In comparison to WT controls, plasma from $Hexb^{-/-}$ control mice exhibited significantly lower concentrations of total

cholesterol (Fig. 8h) and high-density lipoprotein (HDL) cholesterol (Fig. 8i), often referred to colloquially as good cholesterol. Both cholesterol abnormalities were ameliorated with BMT + CSF1Ri treatment in $Hexb^{-/-}$ BMT + CSF1Ri mice compared to $Hexb^{-/-}$ control mice (Fig. 8h, i). Interestingly, we observed a significant elevation in the concentration of alanine aminotransferase (ALT), a liver enzyme which increases in blood plasma following acute liver injury[122], in $Hexb^{-/-}$ BMT mice in comparison to WT BMT mice; this was significantly reduced in $Hexb^{-/-}$ BMT + CSF1Ri mice (Fig. 8j). This elevation in ALT was not present in any other groups. Overall, we report significant normalization in the concentrations of several plasma biomarkers of disease with BMT-based treatments in $Hexb^{-/-}$ mice. These results highlight the benefits of a total-body intervention in SD, rather than a CNS-specific treatment strategy, which would not address SD-related pathology in other organ systems.

### Hexβ is restored with microglial replacement

Following confirmation that $Hexb$-sufficient BMDMs engrafted in the murine SD CNS are able to resolve substrate accumulation and lysosomal abnormalities within neurons, we were interested in whether Hexβ activity was restored the brains of treated mice. While many lysosomal enzymes are only active within the lysosome, previous studies have indicated that Hexβ is enzymatically active outside of the it, including in the extracellular space[123–125]. We therefore utilized a Hexβ activity assay[126] to assess enzyme activity in two protein fractions acquired from frozen brain hemispheres from WT control, $Hexb^{-/-}$ control, $Hexb^{-/-}$ BMT, and $Hexb^{-/-}$ BMT-treated mice.

We first homogenized pulverized brains in a high-salt, detergent-free buffer and collected supernatant to extract salt-soluble proteins while minimizing cell lysis. Typically, efficient dissolution of the cell membrane requires a detergent[127]; therefore, the salt-soluble fraction is likely enriched for extracellular proteins. We then resuspended the pellet in a detergent-containing buffer to lyse cells and extract total protein from the tissue. We detected Hexβ in both fractions in WT mice (Fig. 9b, c). Upon assessing activity in each $Hexb^{-/-}$ group, we found minimal activity in both fractions from $Hexb^{-/-}$ control brains, with no significant increase in $Hexb^{-/-}$ BMT-treated brains in either fraction. We also did not observe a significant difference between $Hexb^{-/-}$ control mice and $Hexb^{-/-}$ BMT + CSF1Ri mice in the detergent-soluble fraction (Fig. 9c). However, in the salt-soluble fraction, we observed significantly

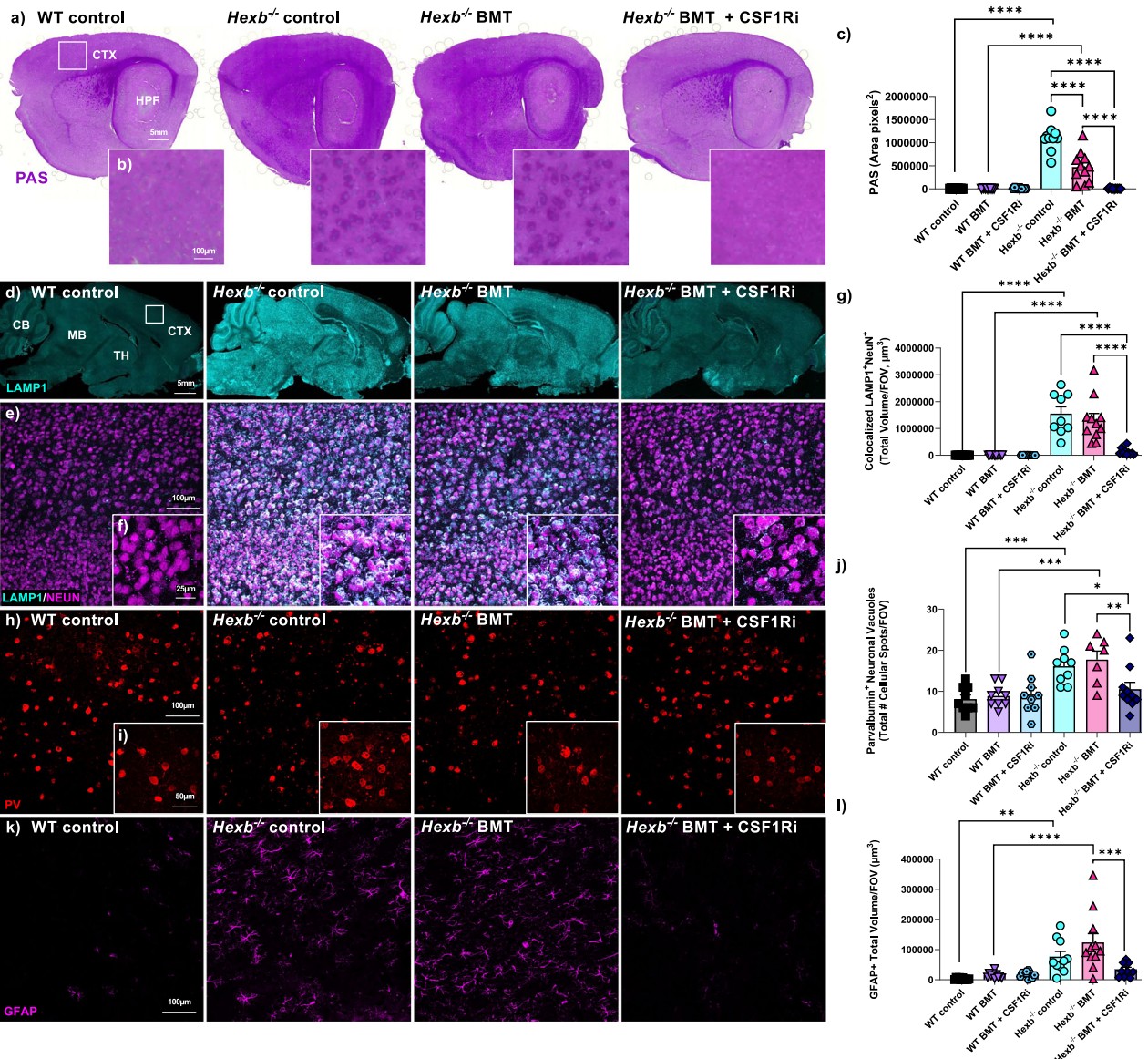

**Fig. 7 | Brain pathological changes associated with loss of *Hexb* are rescued following combined BMT and CSF1Ri treatment. a** Representative sagittal brain sections and (**b**) brightfield cortex images of Periodic acid Schiff (PAS, purple) staining, which detects glycolipids, in the brains of wildtype (WT), *Hexb*⁻/⁻, bone marrow transplant (BMT)-treated *Hexb*⁻/⁻, and BMT + colony-stimulating factor 1 receptor inhibitor (CSF1Ri)-treated *Hexb*⁻/⁻ mice. **c** Bar graph quantification of PAS staining in the cortex of all groups within field of view (FOV). $p < 0.0001$ for all significant comparisons. **d** Representative sagittal brain sections stained for lysosomal-associated membrane protein 1 (LAMP1, cyan), in WT, *Hexb*⁻/⁻, *Hexb*⁻/⁻ BMT, and *Hexb*⁻/⁻ BMT + CSF1Ri. CTX, cortex; HPF, hippocampal formation; CB, cerebellum; MB, midbrain; TH, thalamus. **e** Representative confocal images and (**f**) higher resolution images of LAMP1 (cyan) and NeuN (magenta), a marker for neurons, in the cortex of WT, *Hexb*⁻/⁻, *Hexb*⁻/⁻ BMT, and *Hexb*⁻/⁻ BMT + CSF1Ri showing colocalization (white) of LAMP1⁺ within NeuN⁺ neurons. **g** Bar graph of colocalized LAMP1⁺ and NeuN⁺ staining volume in all groups within FOV. $p < 0.0001$ for all significant comparisons. **h** Representative confocal images and (**i**) higher

resolution cortex images in WT, *Hexb*⁻/⁻, *Hexb*⁻/⁻ BMT, and *Hexb*⁻/⁻ BMT + CSF1Ri mice stained for parvalbumin (PV, red) showing the presence of vacuoles within PV⁺ cells. **j** Bar graph of quantification of vacuoles within PV⁺ neurons within FOV for all groups. *p*-values as follows: WT vs. *Hexb*⁻/⁻, 0.0004; WT BMT vs. *Hexb*⁻/⁻ BMT, 0.0002; *Hexb*⁻/⁻ vs *Hexb*⁻/⁻ BMT + CSF1Ri, 0.0215; *Hexb*⁻/⁻ BMT vs *Hexb*⁻/⁻ BMT + CSF1Ri, 0.0049. **k** Representative confocal cortex images in WT, *Hexb*⁻/⁻, *Hexb*⁻/⁻ BMT, and *Hexb*⁻/⁻ BMT + CSF1Ri mice stained for Glial Fibrillary Acidic Protein (GFAP, magenta). **l** Bar graph of quantification of total GFAP + volume per FOV in the cortex all groups. *p*-values as follows: WT vs. *Hexb*⁻/⁻, 0.0013; WT BMT vs. *Hexb*⁻/⁻ BMT, < 0.0001; *Hexb*⁻/⁻ BMT vs *Hexb*⁻/⁻ BMT + CSF1Ri, 0.0002. Source data are provided as a Source Data file. Data represented as mean ± SEM ($n = 9$ WT BMT + CSF1Ri; $n = 10$ WT, WT BMT, *Hexb*⁻/⁻, and *Hexb*⁻/⁻ BMT + CSF1Ri; $n = 11$ *Hexb*⁻/⁻ BMT); groups compared by two-way ANOVA with Tukey's post-hoc test to examine biologically relevant interactions unless otherwise noted; *$p < 0.05$, **$p < 0.01$, ***$p < 0.001$, ****$p < 0.0001$).

---

increased Hexβ activity in the *Hexb*⁻/⁻ BMT + CSF1Ri group in comparison to both *Hexb*⁻/⁻ control and *Hexb*⁻/⁻ BMT groups. These data indicate that microglial replacement in *Hexb*⁻/⁻ mice partially restores Hexβ activity in the salt-soluble, extracellular enriched fraction. This finding further highlights the potential efficacy of a microglial replacement approach for treating SD in reconstituting the enzyme in the CNS, while also

demonstrating that full enzyme reconstitution to WT levels is not necessary to correct pathological hallmarks.

**Primary microglia secrete Hexβ under physiological conditions**
Given the restoration of activity in an extracellularly enriched protein fraction and the ability of engrafted BMDMs to correct a litany of

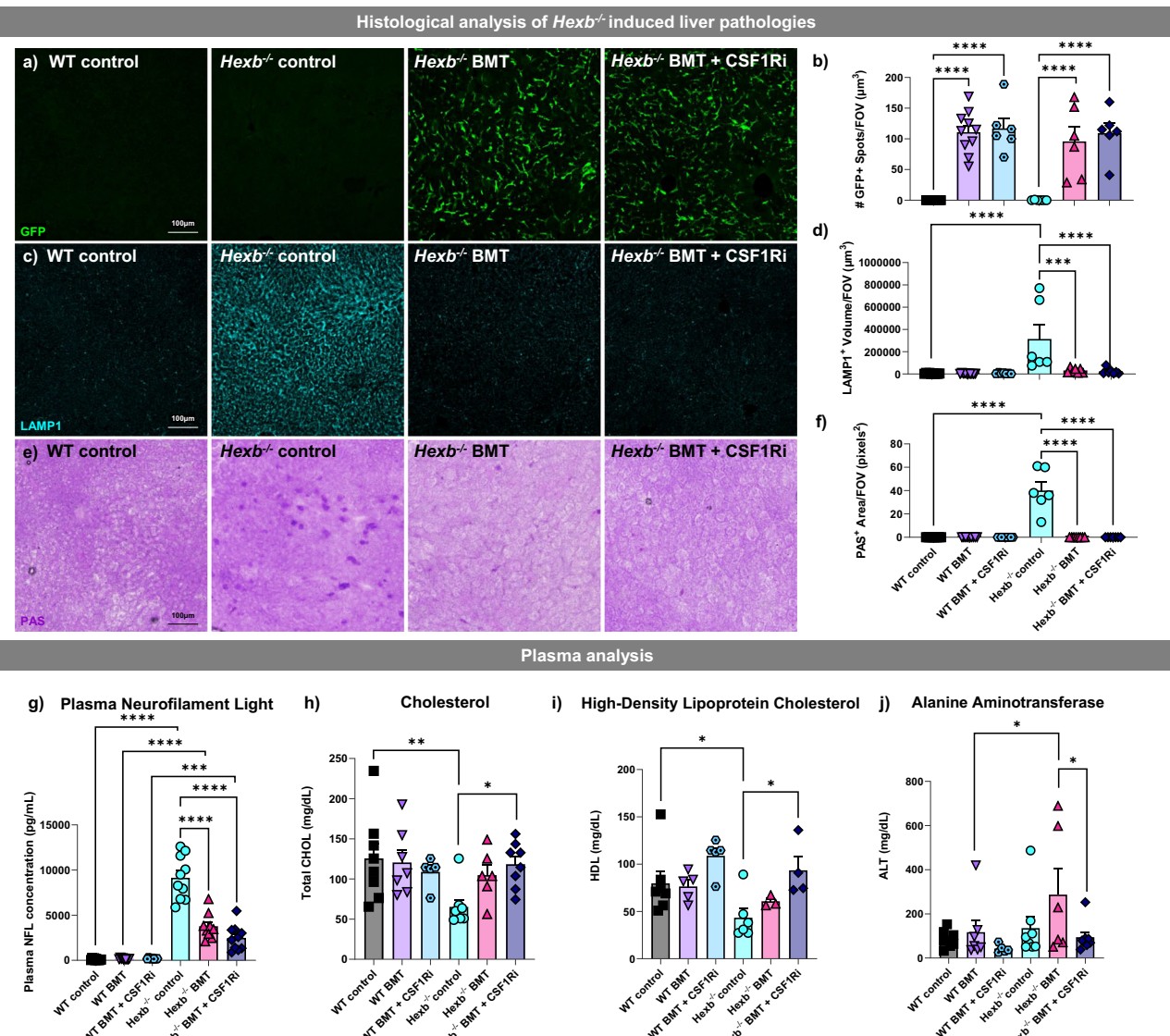

**Fig. 8 | Changes associated with loss of *Hexb* outside of the central nervous system are rescued with BMT. a–f** Representative confocal images and quantifications of liver sections from WT, *Hexb*−/−, bone marrow transplant (BMT)-treated *Hexb*−/−, and BMT + colony-stimulating factor 1 receptor inhibitor (CSF1Ri)-treated mice immunolabeled for (**a**) green fluorescent protein (GFP, green) quantified for (**b**) number of GFP+ cells (spots) (*p* < 0.0001 for all significant comparisons), **c** lysosomal-associated membrane protein 1 (LAMP1, cyan) (**d**) quantified for LAMP1+ volume (p ≤ 0.0001 for all significant comparisons), and (**e**) Periodic acid Schiff (PAS, purple), which detects glycolipids, **f** quantified by area (*p* < 0.0001 for all significant comparisons). **g** Measurement of plasma neurofilament light (NfL) in all groups. *p* values as follows: WT vs. *Hexb*−/−, < 0.0001; WT BMT vs. *Hexb*−/− BMT, < 0.0001; WT BMT + CSF1Ri vs. *Hexb*−/− BMT + CSF1Ri, 0.0002; *Hexb*−/− vs. *Hexb*−/− BMT, < 0.0001; *Hexb*−/− vs *Hexb*−/− BMT + CSF1Ri, < 0.0001. **h–j** Measurement of (**h**) total plasma cholesterol (CHOL) concentration (*p*-values: WT vs. *Hexb*−/−, 0.0019; *Hexb*−/−

vs *Hexb*−/− BMT + CSF1Ri, 0.0155; *n* = 5 WT BMT + CSF1Ri, *n* = 6 *Hexb*−/− BMT, *n* = 7 WT BMT, *n* = 8 WT, *Hexb*−/−, *Hexb*−/− BMT + CSF1Ri), (**i**) high-density lipoprotein (HDL) cholesterol (*p*-values: WT vs. *Hexb*−/−, 0.0144; *Hexb*−/− vs *Hexb*−/− BMT + CSF1Ri, 0.0120; n = 3 *Hexb*−/− BMT, *n* = 4 *Hexb*−/− BMT + CSF1Ri, *n* = 5 WT BMT, WT BMT + CSF1Ri, *n* = 6 *Hexb*−/−, *n* = 7 WT), and (**j**) alanine aminotransferase (ALT) (*p*-values: WT vs. *Hexb*−/−, 0.0373; *Hexb*−/− vs *Hexb*−/− BMT + CSF1Ri, 0.0400; *n* = 5 WT BMT + CSF1Ri, *n* = 6 *Hexb*−/− BMT, *n* = 7 WT BMT, *n* = 8 WT, *Hexb*−/−, *Hexb*−/− BMT + CSF1Ri) in all groups. Source data are provided as a Source Data file. Data are represented as mean ± SEM; groups compared by two-way ANOVA with Tukey's post-hoc test to examine biologically relevant interactions unless otherwise noted. For liver: *n* = 6 *Hexb*−/−, *n* = 7 WT BMT + CSF1Ri, *Hexb*−/− BMT, *Hexb*−/− BMT + CSF1Ri, *n* = 10 WT, WT BMT. For NfL: *n* = 9 WT BMT + CSF1Ri, *n* = 10 WT BMT, *Hexb*−/−, *Hexb*−/− BMT + CSF1Ri, *n* = 11 WT, *Hexb*−/− BMT. *p* < 0.05, **p* < 0.01, ***p* < 0.001, ****p* < 0.0001.

disease-associated neuronal phenotypes, we theorized that myeloid cells may be supporting neuronal function through the secretion of Hexβ. Thus, we first sought to identify whether microglia release enzymatically active Hexβ protein in vitro. To accomplish this, cortices of 3 to 5-day old neonatal wildtype pups were collected, dissociated, and incubated for 14 days to generate a primary cell culture of mixed glial cells. Following this, microglia were isolated via gentle shaking and plated for 48 hours before collection of the supernatant and cell lysate (Fig. 9d). Using the Hexβ activity assay, we found that Hexβ

activity was present in the supernatant media collected from primary microglial cultures, which was not present in media alone (Fig. 9e). This result indicates that microglia passively secrete enzymatically active Hexβ under homeostatic conditions in vitro.

To further explore the mechanism of Hexβ release from myeloid cells, we investigated several of the main pathways of lysosomal enzyme secretion, including lysosomal exocytosis, exosome release, and calcium-mediated intracellular pathways[128,129]. We incubated primary microglial cultures with inhibitors of each pathway (vacuolin-1,

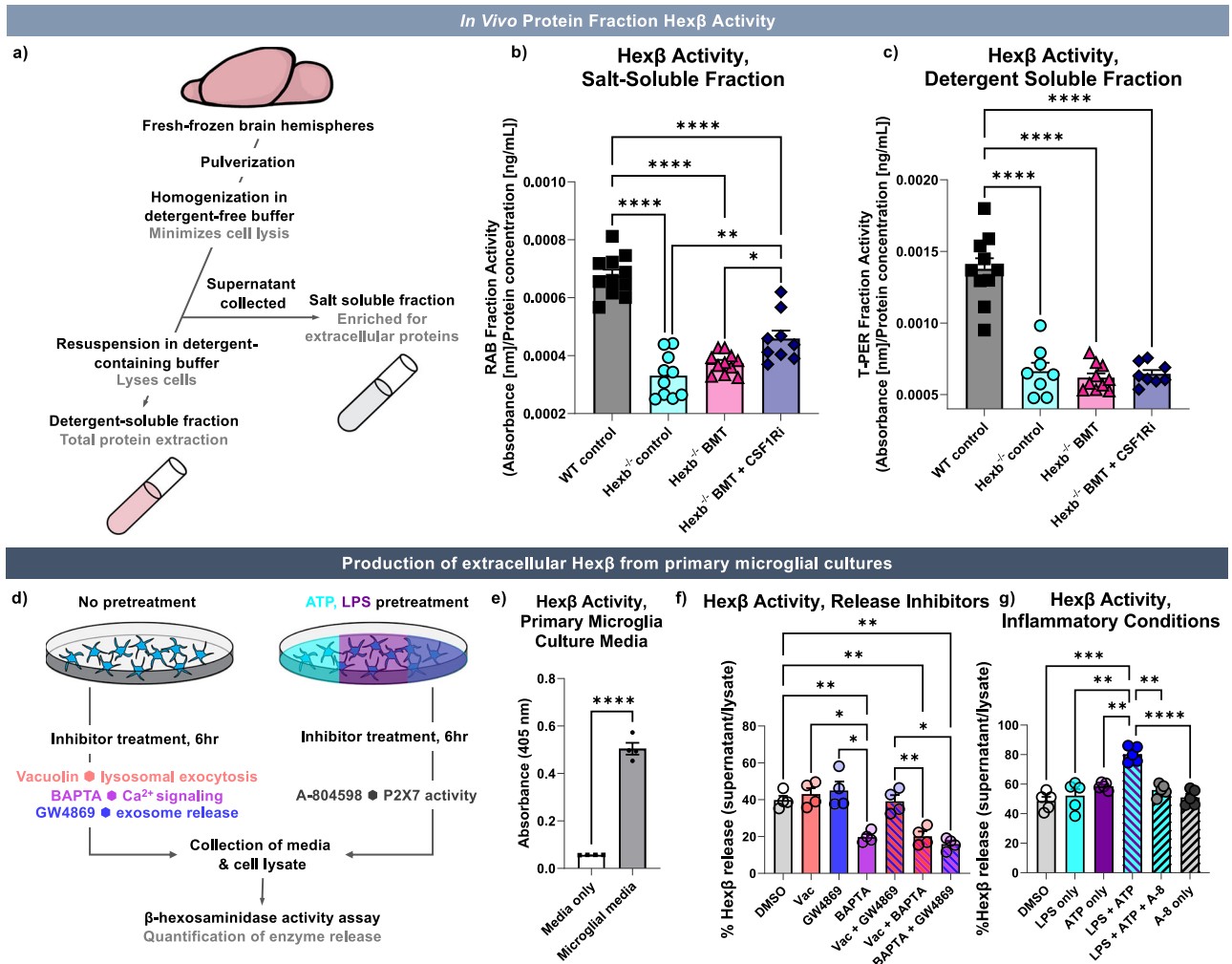

**Fig. 9 | Hexβ is restored in an extracellular-enriched brain protein fraction in *Hexb⁻/⁻* mice treated with microglial replacement, and is secreted by microglia in vitro. a** Schematic of protein fraction collection. WT, *Hexb⁻/⁻*, bone marrow transplant (BMT)-treated *Hexb⁻/⁻*, and BMT + colony-stimulating factor 1 receptor inhibitor (CSF1Ri)-treated *Hexb⁻/⁻* mouse brains were homogenized in a high-salt, detergent-free buffer to limit cell lysis/enrich for extracellular proteins. Cells were then lysed in a detergent-containing buffer. **b, c** Bar graphs of absorbance values from β-hexosaminidase (Hexβ) enzymatic activity assay normalized to protein concentration in (**b**) reassembly buffer (RAB) salt-soluble protein fraction (*p*-values: WT vs. all *Hexb⁻/⁻* groups, < 0.0001; *Hexb⁻/⁻* vs. *Hexb⁻/⁻* BMT + CSF1Ri, 0.0012; *Hexb⁻/⁻* BMT vs. *Hexb⁻/⁻* BMT + CSF1Ri, 0.0415) and (**c**) Total Protein Extraction Reagent (T-PER) buffer detergent-soluble protein fraction (*p* < 0.0001 all significant comparisons). **d** Schematic of in vitro experiments. Primary microglia incubated with inhibitors Vacuolin-1 (lysosomal exocytosis), BAPTA (calcium signaling), and/or GW4869 (exosome release), or primed with lipopolysaccharide (LPS), incubated with A-804598 (P2X7 purinergic receptor inhibitor), and/or treated with adenosine triphosphate (ATP). **e** Bar graph of Hexβ activity in media

alone and microglial culture media. Groups compared using two-tailed unpaired Student's *T* test, *p* < 0.0001. **f** Bar graph of Hexβ release, dimethyl sulfoxide (DMSO, control) or inhibitor-treated primary microglia measured as a ratio of Hexβ activity (media normalized to cell lysate). *p* values: DMSO vs. BAPTA, 0.0012; DMSO vs. Vacuolin + BAPTA, 0.0038; DMSO vs. BAPTA + GW4869, 0.0015; Vacuolin vs. BAPTA, 0.0103; GW4869 vs. BAPTA, 0.0249; Vacuolin + GW4869 vs. Vacuolin + BAPTA, 0.0014; Vacuolin + GW4869 vs. BAPTA + GW4869, 0.0102. **g** Bar graph of Hexβ release in DMSO (control) or LPS, ATP, and/or A-804598-treated primary microglia. *p*-values: DMSO vs. LPS + ATP, 0.0002; LPS vs. LPS + ATP, 0.0027; ATP vs. LPS + ATP, 0.0083; LPS + ATP vs. A-804598, < 0.0001; LPS + ATP vs. LPS + ATP + A-804598, 0.0016. Source data are provided as a Source Data file. Data represented as mean ± SEM (Protein fractions, *n* = 10 for WT, Hexb-/-, and Hexb-/- BMT + CSF1Ri and *n* = 11 for Hexb-/- BMT; activity assay *n* = 4-5 biological replicates); groups compared by one-way ANOVA with Tukey's post-hoc test to examine biologically relevant interactions unless otherwise noted; statistics derived from statistical significance, *p* < 0.05, **p < 0.01, ***p < 0.001, ****p < 0.0001.

lysosomal exocytosis; GW4869, exosome release; BAPTA-AM, calcium signaling)[130–132] for 6 hours and assessed Hexβ activity (Fig. 9f). Vacuolin-1 and GW4869-treated microglia did not exhibit significantly reduced Hexβ release in comparison to control cells, nor did cells treated with both vacuolin-1 and GW4869. These data indicate that microglial Hexβ release in vitro is not driven by lysosomal exocytosis or exosome release. However, we found that treatment with BAPTA-AM significantly reduced the release of active Hexβ by microglia compared to controls, with BAPTA-AM-treated microglia exhibiting a > 50% reduction in activity. Combined treatment with BAPTA and

vacuolin or GW4869 did not further reduce Hexβ release in comparison to BAPTA alone. These findings suggest that Hexβ is passively secreted by microglia in a calcium-dependent manner independent of lysosomal exocytosis or exosome release.

Considering that inflammatory and/or pathological conditions increase the secretion of other lysosomal enzymes, we next hypothesized that inflammation-mimicking conditions would elicit increased release of Hexβ in cultured microglia[133]. To simulate inflammatory conditions, we incubated cells with lipopolysaccharide (LPS), adenosine triphosphate (ATP), or a combination of both. LPS is frequently

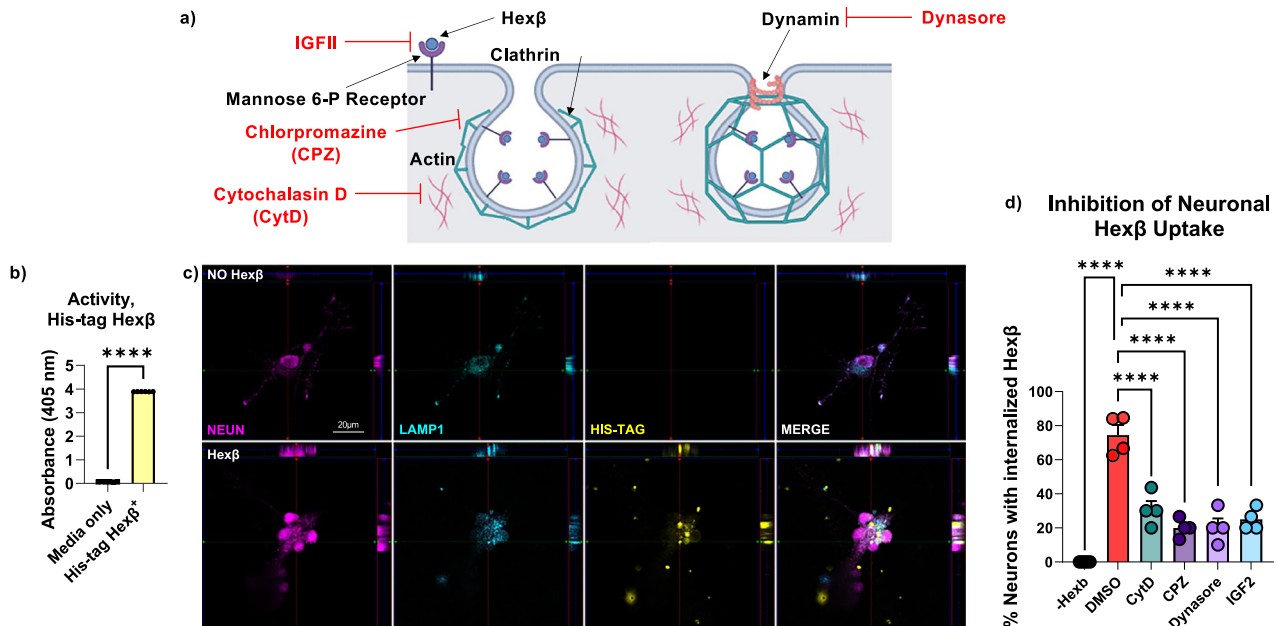

**Fig. 10 | Incorporation of extracellular Hexβ into neuronal lysosomes.**
**a** Schematic depicting the clathrin-mediated endocytosis (CME) pathway and mechanism of action of inhibitors, Cytochalasin D (CytD), Chlorpromazine (CPZ), Dynasore and IGFII, a competitive allosteric inhibitor of the cation-independent mannose-6-phosphate receptor (CI-MPR). **b** Bar graph of Hexβ activity assay measured by absorbance value in media only and media containing his-tagged recombinant Hexβ protein, demonstrating that the his-tagged Hexβ protein is enzymatically active. Groups compared using a two-tailed unpaired Student's *T* test, *p* < 0.0001. **c** Confocal images of mouse hippocampal neurons treated with media +/− his-tagged Hexβ protein immunolabeled for neurons (NeuN, magenta), lysosomal-associated membrane protein 1 (LAMP1, cyan), his-tagged Hexβ protein (HIS-TAG, yellow), and merged image showing orthogonal x/z and z/y projections at top and right of image showing colocalization of LAMP1+ and HIS-TAG+ staining

within NeuN+ neurons after treatment with Hexβ (white). **d** Bar graph representing the percentage of neurons with intracellular incorporation of his-tagged Hexβ protein as identified by orthogonal imaging of HIS-TAG staining within NeuN+ neurons showing percentage of imaged neurons without intracellular his-tagged Hexβ staining and neurons with intracellular his-tagged Hexβ staining +/− endocytosis inhibitors/CI-MPR inhibitor. *p* < 0.0001 for all significant comparisons. Schematic created in BioRender. Butler, C. (2025) https://BioRender.com/nuiylri. Source data are provided as a Source Data file. Data are represented as mean ± SEM (*n* = 4 biological replicates for neuronal cultures/experiment with at least *n* = 10 neurons imaged per condition/ per experiment); groups compared by one-way ANOVA with Tukey's post-hoc test to examine biologically relevant interactions unless otherwise noted; statistics derived from statistical significance, *p < 0.05, **p < 0.01, ***p < 0.001, ****p < 0.0001.

used to induce acute inflammation both in vitro and in vivo; it activates immune cells via activation of toll-like receptor 4 (TLR4), inducing release of inflammatory cytokines[134]. ATP accumulates in the extracellular space in inflammatory conditions, is released by damaged and/ or dying cells, and can act as a damage-associated molecular pattern to induce an inflammatory response[135]. Neither LPS or ATP-treated microglia exhibited increased Hexβ release in comparison to untreated control cells (Fig. 9g). However, cells incubated with a combination of both ATP and LPS demonstrated significantly higher levels of Hexβ release than control and LPS-treated cells. These data suggest that the combination of LPS priming and subsequent exposure to ATP, which mimics physiological inflammatory conditions, is important for the increased release of Hexβ from microglia; this is consistent with previous reports regarding other lysosomal enzymes[135–137].

A key mediator of inflammation in microglia is the ATP-sensitive P2X7 purinergic receptor, which acts as a scavenger receptor in microglial phagocytosis in the absence of stimulation[138]. Activation of P2X7 by ATP and more potent analogs causes the influx of calcium and leads to microglial activation, cytokine release, and lysosomal destabilization/leakage[139–144]. Given the efficacy of calcium-chelating BAPTA-AM in blunting Hexβ release, we theorized that increased Hexβ secretion induced by ATP + LPS treatment may be mediated by the P2X7 receptor. To test this hypothesis, we primed microglia with LPS for 3 hours and then pre-treated cultured microglia with P2X7 inhibitor A-804598 for 10 minutes before adding exogenous ATP for 20 minutes. As predicted, P2X7 inhibition significantly reduced Hexβ release

in comparison to cells treated with LPS + ATP alone (Fig. 9g). Hexβ release from cells treated with LPS + ATP + A-804598 did not significantly differ from that of control cells. A-804598 alone without ATP or LPS did not decrease Hexβ release in comparison to untreated control cells. These data indicate that the increased release of Hexβ by microglia following inflammation-mimicking LPS + ATP treatment is mediated by the P2X7 receptor, but secretion of Hexβ under homeostatic/non-inflammatory conditions is P2X7 independent.

## Mechanisms of primary neuronal Hexβ uptake

Having established that microglia secrete enzymatically active Hexβ, we were next interested in the capacity of wildtype neurons to take up Hexβ from the extracellular space and by which mechanism. To accomplish this, we acquired his-tagged recombinant mouse Hexβ protein and first confirmed its enzymatic activity using the Hexβ activity assay to assure physiological relevance (Fig. 10a). Dissociated E18 hippocampal neurons were then plated and cultured for one week before the addition of exogenous Hexβ. It is widely known that neurons can take up proteins from the extracellular space via receptor-/clathrin-mediated endocytosis (CME)[145,146]. In addition, it has been previously shown that lysosomal proteins, including the heteromeric isoform of the β-hexosaminidase enzyme (HexA) can be taken up into cells via the cation-independent mannose-6-phosphate receptor (CI-MPR) expressed on the cell surface, which may be mediated by CME[146,147]. Importantly, the mannose-6-phosphate (M6P) pathway is a cellular pathway by which most lysosomal enzymes, including Hexβ, are sorted to the endosome/lysosome, but a large percentage (~ 40%)

instead escapes and is secreted out of the cell[12,148,149]. The extracellular enzyme can then be taken up by surrounding cells via cell-surface MPRs[150,151]. To determine whether Hexb is taken up into neurons via CME in an MPR-dependent manner, we pretreated neuronal cultures with several inhibitors of endocytosis (Cytochalasin D; CytD, chlorpromazine; CPZ, and dynasore) as well as IGF-II, an allosteric inhibitor of CI-MPRs[152].

In detail, the mechanisms of action for each CME inhibitor are as follows: cytochalasin D (CytD) - inhibits actin polymerization and therefore cytoskeletal remodeling required for formation of clathrin vesicles; chlorpromazine (CPZ) - inhibits AP-2, an adapter protein essential for assembly and disassembly of clathrin lattices on cell surfaces and endosomes; and dynasore - inhibits dynamin, a protein required for scission of the clathrin coated vesicles from the plasma cell membrane. Neurons were pretreated with each inhibitor for one hour, then incubated with media containing 10 μg of his-tagged Hexβ for 24 hours. Neurons were then fixed and subsequently stained with NeuN, His-tag, and LAMP1 antibodies to identify neurons, Hexβ protein, and lysosomal membranes, respectively.

Using super-resolution confocal imaging, we observed that NeuN⁺ neurons incubated with his-tagged Hexβ showed integration of the protein into the cell bodies of ~ 74% of total neurons imaged (Fig. 10c, d; 10–15 neurons/neuron clusters imaged per treatment condition). The integration of his-tagged Hexβ in vitro indicates that neurons have the capacity to take up extracellular Hexβ. In addition, we observed colocalized staining between LAMP1 and anti-His within NeuN⁺ cell bodies (Fig. 10c). This colocalization indicates that following uptake into neurons, the Hexβ protein is localized to the lysosomal compartment. Finally, we observed significant reductions in intracellular Hexβ after treatment with each CME reuptake inhibitor as well as with IGF-II (Fig. 10d and Supplementary Fig. 15). These data suggest that neurons are capable of taking up extracellular Hexβ through CI-MPRs via CME prior to integration into the lysosome in vitro. Taken together, these data provide evidence that microglial secretion and neuronal uptake of Hexβ is a potential underlying mechanism for the rescue of neuronal-related disease phenotypes following microglial replacement in the SD CNS.

## Discussion

Our data demonstrates robust correction of SD phenotypes in Hexb⁻/⁻ model mice following BMT-based microglial replacement. We show via spatial transcriptomics that microglia and CNS-engrafted macrophages/monocytes are the only cell types that express Hexb in the wildtype CNS, then demonstrate that replacement of Hexb-deficient microglia with Hexb-sufficient BMDMs leads to the normalization of myeloid cell morphology, reversal of disease-associated changes in gene and protein expression, and clearance of enzymatic substrate pathology and lysosomal abnormalities in the brain. We also provide in vitro data which identifies the capacity of (1) microglia to release Hexβ and (2) neurons to take up Hexβ and integrate it into the lysosomal compartment. Taken together, these data provide evidence that myeloid-derived Hexβ may play a critical role in restoring normal neuronal function in SD. While our data and previous studies demonstrate that BMT alone is not sufficient to correct pathological hallmarks of SD in the CNS, our CSF1Ri-mediated approach to induce broad peripheral cell engraftment drastically improves outcomes following BMT.

In addition to demonstrating correction of the murine SD CNS, we also provide single-cell spatial transcriptomic and proteomic datasets from brain sections at an advanced disease stage in the Hexb⁻/⁻ mouse model of SD. We found that Hexb⁻/⁻ mice display a strong disease-associated gene expression signature in myeloid cells, oligodendrocytes, astrocytes, and neurons, with the latter demonstrating activation of apoptotic pathways and perturbations in neurotransmission and cellular metabolism. Many of the noted differentially expressed genes aligned with genes previously reported in RNA sequencing datasets derived from human SD and TSD patients, underscoring the strength of the model in recapitulating human disease. We also identified region-specific vulnerabilities which are highly consistent with reports from human SD patients, namely in the thalamus, white matter tracts, and cortex. Notably, loss of Hexb was associated with marked gene expression changes in the myeloid cell population, which was also the only cell type found to express Hexb in wild-type brains.

While the mechanism by which ganglioside accumulation causes neurodegeneration is not fully understood, it is clear that microglial activation and peripheral macrophage infiltration are important aspects of the pathogenesis of SD. Previous investigations have shown that GM2 ganglioside, which accumulates abundantly in various cell types in SD, activates microglia via protein kinase C and NADPH oxidase in vitro[153,154]. Moreover, microglial activation precedes neurodegeneration in Hexb⁻/⁻ mice[11], and deletion of neuroinflammatory factors such as tumor necrosis factor-α (TNF-α), activating immune receptor (FcRγ), and macrophage-inflammatory protein 1α (MIP-1 α/CCL3) reduces neurodegeneration and slightly extends the lifespan of Hexb⁻/⁻ mice[47,155,156]. These observations have led to speculation that microglial dysregulation caused by loss of Hexb and subsequent activation is the driving factor in disease progression in SD. However, it should be noted that these activation-related interventions in Hexb⁻/⁻ mice did not reduce pathological neuronal GM2 ganglioside burden. In fact, accumulating evidence counters the assumption that microglial activation is driving neuronal pathology in SD mice. In one study utilizing Hexb⁻/⁻ mice, conditional expression of human Hexβ protein exclusively in neurons extended lifespan and substantially improved neuropathology in Hexb⁻/⁻ mice, despite no reduction in microglial or astrocyte activation[157]. This finding suggests that the restoration of neuronal Hexβ is of greater importance than reducing inflammation in ameliorating disease phenotypes. Studies utilizing viral vector-based gene therapy to treat disease in various animal models of SD have demonstrated similar efficacy, even though they have shown mixed results in terms of reducing microglial activation. The viral vector-induced expression of Hexb in neurons in Hexb⁻/⁻ mice, sheep, primates, and felines has been highly effective in reducing pathology and extending lifespan; notably, these viruses do not infect microglia[119,158–163]. Finally, recent work in Hexb⁻/⁻ mice and other LSD model mice indicates that a neuron-intrinsic mechanism drives cell death and disease progression as a result of lysosomal dysfunction; this pathway does not depend on microglial involvement[9]. As a whole, these findings suggest that microglial activation/neuroinflammation alone is not sufficient to explain the neuronal pathology and neurodegeneration observed in SD. In fact, it is clear that the Hexβ protein in neurons plays an important role in maintaining cellular health and lysosomal function, despite a confirmed lack of Hexb transcript expression in neurons themselves. It is possible that the critical relationship between neurons and microglia in SD is one of enzyme provision rather than inflammation.

Our data strongly supports a relationship between myeloid-derived Hexβ and the regulation/restoration of neuronal health. The exclusive expression of Hexb transcripts in microglia and BMDMs in our spatial transcriptomic datasets corroborates previous findings identifying it as a myeloid-specific gene. The engraftment of wild-type donor-derived BMDMs in the CNS corrected numerous neuron-specific disease signatures identified in Hexb⁻/⁻ mice, including expression of genes related to cellular stress and apoptosis, accumulation of enzymatic substrate, and phenotypes associated with lysosomal dysfunction. In the context of results from other treatment modalities and previous reports regarding Hexb expression, our findings suggest that myeloid cells may be the source of functional Hexβ in the homeostatic CNS.

We next sought to determine if—and how—cultured microglia secrete Hexβ for neuronal uptake. Recent work has revealed that CNS myeloid cells can secrete lysosomal enzymes in a manner which affects

neuronal health[164,165]. In line with this, our in vitro data demonstrate that microglia also secrete enzymatically active Hexβ, building upon previous reports from cultured murine microglia and human-derived monocytes[123,124,128]. Inhibitors of lysosomal exocytosis and exosome release did not reduce in vitro Hexβ secretion, but treatment with BAPTA-AM, a calcium chelator, lowered Hexβ secretion by ~ 50%. We posited that calcium-dependent secretion of Hexβ is likely to be a result of escape from the M6P-dependent pathway, a well-described pathway in the trans-Golgi network which transports most lysosomal enzymes, including Hexβ; up to 40% of the enzyme escapes this pathway and is instead secreted into the extracellular space[12,148,149]. Secreted enzyme can then be taken up by surrounding cells via cation dependent or independent MPRs expressed on the cell surface[150,151]. The remaining secretion, which was not blocked by BAPTA, may have thus been mediated by the cation-independent form of the M6P receptor.

To further investigate the mechanisms of enzyme transfer and evaluate the M6P hypothesis, we first established that neurons are capable of taking up extracellular Hexβ and integrating it into the lysosomal compartment in vitro. We then sought to identify the mechanism(s) of neuronal uptake of Hexβ and found that IGF-II, an allosteric inhibitor of CI-MPR[152], greatly reduced the number of imaged neurons with intracellular Hexβ, implicating MPRs in neuronal uptake of Hexβ as expected. It has been previously shown that HexA, the heteromeric isoform of the β-Hexosaminidase enzyme, is taken up by cultured fibroblasts via MPRs[147], so it is highly likely that Hexβ is taken up in a similar fashion. Furthermore, using inhibitors to different components of the CME pathway, we demonstrate that neuronal uptake of Hexβ is CME-dependent, a complimentary finding given that the MPRs can facilitate clathrin-dependent receptor internalization as well as the close relationship between clathrin-coated vesicle formation and M6P receptor-mediated activity in the trans-Golgi network[166,167]. We also show that inflammation-mimicking conditions (LPS + ATP) increase Hexβ secretion, which is abolished by inhibition of the purinergic receptor P2X7, implicating this receptor in increased enzyme secretion under pathological conditions. These in vitro experiments provide insight into the potential mechanism(s) by which myeloid cell Hexβ release plays a role in neuronal function, and, by extension, how neuronal lysosomal abnormalities may be corrected following BMT + CSF1Ri in Hexb[-/-] mice. Future studies are necessary to confirm the mechanism of enzyme transfer in vivo.

Our BMT + CSF1Ri approach offers several advantages over other therapeutic interventions, including substrate reduction, enzyme replacement and gene therapies. Previous attempts at artificially rebalancing enzyme-substrate concentrations to treat disease have had mixed results. Therapies directed at reducing enzymatic substrate, though effective in other LSDs, only resulted in a partial delay of disease progression in SD model mice, and minimal human patient improvement[168–170]. Moreover, enzyme replacement therapy is limited by difficulties in accessing the CNS and a lack of feasible delivery routes[171,172]. Gene therapy, another promising avenue for the long-term treatment of SD, has also had major drawbacks in Hexb[-/-] animal models and other disease contexts. A primate study which achieved successful Hexβ reconstitution in the CNS, unfortunately, also reported heavy neurotoxicity, and gene therapy as a whole is presently limited by safety concerns, immunogenicity, and difficulty in accessing the CNS[173–175]. Our study highlights that CNS-engrafted Hexb-expressing cells have the potential to reconstitute an enzyme in a sustained, physiologically relevant manner and provide long-term reduction of substrate. We accomplish this by combining BMT with CSF1R inhibition, which has potential for immediate clinical translation. While BMT once carried significant risk, it has seen major advances in safety and efficacy over recent decades and dramatic increases in long-term survival such that it is now considered the gold standard in various conditions[12,176,177]. However, there are still challenges to utilizing BMT.

Perhaps the largest barrier in using BMT to treat CNS conditions is its limited access to the brain parenchyma and failure to correct the CNS. In the present study, by following myeloablative conditioning (irradiation) with CSF1Ri treatment and withdrawal, we are able to overcome this barrier and induce the broad influx of BM-derived cells into the CNS in a mouse model of SD. It is also worth noting for the sake of translational relevance that we designed experimental groups and group sizes in order to evaluate sex as a biological variable, and we did not observe sex differences in any of the evaluated parameters. Importantly, CSF1R inhibitors are already an approved class of drug for the treatment of Tenosynovial giant cell tumor, and recent work has established the viability of a CSF1R inhibition-based treatment paradigm for Sandhoff disease, further emphasizing the translatability of this strategy[178,179].

Though BMDMs perform many of the same immunological functions as microglia, they are not a perfect substitute. Microglia are highly specialized to the environment and demands of the CNS, and BMDMs maintain a distinct transcriptional and phenotypic identity when engrafted in the brain[180–183]. Infiltration of activated monocytes/macrophages in pathological contexts also has deleterious effects[19,184–186]. However, the consequences of having BMDMs engrafted in the CNS are secondary to the potential benefits in a context as severe as SD, especially with no available treatment and other experimental treatments limited by safety concerns. An optimal approach may involve a combination of BMT with administration of induced pluripotent stem cell (iPSC)-derived microglia, a growing field of inquiry[187–190]; however, administration of iPSC-derived microglia alone is unlikely to address the periphery, an important consideration in SD as demonstrated by our observation of correctible liver pathology. Head irradiation also comes with several drawbacks, including cognitive and synaptic deficits and microglial activation, though CSF1Ri-mediated microglial depletion has been shown to ameliorate these effects[19,191–193]. Alternative myeloablative conditioning regimes such as busulfan treatment are also compatible with CSF1Ri to induce BMDM infiltration[31–33]; however, these approaches carry their own drawbacks, especially in pediatric administration, including neurotoxicity and other neurocognitive effects[194].

Future advances in the safety and tolerability of BMT and optimization of microglial replacement will make this approach more widely applicable in patients. Our current approach involves the transplantation of recipient mice with a heterogeneous mixture of whole bone marrow donor cells, and while we hypothesize, based on prior research, that the donor-derived cells engrafted in the CNS are derived from HSCs[33], further experimentation is required to identify the specific cell population which gives rise to the therapeutic CNS-engrafted cells. We also note that our endpoint of 16 weeks of age, while demonstrating robust correction of the SD CNS at a disease-relevant time point, is not sufficient to assess the duration of correction with microglial replacement. A long-term survival study would be required to determine if the CNS remains normalized across the lifespan. For these reasons, we caution against overinterpretation of the current results prior to further optimization. Overall, however, our approach harnesses a commonly utilized clinical practice in BMT and the innate properties of myeloid cells to deliver Hexβ to correct the SD CNS, with BMDMs replacing microglia as the putative cellular source of Hexβ. Further research and refinement of this approach to mitigate the present limitations will improve its viability and enhance a strategy that could be applied in other neurodegenerative LSDs and a litany of additional CNS conditions in the future.

## Methods

All animal experiments were performed in accordance with animal protocols approved by the Institutional Animal Care and Use Committee (IACUC) at the University of California, Irvine (protocol numbers: AUP-20-144, AUP-23-086, and AUP-21-021 with MUA M2102), an

American Association for Accreditation of Laboratory Animal Care (AAALAC)-accredited institution and were conducted in compliance with all relevant ethical regulations for animal testing and research.

## Compounds
PLX5622 was provided by Plexxikon Inc. and formulated in AIN-76A standard chow by Research Diets Inc. at 1200 ppm.

## Animals
**Mice.** All mice were obtained from The Jackson Laboratory. We utilized B6;129S-*Hexb*[tm1Rlp]/J mice in this study, which harbor a loss-of-function mutation in the *Hexb* gene, described in detail previously[7] (strain #002914). Heterozygous breeding pairs were used to generate *Hexb*[-/-] mice and WT littermate controls. For BMT, bone marrow cells were isolated from sex-matched CAG-EGFP donor mice (strain #006567). Animals were housed in autoclaved individual ventilated cages (SuperMouse 750, Lab Products, Seaford, DE) containing autoclaved corncob bedding (Envigo 7092BK 1/8" Teklad, Placentia, CA) and two autoclaved 2" square cotton nestlets (Ancare, Bellmore, NY) plus a LifeSpan multi-level environmental enrichment platform. Ad libitum acces to water (acidified to pH2.5–3.0 with HCl then autoclaved) and food (LabDiet Mouse Irr 6 F; LabDiet, St. Louis, MO) was provided. Cages were changed every 2 weeks with a maximum of five animals per cage. Room temperature was maintained at $72 \pm 2\,°F$ with ambient room humidity (average 40–60% RH, range 10 –70%). Light cycle was 12 h light / 12 h dark, with lights on at 06:30 h and off at 18:30 h. Animals were assigned to treatment groups by randomization and were balanced for sex. Experimenters were blinded to genotype and treatment group during behavioral testing and analysis of histological data.

**Genotyping.** Genotyping for the *Hexb* mutation was performed using two primer sets to amplify both the wildtype (Forward 5′-ATT TTA AAA TTC AGG CCT CGA-3′ and Reverse 5′-CAT TCT GCA GCG GTG CAC GGC) and *Hexb* mutant (5′-CAT AGC GTT GGC TAC CCG TGA-3′) sequences using cycling conditions from The Jackson Laboratory and a JumpStart taq antibody (mouse stock #002914, JumpStart A7721-200TST MilliporeSigma, Burlington, MA).

## Animal treatments
**Bone marrow transplant.** 42 (21 male, 21 female in order to fully explore sex as a biological variable; 21 WT, 21 *Hexb*[-/-]) 4–6 week-old mice were irradiated with 9 Gy (XRAD 320 irradiator, Precision X-ray, North Branford, CT), anesthetized with isoflurane (5% induction and 2% maintenance isoflurane, vol/vol) and reconstituted via retroorbital injection with $2 \times 10^6$ whole bone marrow cells isolated from CAG-EGFP donor mice in $50\,\mu L$ of sterile saline solution. The irradiator was equipped with a hardening filter (0.75 mm Sn + 0.25 mm Cu + 1.5 mm Al; HVL = 3.7 mm Cu, half value layer) to eliminate low-energy X-rays. X-ray irradiation was delivered at a rate of 1.10 Gy/min. Following transplant, mice were transferred to sterile cages with autoclaved bedding and water, supplemented with DietGel® (76 A formulation, one half cup/cage at point of cage transfer, ClearH2O, INC., Westbrook, ME) and fed with Uniprim® antibiotic supplement diet (Envigo Bioproducts, Madison, WI) for 14 days to support recovery and prevent opportunistic infection.

**Microglial depletion.** Following a two-week recovery period, 20 (10 male, 10 female; 10 WT, 10 *Hexb*[-/-]) mice were fed ad libitum with PLX5622 at a dosage of 1200 ppm (to eliminate microglia) or vehicle (control) for 7 days. Mice were then returned to the vivarium diet.

## Behavioral monitoring
**Rotarod task.** Motor function was monitored on a weekly basis from 11–16 weeks of age using an accelerating Rotarod (Ugo Basile, Gemonio, Italy). Each mouse was placed on the rotarod beam while it was stationary, then acceleration was initiated. The Rotarod apparatus automatically tracked the duration between initiation of acceleration and the mouse falling to the base of the apparatus. A total of five consecutive trials were performed per mouse each week, and trial times were averaged for each mouse. If a mouse was unable to maintain position on the stationary beam and fell to the base prior to initiation of acceleration for a trial, a score of zero was manually entered.

**Weight monitoring.** Mice were weighed every other day starting at 13 weeks of age until the point of sacrifice to assess condition and progression to humane endpoint, defined as a loss of 20% of original body weight. If mice reached humane endpoint prior to 16 weeks of age, mice were sacrificed and tissue was collected as described below. Mice which did not reach 16 weeks of age were not included in the final dataset or analyses.

## Tissue preparation for histology
Mice were euthanized by $CO_2$ inhalation at 16 weeks of age or upon reaching the humane endpoint, depending on which was reached first. Mice were then transcardially perfused with 1X phosphate-buffered saline (PBS). Brain hemispheres were divided along the midline, and the left lobe of the liver was cut in half. One hemisphere of each brain and one half of each liver were fresh-frozen on dry ice and stored at $-80\,°C$, while the other hemisphere and liver half were fixed in 4% paraformaldehyde (PFA) in PBS (Thermo Fisher Scientific, Waltham, MA) for 48 hr at $4\,°C$ for immunohistochemical analysis, then cryoprotected in 30% sucrose + 0.05% sodium azide. PFA-fixed brain halves were then embedded in optimal cutting temperature media (OCT; Tissue-Tek, Sakura Fintek, Torrance, CA) and sectioned into either $10\,\mu m$ or $35\,\mu m$ sagittal slices using a cryostat (CM1950, LeicaBiosystems, Deer Park, IL). $10\,\mu m$ sections were mounted directly on slides for immunohistochemistry. $35\,\mu m$ sections were washed three times with fresh 1X phosphate-buffered saline (PBS) to remove excess OCT before transferring to a 1x PBS + 30% glycerol + 30% ethylene glycol solution for storage at $-20\,°C$. PFA-fixed liver halves were sectioned into $35\,\mu m$ slices using a Leica SM2000R freezing microtome, and sections were stored in 1x PBS + 30% glycerol + 30% ethylene glycol at $-20\,°C$. Brains and livers were protected from light to maintain GFP fluorescence.

## Flow cytometry
At the time of sacrifice, bone marrow, whole blood, and/or whole brains were harvested and analyzed by flow cytometry for hematopoietic stem cell and granulocyte chimerism and/or brain parenchymal engrafted cell profiling. Bone marrow/hematopoietic stem cells were extracted from femurs and tibia by flushing with ice-cold PBS. Whole blood/granulocytes were collected in EDTA via cardiac puncture following $CO_2$ euthanasia. Samples were centrifuged at $400 \times g$ for 5 min. Supernatant was discarded, then samples were incubated with 1 mL of 1 x ACK Lysing Buffer (A1049201, Gibco, Waltham, MA) for 1 min at RT, protected from light. Reaction was quenched with 9 mL of ice-cold PBS, and cells were again centrifuged at $400 \times g$ for 5 min. Supernatant was discarded, and pellet was resuspended in 1 mL PBS. Finally, samples were centrifuged at 800 g for 5 min, the supernatant was discarded, and the pellet was reconstituted in $225\,\mu L$ of PBS. Brains were dissociated using the Miltenyi Biotec Multi Tissue Dissociation Kit 1 (# 130-110-201, Milenyi, Auburn, CA) on a gentleMACS Octo Dissociator with Heaters (# 130-096-427, Miltenyi) following the Miltenyi protocol entitled "Dissociation of inflamed neural tissue using the Multi Tissue Dissociation Kit 1" to preserve immune cell epitopes for flow cytometric processing. Myelin and cell debris were removed using Debris Removal Solution (# 130-109-398, Miltenyi) without performing red blood cell lysis to preserve cell viability. Cells were then stained for flow cytometric analysis as previously described[19] with the following surface antibodies purchased from Biolegend (San Diego, CA) and

diluted in PBS at 1:200 unless otherwise noted: CD34 (eFlour660 (1:50, #50-0341-80, eBioscience), PE (Invitrogen, 1:100, #PIMA517831)), Sca-1-AF700 (1:100, #108141), Ter119-PE/Cy5 (#116209), ckit/CD117-PE/Cy7 (#25-1171-81, eBioscience), CD150/SLAM (PerCP-eFlour710 (#46-1502-82, eBioscience), BV605 (BD Horizon, 1:100, #BDB567309)), CD11b-APC (#101212), Gr1/Ly6C-AF700 (#108422), CD45-APC/Cy7 (1:100, #103116), NK1.1-PE (#108707), CD27-APC/Cy7 (#124226), DAPI (1:400, #422801), CD3-BV421 (1:100, #100227), CD19-BV421 (1:100, #115537), Ly6G-PE/Cy7 (1:100, #127617), CCR2-PE (1:100, #150609), CD16/32-BV605 (BD Horizon, 1:400, #BDB563006). Flow cytometry analysis was performed using a BD LSRFortessa X20 Benchtop Flow Cytometer (BD Biosciences, Franklin Lakes, NJ). Single stain controls were used to calculate compensation for spectral overlap, and fluorescence minus one controls were used to establish accurate gating. Data was analyzed in BD FACSDiva, FCS Express, and FlowJo software.software.

### GM2 Ganglioside quantification by MALDI Mass Spectrometry

Brain samples were pulverized, and the weights of allocated samples in mg were noted. Total gangliosides were purified from homogenized brain samples. Samples were dissolved in a measured volume of mass spec grade methanol according to the expected expression level of GM2 based on sample group (for e.g., the lowest GM2 sample was dissolved in 25 µL, the highest GM2 in 100 µL, etc.) Known volume of sample was spiked with a known amount of internal standard GM2-d7 (C18:0-d7 GM2 Ganglioside (GalNAcb1,4(Neu5Aca2,3)Galb1,4Glcb-Ceramide(d18:1/18:0-d7), 860121, Avanti Research, Birmingham, AL) and mixed with super-DHB matrix before spotting on the MALDI plate.

**Spotting samples.** 1.5 µL of sample was spiked with 1.5 µL of internal standard (IS conc. 1 µg/10 µL), followed by mixing with 2 µL of SDHB MALDI matrix and 1 µL of acidified water (mass spec grade water containing 0.5% of TFA). 1 µL of premixed samples were applied per spot on the MALDI plate and allowed to crystallize prior to analyzing by MALDI. The mass spec was acquired in negative reflectron mode using the Autoflex MALDI-Tof/Tof mass spectrometer (Bruker, Billerica, MA).

### Plasma lipid and neurofilament light chain measurement

Blood plasma was collected at the time of sacrifice and analyzed using the Piccolo® blood chemistry analyzer from Abaxis (Union City, CA) following manufacturer instructions. Plasma samples were thawed from −80 °C one at a time and diluted 1:1 with distilled water (ddH$_2$O), then 100 µl of diluted sample was loaded onto a Piccolo lipid plus panel plate (#07P0212, Abaxis). Various lipid parameters including total cholesterol (CHOL), high-density lipoprotein (HDL), non-HDL cholesterol (nHDLc), triglycerides, low-density lipoprotein (LDL), and very low-density lipoprotein (vLDL) were analyzed and plotted. Some parameters could not be assessed in samples with high heme content. Lipid and general chemistry controls (#07P0401, Abaxis) were utilized to verify the accuracy and reproducibility of the measurements. Quantitative biochemical analysis of plasma neurofilament light chain (NfL) was performed with the R-Plex Human Neurofilament L Assay (K1517XR-2; Meso Scale Discovery).

### Histology

**Immunohistochemistry.** Fluorescent immunohistochemical labeling followed a standard indirect technique as described previously[195]. Briefly: free-floating sections underwent a series of washes at room temperature in 1x PBS three times for 10, 5, and 5 min. Sections were then immersed in blocking serum solution (5% normal goat serum with 0.2% Triton X-100 in 1X PBS) for 1 h at room temperature, followed by overnight incubation at 4 °C in primary antibodies diluted to concentrations described below with blocking solution on a shaker. Finally, sections were incubated, covered, with fluorescent secondary antibodies at a 1:200 dilution in blocking solution at room temperature on a shaker for 1 hr, followed by 3 washes in 1X PBS prior to

mounting on microscope slides and coverslipping with Fluoromount-G with or without DAPI (0100–20 and 0100–01; SouthernBiotech, Birmingham, AL).

Brain and liver sections were stained with combinations of antibodies against ionized calcium-binding adapter molecule 1 (IBA1, 1:1000; #019–19741, Wako, Osaka, Japan), glial fibrillary acidic protein (GFAP; 1:1000; AB134436; Abcam, Cambridge, MA, United States, green fluorescent protein (GFP, 1:200; ab13970, Abcam, Waltham, MA), neuronal nuclei (NeuN, 1:1000; Ab104225; Abcam), lysosome-associated membrane protein 1 (LAMP1, 1:200; Ab25245, Abcam), and parvalbumin (Pvalb, 1:500; MAB1572, Millipore, Burlington, MA). Sections were then stained with secondary antibodies secondary antibodies Alexa Fluor 633 (A21094, Thermofisher), Alexa Fluor 555 (A21422, Thermofisher) and Alexa Fluor 488 (A11034, Thermofisher), at a 1:200 concentration. Whole brain and liver images were captured with a Zeiss Axio Scan Z1 Slidescanner using a 10 × 0.45 NA Plan-Apo objective. High-resolution fluorescent images of brain sections were captured using a Leica TCS SPE-II confocal microscope and LAS-X software. One 20X Z-stack (2 µm step interval, within a depth of 35–40 µm) field-of-view (FOV) per brain region was captured per mouse, and max projections of 63X Z-stacks were used for representative images where indicated. Liver sections were imaged using a Zeiss LSM 900 Airyscan 2 microscope and Zen image acquisition software (Zen Blue, Carl Zeiss, White Plains, NY). All images were collected using a 20x / NA 0.8 lens, and one Z-stack image (within a depth of 35–40 µm) per mouse/sex/group was acquired in each liver. Airyscan processing of all channels and z-stack images was performed in Zen software, and Bitplane Imaris Software was used for quantification of 20x confocal images.

**Periodic acid schiff.** Free-floating brain and liver sections underwent three 1x PBS washes as described above. Sections were there placed in a 0.5% periodic acid solution diluted in Millipure water and incubated for 5 min. Sections were then briefly washed in ddH$_2$O 3 times for 1.5 min each. Sections were then incubated for 15 min in Schiff's reagent (3952016, MilliporeSigma, Burlington, MA) and washed in tap water 4 times for 1 min and 15 s each and washed once briefly in ddH$_2$O. Sections were mounted and coverslipped as above. 10x brightfield images were captured with an Olympus BX60F5 microscope (Hachioji-shi, Tokyo, Japan) with an attached Nikon camera (DS-Fi3; Shinagawa-ku, Tokyo, Japan) using NIS-Elements D 5.30.05 64-bit. ImageJ analysis software was used for the quantification of brightfield images.

### Image analysis

**Imaris.** Confocal images were quantified using the spots and surfaces modules in Imaris v9.7 software. Volumetric measurements (i.e., GFP$^+$ staining volume, IBA1$^+$ microglia volume) were acquired automatically utilizing the surfaces module in confocal images of livers and cortex, corpus callosum, and cerebellum brain regions. Quantitative comparisons between experimental groups were always carried out in simultaneously stained sections. For whole-brain GFP quantification, the spots algorithm was used to automatically quantify GFP$^+$ cells in a defined region of identical size for each brain that included the midbrain and forebrain, but not the hindbrain. For microglial morphology, microglial branching and filament area were assessed using the filaments module.

**ImageJ.** Brightfield images were converted to 8-bit gray-scale and quantified using Fiji ImageJ. The threshold feature was adjusted and used to distinguish the signal from the background before the percent coverage was measured. Standardized limits for deposit size and circularity were applied to each image to further distinguish signal from background. To quantify PAS percent area coverage, the whole FOV of each 10x cortical image was analyzed, and PAS$^+$ deposits identified by thresholding were reported in pixel coverage over the total image.

## Statistics and reproducibility

Bone marrow transplant-recipient mice (WT and *Hexb*[-/-]) were generated across 11 distinct cohorts to assure reproducibility with different litters, different donors, different irradiation administrations, etc. Cell culture experiments were repeated 3-4 times with distinct samples at separate times to assure reproducibility and consistency. Statistical analysis was performed with GraphPad Prism software (v.10.0.1). To compare two groups, the unpaired Student's *t* test was used. To compare unpaired groups, a one-way ANOVA with Tukey's post hoc test was used. To compare paired groups with repeated measures for identical subjects over time (i.e., weekly Rotarod testing), a repeated measures ANOVA with Tukey's post-hoc test was used. To compare condition-paired groups, a two-way ANOVA with Tukey's post hoc test (3 groups, 2 conditions) or Sidak's test (2 groups, 2 conditions) was used. Significance of single outliers was calculated using the Grubbs' extreme studentized deviate method, and significant ($p < 0.05$) outliers were excluded. For all analyses, statistical significance was defined by a *p*-value below 0.05. All bar graphs are represented as group mean ± SEM, significance is expressed as follows: *$p < 0.05$, **$p < 0.01$, ***$p < 0.001$, ****$p < 0.0001$. n represents the number of mice within each group.

## Spatial transcriptomic & proteomic analysis

**Section preparation.** One day prior to the experiment, PFA-fixed brain hemispheres were embedded in optimal cutting temperature (OCT) compound (Tissue-Tek, Sakura Fintek, Torrance, CA), and 10 μm sagittal sections were cut using a cryostat (CM1950, LeicaBiosystems, Deer Park, IL). Six hemibrains were mounted onto VWR Superfrost Plus slides (Avantor, 48311−703) and kept at −80 °C overnight. For *Hexb*[-/-] BMT groups and the WT control group, *n* = 3 mice per experimental condition were utilized (wild-type control, *Hexb*[-/-] control, *Hexb*[-/-] BMT, *Hexb*[-/-] BMT + CSF1Ri) for transcriptomics and proteomics. When selecting representative brains, we considered BMDM infiltration levels from both *Hexb*[-/-] BMT groups, choosing brains with similar total forebrain GFP+ staining to group averages. Tissue was processed in accordance with the Nanostring CosMx fresh-frozen slide preparation manual for RNA and protein assays (NanoString University).

**Slide treatment, RNA, day 1.** Slides were removed from −80 °C and baked at 60 °C for 30 min. Slides were then processed for CosMx: three 1X washes PBS for 5 min each, 4% sodium dodecyl sulfate (SDS; CAT#AM9822) for 2 min, three 1X PBS washes for 5 min each, 50% ethanol for 5 min, 70% ethanol for 5 min, and two washes with 100% ethanol for 5 min each before allowing slides to air dry for 10 min at room temperature. Antigen retrieval was performed using a pressure cooker maintained at 100 °C for 15 min in preheated 1X CosMx Target Retrieval Solution (Nanostring, Seattle, WA). Slides were then transferred to DEPC-treated water (CAT#AM9922) and washed for 15 s, incubated in 100% ethanol for 3 min, and air dried at room temperature for 30 min. Slides were incubated with digestion buffer (3 μg/mL Proteinase K in 1X PBS; Nanostring) for tissue permeabilization, then washed 2 times in 1X PBS for 5 min. Fiducials for imaging were diluted to 0.00015% in 2X SSC-T and incubated on slides for 5 min. Following fiducial treatment, slides were protected from light at all times. Tissues were then post-fixed with 10% neutral buffered formalin (NBF; CAT#15740) for 1 min, washed twice with NBF Stop Buffer (0.1 M Tris-Glycine Buffer, CAT#15740) for 5 min each, and washed with 1x PBS for 5 min. Next, NHS-Acetate (100 mM; CAT#26777) mixture was applied to each slide and incubated at room temperature for 15 min. Slides were washed twice with 2X SSC for 5 min each. Slides were then incubated for 16−18 h in a hybridization oven at 37 °C with a modified 1000-plex Mouse Neuroscience RNA panel (Nanostring) for in situ hybridization with the addition of an rRNA segmentation marker.

**Slide treatment, RNA, day 2.** Following in situ hybridization, slides were washed twice in pre-heated stringent wash solution (50% deionized formamide [CAT#AM9342], 2X saline-sodium citrate [SSC; CAT#AM9763]) at 37 °C for 25 min each, then washed twice in 2X SSC for 2 min each. Slides were then incubated with DAPI nuclear stain for 15 min, washed with 1X PBS for 5 min, incubated with GFAP and histone cell segmentation markers for 1 h, and washed three times in 1X PBS for 5 min each. Flow cells were adhered to each slide to create a fluidic chamber for spatial imaging. Slides were loaded into and processed automatically with the CosMx instrument. Approximately 300 fields of view (FOVs) were selected on each slide, capturing hippocampal, corpus callosum, upper thalamic, upper caudate, and cortical regions for each section. Slides were imaged for approximately 7 days, and data were automatically uploaded to the Nanostring AtoMx online platform. Pipeline pre-processed data was exported as a Seurat object for analysis with R 4.3.1 software.

**Side treatment, protein, day 1.** Slides were removed from −80 °C and baked at 60 °C for 30 min, then washed three times with 1X Tris Buffered Saline with Tween (TBS-T; CAT#J77500.K2) for 5 min each. Antigen retrieval was performed using a pressure cooker held at 80 °C in pre-heated Tris-EDTA buffer (10 mM Tris Base [CAT#10708976001], 1 mM EDTA solution, 0.05% Tween 20, pH 9.0) for 7 min. Following antigen retrieval, slides were allowed to cool to room temperature for 5 min, then washed three times in 1X TBS-T for 5 min each. Slides were incubated with Buffer W (Nanostring) for 1 h at room temperature. Slides were then incubated for 16−18 h at 4 °C with the CosMx 64-plex protein panel and segmentation markers (GFAP, IBA1, NEUN, and S6).

**Side treatment, protein, day 2.** Following incubation, slides were washed three times with 1X TBS-T for 10 min each, then washed with 1X PBS for 2 min. Fiducials for imaging were diluted to 0.00005% in 1X TBS-T and incubated on the slide for 5 min. Slides were then washed with 1X PBS for 5 min, incubated in 4% PFA for 15 min, and washed three times with 1X PBS for 5 min each. Slides were incubated with DAPI nuclear stain for 10 min, then washed twice with 1X PBS for 5 min each. Slides were then incubated with 100 mM NHS-Acetate for 15 min and washed with 1X PBS for 5 min. Flow cells were adhered to each slide to create a fluidic chamber for spatial imaging. Slides were loaded into and processed automatically with the CosMx instrument. Approximately 600 FOVs were selected per slide, capturing each full section. Slides were imaged for ~6 days data were automatically uploaded to the Nanostring AtoMx online platform. Pipeline pre-processed data was exported as a Seurat object for analysis with R 4.3.1 software.

**Spatial transcriptomics data analysis.** Spatial transcriptomics datasets were processed as previously described[196] with Seurat 5.0.1, as follows: principal component analysis (PCA) and uniform manifold approximation and projection (UMAP) analysis were performed to reduce the dimensionality of the dataset and visualize clusters in space. Unsupervised clustering at 1.0 resolution yielded 39 clusters for the WT control versus *Hexb*[-/-] control dataset and 38 clusters for the dataset, which included WT controls, *Hexb*[-/-] controls, *Hexb*[-/-] BMT, and *Hexb*[-/-] BMT + CSF1Ri. Clusters were annotated with a combination of automated and manual approaches: (1) label annotations from the Allen Brain Atlas single-cell RNA-seq reference dataset (for cortex and hippocampus) were projected onto our spatial transcriptomics dataset[36], and (2) cluster identities were further refined via manual annotation based on gene expression of known marker genes and location in XY space. Cell proportion plots were generated by first plotting the number of cells in each broad cell type, then scaling to 1. normalized percentages for each group, calculated by dividing the number of cells in a given cell type-group pair by the total number of cells in that group, and 2. dividing by the sum of the proportions across

the cell type to account for differences in sample sizes. Differential gene expression analysis per cell type between groups was performed on scaled expression data using MAST to calculate the average difference[197], defined as the difference in log-scaled average expression between the two groups for each broad cell type. DEG scores were calculated between group pairs for each subcluster by summing the absolute $log_2$ fold change values of all genes with statistically significant gene (i.e., $p_{adj} < 0.05$) differential expression patterns between two groups. Data visualizations were generated using ggplot2 3.4.4[198].

### Biochemical analysis of salt-soluble and detergent soluble protein fractions

**Fractionation.** Protein fractions were obtained from fresh frozen hemispheres, which were first pulverized using a Bessman Tissue Pulverizer. Samples were then homogenized in 10 µL high salt reassembly buffer per 1 mg of sample (RAB buffer; C752K77; Thermo Fisher; 100 mM MES, 1 mM EGTA, 0.5 mM MgSO4, 750 mM NaCl, 20 mM NaF, 1 mM Na3VO4, pH = 7.0). Homogenates were centrifuged at $44,000 \times g$ for 20 min, and the supernatant was collected as the salt-soluble fraction. The pellet was then resuspended in 10 µL of detergent-containing Tissue Protein Extraction Reagent (T-PER 25 mM bicine and 150 mM sodium chloride (pH 7.6); Life Technologies, Grand Island, NY) per 1 mg of original sample to gently extract total protein, then centrifuged at $44,000 \times g$ for 1 h. Supernatant was then collected as the detergent-soluble fraction. Fractions were analyzed using the Hexβ activity assay as described below, and values were normalized to protein concentration for each sample.

**Protein quantification.** Total protein in salt-soluble and detergent-soluble fractions was quantified using the Pierce™ 660 nm Protein Assay Kit (#22662, Thermo Fisher, Waltham, MA). BSA standards were created for RAB and T-PER-extracted samples by diluting kit-supplied protein assay standards in extraction media 1:1. Fractions were removed from − 80 °C and thawed on ice. A 1:5 dilution of each sample was created by dilution in extraction media. Diluted BSA standards and 10 µL of diluted samples were loaded onto 96-well plates in triplicate, then 150 µL of Pierce Reagent was added to each well with a multi-channel pipette using reverse pipetting. Plates were agitated on a plate shaker for 1 min, then incubated at room temperature for 5 min before absorbances were read on a 96-well colorimetric and fluorescent microplate reader. Average absorbances were calculated for each sample, and protein concentration was determined using the standard curve of each plate.

### Primary glial cultures from neonatal mice

Primary mixed microglia-astrocyte cultures were generated as previously described[199]. Whole brains were extracted from neonatal 3- to 5-day-old mice, and cortical tissue was cut into small pieces before digestion with trypsin. Trypsin was quenched using glia media (DMEM supplemented with 10% performance plus heat-inactivated serum (#10082147; Thermo Fisher) and 1% penicillin/streptomycin (P4333-100 ML; Millipore Sigma) and tissue was dissociated by pipetting up and down 20 times with a 1000 µL tip a total of 6 times. Following dissociation, the tissue was centrifuged at $150 \times g$ for 7 min with slow start-stop at room temperature. Cells were resuspended in fresh glia media and filtered using 100 µm cell strainers (#352360; Falcon, Abilene, TX), followed by 40 µm strainers (#352340; Falcon). Finally, the cells were reconstituted with 10 mL of glia media and placed in T-75 cm2 flasks. Flasks were precoated with 0.002% poly-lysine (P4707-50 ML; Sigma-Aldrich) for at least 30 min at room temperature. After 24 h, debris was removed by gentle tapping of flasks and removal of media, and 20 mL of fresh media was added to the cell cultures. After 14 days in vitro, mixed microglia-astrocyte cultures were used for experiments. Primary microglia were removed from mixed microglia-astrocyte culture by gentle shaking as described previously[184].

Microglia were seeded at 20,000/150 µl on pre-coated 0.002% poly-lysine 96-well plates in 1:2 (conditioned: fresh) media and left to adhere for 48 h.

### β-hexosaminidase activity assay

The Hexβ activity assay was run as previously described[126]: cells were washed with 1x PBS followed by 150 µl of fresh media and treated with DMSO (vehicle), 400 nM vacuolin-1 (673000, Millipore Sigma), 10 µM BAPTA-AM (A1076, Millipore Sigma) and 20 µM GW4869 (D1692, Millipore Sigma) for 6 h. For P2X7 inhibition experiments, cells were treated with 1 µg/ml LPS (L4130, Millipore Sigma) for 3 h, followed by 10 µM of P2X7 inhibitor, A-804598 (A16066, Tocris) for 10 min and 1 mM ATP (A0157, TCI) for 20 min. The media (supernatant) was collected, and cells were lysed in 150 µl of 1 x M-PER supplemented with protease and phosphatase inhibitors for 20 min, on ice. Following lysis, samples were centrifuged at $15,000 \times g$ for 10 min. The β-hexosaminidase assay was performed in a 96-well plate by mixing 50 µL of 2 mg/mL 4-nitrophenyl N-acetyl-β-D-galactosaminide (N9003, Millipore Sigma) in 0.1 M citrate buffer (pH 4.5) with 75 µL of supernatant or cell lysate and incubating for 1 h at 37 °C. Following 1 h incubation, 100 µl of 0.2 M borate buffer (pH 9.8) was added to stop the reaction. The plate was read at 405 nm using an absorbance plate reader. Percentage values were obtained by dividing the reading from the supernatant with that of the cell lysate.

### E18 hippocampal neuron cultures and imaging

Dissociated E18 hippocampal neurons were purchased from Brain Bits by Transnetyx (C57EDHP). 50,000 cells were plated on (50 µg/ml) poly-D-lysine (A3890401, Thermofisher) pre-coated glass bottom plates (P35G-1.5-14-C, Mattek) in NbActiv1 media (Neurobasal/B27/Glutamax). Half media swaps were completed every 3-4 days with NbActiv1 media without Glutamate supplemented. Neurons were left for a minimum of 6 days before experiments were conducted. Neurons were pretreated with the following inhibitors for 1 h before addition of Hexβ: 100 nM cytochalasin D (CytD, C8273, Sigma), 10 µM chlorpromazine (CPZ, HY-B0407A, MedChemExpress), 80 µM Dynasore (HY-15304, MedChemExpress) and 25 ng/ml mouse insulin growth factor II (IGFII, 792MG050, R&D Systems). 10 µg of recombinant C terminal 6x his-tagged mouse Hexβ was added to neurons for 24 h. Following incubation, cells were washed 3 × 5 min each with 1X PBS. Neurons were fixed with 4% PFA for 15 min at room temperature. Following fixation, cells were washed a further 3 × 5 min each before adding blocking buffer (5% normal goat serum with 0.02% triton-x 100 in 1X PBS) for 1 h, at room temperature with gentle shaking. Cells were then incubated with primary antibodies overnight at 4 C. Primary antibodies used: Rat anti-mouse LAMP1 (1:250; Ab25245, ABCAM), rabbit anti-6x his-tag (1:500; MA5-33032, Invitrogen) and mouse anti-mouse NeuN (1:500; MA5, 33103, Invitrogen). Following incubation, cells were washed 3 times for 5 min each with 1X PBS and incubated with secondary antibodies Alexa Fluor 633 (A21094, Thermofisher), Alexa Fluor 555 (A21422, Thermofisher) and Alexa Fluor 488 (A11034, Thermofisher), at a 1:250 concentration for 1 h at room temperature with gentle shaking. Finally, cells were washed 3 × 5 min each in 1X PBS and imaged using LSM 900 (Carl Zeiss) 10 × 0.45 NA air objective and 4 × digital zoom. 10–15 neurons were imaged per treatment condition.

### Reporting summary

Further information on research design is available in the Nature Portfolio Reporting Summary linked to this article.

## Data availability

The gene and protein expression data annotated for cell types generated in this study by CosMx spatial molecular imaging have been deposited as .rds files at https://doi.org/10.5061/dryad.3tx95x6rq. One spatial transcriptomics dataset is from only wildtype control

and Hexb knockout control mice (HexbKO_wt_vs_con_with_allenref_labels_public.rds). One dataset is a spatial transcriptomics experiment performed on wildtype control, Hexb knockout control, Hexb knockout treated with bone marrow transplant (BMT), and Hexb knockout treated with BMT and colony-stimulating factor 1 receptor (CSF1R) inhibition, which results in broad replacement of microglia with bone marrow-derived macrophages/monocytes (HexbKO_full_with_allenref_labels_public.rds). The final dataset is a spatial proteomics experiment performed on these same four groups (WT control, Hexb knockout control, Hexb knockout BMT, and Hexb knockout BMT + CSF1R inhibition) (HexbKO_protein_seurat_w_group_assignments_public.rds). Source data are provided in this paper.

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

## Acknowledgements

This work was supported by the National Institutes of Health (NIH) under awards: R01NS083801 (NINDS), R01AG081599 (NIA), RF1AG056768 (NIA), and U54AG054349 (NIA Model Organism Development and Evaluation for Late-onset Alzheimer's Disease [MODEL-AD]) to K.N.G., T32NS082174 (NINDS) to K.I.T., and T32NS121727 (NINDS) to N.K. We thank Brian L. West and Andrey Rymar at Plexxikon, Inc. for providing and formulating CSF1Ri chow. We thank Vanessa Scarfone and Pauline Nguyen of the Flow Cytometry Core in the Sue & Bill Gross Stem Cell Research Center for their critical support in flow cytometry experiments and analysis. Images of mice in Figs. 1 and 2 and Supplementary Figs. 6 and 7 adapted from Servier Medical Art (https://smart.servier.com/), licensed under CC BY 4.0 (https://creativecommons.org/licenses/by/4.0/).

## Author contributions

K.I.T.: Conceptualization, data curation, formal analysis, investigation, methodology, project administration, visualization, writing – original draft. C.A.B.: Formal analysis, investigation, methodology. N.K.: Data curation, methodology, resources. Z.R.S.: Formal analysis, investigation. K.J.G.D.: Formal analysis, investigation. G.O.M.: Formal analysis, investigation. B.P.C.: Methodology, formal analysis, investigation. M.P.: Methodology, formal analysis, investigation. C.A.P.: Investigation. E.Z.T.: Formal analysis, investigation. R.P.K.: Methodology, resources. S.M.E.: Methodology, resources. V.S.: Methodology, supervision, resources. M.M.A.: Methodology, supervision, resources. L.A.H.: Conceptualization, methodology, project administration, supervision, writing – original draft. K.N.G.: Conceptualization, funding acquisition, methodology, project administration, resources, supervision, writing – original draft.

## Competing interests

K.N.G. is on the scientific advisory board of Ashvattha Therapeutics, Inc. All other authors declare no conflict of interest.
