## [Transparent Peer Review file · Nature Communications]

Microglial replacement in a Sandhoff disease mouse model reveals myeloid-derived β -hexosaminidase is necessary for neuronal health

Corresponding Author: Professor Kim Green

Version 0:

Reviewer comments:

Reviewer #1

(Remarks to the Author)

This study investigates Sandhoff disease, a neurodegenerative disorder caused by a deficiency in the β subunit of β -hexosaminidase, leading to GM2 ganglioside accumulation in lysosomes. Spatial transcriptomics and proteomics were used to characterize the molecular and cellular features of the disorder. To assess the therapeutic potential of microglia depletion, Sandhoff disease mice received a combination of bone marrow transplant and colony-stimulating factor 1 receptor inhibition to replace Hexb-deficient microglia with functional myeloid cells. This treatment reversed apoptotic gene expression, improved behavioral outcomes, restored enzymatic function, reduced substrate accumulation, and normalized neuronal lysosome morphology.

1. The authors claim that Hexb is "highly specific or exclusive to microglia" and that myeloid-derived Hexb is necessary for "normal neuronal function." They also suggest that "myeloid cells are likely the source of functional Hex β in the homeostatic CNS, necessary for neuronal health and lysosomal function." While I agree that Hexb expression is very high in microglia, to the extent that it is used as a microglial marker, I question the assertion that its expression is "exclusive" to microglia and that microglial Hexb is solely responsible for normal neuronal function. Neurons must express at least low levels of Hexb to prevent GM2 ganglioside buildup, as shown by the absence of GM2 accumulation in cultured neurons. In the context of Sandhoff disease, it is plausible that Hexb-sufficient microglia or macrophages could supply the enzyme to deficient neurons to prevent GM2 ganglioside buildup. Clarification on these points would be helpful. Ultimately, definitive evidence that microglia are the sole source of Hexb in the CNS would require a conditional knockout of Hexb specifically in microglia.

2. A key conclusion of the manuscript is that the combined bone marrow transplant (BMT) and CSF1 receptor inhibition (CSF1Ri) approach is significantly more effective than BMT alone and may enable "long-term substrate reduction." Previous studies on BMT in the Sandhoff disease model have shown a lifespan of eight months, doubling their normal lifespan. However, the current study includes a lifespan analysis only up to four months. Extending this experiment with humane endpoints would provide stronger evidence on whether the combined treatment offers a more substantial improvement than BMT alone.

3. GM2 ganglioside accumulation in neurons is the biochemical hallmark of Sandhoff disease. If Hexb enzyme is transferred from wild-type microglia to enzyme-deficient neurons, it would be expected to reduce GM2 ganglioside burden. The authors used PAS staining to show a reduction in lysosomal storage, but PAS staining is not specific to GM2 ganglioside and instead reacts with various carbohydrate substrates. Demonstrating a significant reduction of CNS GM2 levels in BMT + CSF1Ri mice compared to BMT-only mice would provide strong evidence for the central hypothesis of the study: that enzyme supplied by microglia or macrophages is transferred to neurons and reduces GM2 ganglioside storage. in Sandhoff disease.

Reviewer #2

(Remarks to the Author)

The current manuscript by Tsourmas and colleagues is conceptionally interesting, describing the effect of a complete microglial exchange/replacement in the mouse model of Hexb-deficiency that mimics human Sandhoff disease to some extent.

Despite the fact that the idea of microglial replacement by bone marrow reconstitution is not new at all and the combination with prior microglia depletion with as also already introduced by the Bo Peng and other groups before (Zhang et al. *Neurosci Bull* 2023, Xu Z et al. *STAR Protoc* 2021, Xu Z et al. *Cell Rep* 2020) some aspects of this study are interesting.

The novelty is based on the fact that the authors are able to rescue the lethal phenotype associated with Hexb-deficiency in mice. However, despite being interesting, the mechanisms are largely unclear and further extensive experiments are clearly needed to substantiate their findings.

In general, the manuscript is quite overloaded, very dense and difficult to read with already 175 references. I would therefore strongly suggest to improve reading and understanding by focusing on the main points of the paper. In detail, authors should put emphasis on their findings in figures 2-6 highlighting bone marrow chimeras and their phenotypes and add more supportive experimental data. In contrast, figure 1 is interesting but mainly shows that a) Hexb is microglia-specific (that was already nicely shown by Hickman et al. *Nat Neurosci* 2013) and b) that Hexb microglia show a "DAM-like" signature that is somehow expected. Similarly, figure 7 does not add so much to the microglia replacement findings and could be omitted. One main weakness of the study is the experimental design. The authors used whole bone marrow that is a highly complex mixture of several myeloid cell types (HSC, MDSC, monocytes, CMP etc.) that potentially all can enter the CNS and prevent disease. For the translation value of this study it is highly desirable to identify by adoptive transfer study the nature of the myeloid subset that improves disease.

Another weakness is the lack of mechanistic explanation how Hexb-deficiency leads to microglial activation. This main point was not addressed in this manuscript at all. Fig.7 falls short in providing some mechanisms in this direction.

In sum, the authors should address the main points below to strengthen the validity of their findings and to improve the focus of the manuscript.

Major points:

1. Figure 1: This figure is difficult to digest and does not help the overall story, because a) all CNS cells are explored by biased (!) spatial transcriptomics approach leading to the description of endless neuronal subsets and leaving only very few left CNS myeloid cells, b) the nature of the described myeloid cells remains totally unclear (DCs or monocytes or microglia etc.), c) no protein validation was performed and finally d) Hexb specificity was already shown by others before (Hickman et al. *Nat Neurosci* 2013). In sum, I would suggest to remove the figure to the suppl. section or to omit it. Further, analyses on 3 mice each is not sufficient. More mice are definitely needed.
2. As already mentioned above the Hexb^{-/-} disease rescue by wt bone marrow transfer is exciting but the nature of the disease modulating myeloid cell remains completely unclear. Due to the highly complex mixture of the bone marrow compartment several myeloid cells can be the candidates. Thus, the cell of action needs to be identified (HSC versus MDSC versus monocytes versus CMP etc.).
3. Figure 7: This figure starts to explain the crosstalk between Hexb, microglia and neurons. It is somehow a start for a better understanding underlying pathophysiology but fails to explain WHY loss of a microglial enzyme leads to massive neurodegeneration seen in Hexb mice. I would therefore suggest to remove this immature part from the manuscript.

Minor points:

1. The generation of bone marrow chimeras for the delivery of GFP bone marrow was already described by the Priller group in 2021 *Nat Med* and therefore this pioneering study should be cited.
2. Line 81: The term "neuroinflammation" is wrongly used because only conditions with infiltration of hematogenic cells such as lymphocytes, neutrophils etc. can be considered as true neuroinflammation (see also the white consensus paper on this topic Paolicelli et al. *Neuron* 2022).

Reviewer #3

(Remarks to the Author)

This manuscript presents a comprehensive study on the potential of microglial replacement therapy for treating Sandhoff disease (SD) using a combined bone marrow transplant (BMT) and colony stimulating factor 1 receptor inhibition (CSF1Ri) approach in a mouse model. The authors demonstrate that this treatment strategy leads to significant improvements in disease pathology, including reversal of gene expression changes, clearance of substrate accumulation, and normalization of lysosomal phenotypes in neurons.

Below are some major and minor comments:

1) Sample Size for spatial analysis (Figure 1a as an example): The sample size of three mice per group is notably small for a mouse study. While acknowledging the challenges of in vivo experiments, this sample size is insufficient to provide robust and generalizable results. Current standards typically require larger groups, even for human studies, to ensure statistical validity and reproducibility. Consider increasing the number of mice per group to strengthen the study's conclusions.

2) Deconvolution of spatial transcriptomics data to single-cell resolution is particularly challenging, especially in disease settings where microglial populations can fluctuate significantly. Given that reference maps like the Allen Brain Atlas are based on healthy samples, it is unclear how single-cell resolution was confidently achieved in this study. Please provide a detailed explanation of the methods and validations used to ensure accuracy in applying the Allen map to a diseased context.

- 3) Long-term effects: The study focuses on a specific time point (16 weeks). It would be valuable to assess the long-term effects of the treatment, including survival data and whether the improvements are sustained over time.
- 4) Dose-response relationship: Investigate whether different doses of CSF1Ri or varying durations of treatment affect the efficacy of the therapy.
- 5) Behavioral assessments: While the authors present data on motor function using the Rotarod test, including additional behavioral tests to assess cognitive function and other neurological symptoms would provide a more comprehensive view of the treatment's effects.
- 6) Mechanistic Clarity on Hexb Secretion: While the study shows Hexb secretion from microglia and uptake by neurons, the exact mechanisms of lysosomal trafficking and uptake remain speculative. For example, how Hexb integrates into neuronal lysosomes needs further exploration, particularly under pathological conditions.
- 7) Interpretation of Inflammatory Role: While inflammation is not the main focus, the study hints at its interplay with Hexb secretion. However, the manuscript could benefit from discussing whether inflammatory conditions alter Hexb function or neuronal lysosomal health beyond secretion dynamics.
- 8) Sex differences: Is there a sex difference? Analyze whether there are any sex-specific differences in treatment efficacy or disease progression. Or make a statement in that regard.
- 9) Controls and Comparisons: The manuscript does not explicitly address whether the replacement efficacy of BMT + CSF1Ri differs by region. It would strengthen the conclusions if regional efficacy were correlated with therapeutic outcomes (e.g., motor performance vs. corpus callosum infiltration)

Minor Concerns

Figures demonstrating spatial transcriptomics and proteomics results are not included in the snippets. Ensure that the visualizations effectively highlight cell-specific and region-specific changes and that axis labels and legends are clear.

Provide additional quantitative comparisons for the therapeutic effects of BMT vs. BMT + CSF1Ri across different brain regions.

Include a more in-depth discussion on how this work bridges gaps in SD pathology, specifically the non-cell-autonomous roles of microglia.

Expand on the molecular pathways of Hexb secretion and neuronal uptake, particularly the role of P2X7 signaling under inflammatory conditions.

Version 1:

Reviewer comments:

Reviewer #1

(Remarks to the Author)

The authors have responded satisfactorily to my comments. The manuscript presents novel findings that support a significant role for microglia in Sandhoff disease, a disease in which neuronal cell death dominates.

However, I would like to note that, contrary to the authors' statement "neurons within the Hexb^{-/-} mouse brain also lack GM2 accumulation until several months of age; substrate buildup requires sustained deprivation of Hexβ protein in vivo.", GM2 ganglioside can, in fact, be detected in the fetal brain of Sandhoff mice, (<https://doi.org/10.1016/j.neures.2019.07.004>).

Reviewer #2

(Remarks to the Author)

As in the first round of reviews this referee acknowledges especially the novelty of this story, namely that bone marrow transfer in microglia-depleted Hexb-deficient recipients is able to stop clinical symptoms. Despite some general improvements of the study, two major but essential points from my first round of reviewing still remain open. I will describe below in depth why it is so important to clarify them before publication of this interesting topic.

Major points:

1. One still remaining weakness of the study is the usage of whole bone marrow to rescue Hexb-deficient brains. As I wrote before, the bone marrow contains a highly heterogeneous mixture of myeloid cells (monocytes, HSC, MDP, CMP etc.) that ALL have the potential to replace microglia cells in the brain. Even short-living monocytes have just very recently shown by the S.Jung group to be potential microglia precursors for aged microglia (PMID: 40279248). Why is important to identify by short-term adoptive transfer experiments (that will last max. three months) the precise nature

of the bone marrow myeloid subset that improves disease? Here is the reason: A similar bone marrow rescue has been shown in *Mecp2*-deficient mice with the overall claim that bone marrow-derived phagocytes could rescue the microglia phenotype in *Mecp2* mice (Derecki NC et al. *Nature* 2012, PMID: 22425995). Based on this report that included an insufficient characterization of the transferred bone marrow subsets, clinical trials of bone marrow transfer for patients with Rett-Syndrome (*Mecp2* deficiency) have been initiated (MT2013-31:Allogeneic Hematopoietic Cell Transplantation for Inherited Metabolic Disorders, Sever Osteoporosis and Males with Rett Syndrome Following Conditioning with Busulfan (Therapeutic Drug Monitoring), Fludarabine +/- ATG. University of Minnesota; Minneapolis, MN USA) with unsuccessful results. Only later, the primary data from the initial *Nature* 2012 paper were shown to be non-reproducible (Wang J et al. *Nature* 2015, PMID: 25993969). Taken together, the easy-to-do adoptive transfer experiments are simply essential for translational reasons. Otherwise, there is a significant risk of mis- or overinterpretations of the data that could be fatal for future patient's curative approaches.

The newly added flow data on Extended Data Figure 6 are not sufficient because they didn't used individually sorted and transferred cells. I simply do not agree with their given arguments.

2. Along this line, this study still lacks on a too early survival time point after bone marrow transfer (16 weeks of age). It is also essential to look for long-term survival including clinical tests, neuropathology, GM2 accumulations, engraftment of donor-derived cells in the CNS and rescue of neuronal numbers as a minimum. Here, I would ask for the 28 weeks' time point (7 months) for a robust experimental group of mice. Again, these later time points are key for future translation aspects in Sandhoff patients. I don't agree that this would induce "major ethical concerns" when control ko animals will not be treated.

3. A clear mechanistic proof of microglia-derived *Hexb* delivery into the lysosomes of *Hexb*-deficient neurons in the putative rescued bone marrow chimeras is still lacking. This should be done in situ, namely in the brain sections of recipient mice.

Version 2:

Reviewer comments:

Reviewer #2

(Remarks to the Author)

This referee still doesn't not agree on all the answers given by the authors to many of the valid points. Especially the tone of replies continues to be not always adequate. However, this referee appreciates the overall changes made during the several rounds of revisions and acknowledges the novelty of this study.

There is only one minor point missing: there was recently a similar study showing comparable effects: PMID: 40305572. I don't think that this study prevents the publication of the current study at *Nat Comm* but should definitely be cited by the authors.

We would like to first thank the reviewers for their time and thoughtful consideration of the manuscript. Their suggestions are insightful and have resulted in major improvements in our study. In response to the concerns raised by the reviewers, we have performed several additional experiments, including:

- GM2 ganglioside glycolipid quantification via MALDI mass spectrometry.
- Identification of the engrafted cell type(s) that are therapeutically relevant in our paradigm using bone marrow transplant experiments, immunohistochemistry, and extensive flow cytometry assessment using two specialized panels on a newly generated cohort of chimeric mice treated with CSF1R inhibitors.
- An additional CosMx spatial transcriptomic experiment, which includes *Gfp* and allows for the identification of engrafted bone marrow-derived cells, to further parse the cellular identity and gene expression profiles of engrafted bone marrow-derived cells.
- Further *in vitro* experimentation to identify the mechanism by which neurons take up extracellular Hex β protein.

We have also clarified several points and made requested adjustments to the manuscript text in response to the reviewers' comments. All changes to the manuscript have been tracked in blue text.

Sincerely,

Kim Green

Reviewer #1 (Remarks to the Author):

This study investigates Sandhoff disease, a neurodegenerative disorder caused by a deficiency in the β subunit of β -hexosaminidase, leading to GM2 ganglioside accumulation in lysosomes. Spatial transcriptomics and proteomics were used to characterize the molecular and cellular features of the disorder. To assess the therapeutic potential of microglia depletion, Sandhoff disease mice received a combination of bone marrow transplant and colony-stimulating factor 1 receptor inhibition to replace Hexb-deficient microglia with functional myeloid cells. This treatment reversed apoptotic gene expression, improved behavioral outcomes, restored enzymatic function, reduced substrate accumulation, and normalized neuronal lysosome morphology.

1. The authors claim that Hexb is "highly specific or exclusive to microglia" and that myeloid-derived Hexb is necessary for "normal neuronal function." They also suggest that "myeloid cells are likely the source of functional Hex β in the homeostatic CNS, necessary for neuronal health and lysosomal function." While I agree that Hexb expression is very high in microglia, to the extent that it is used as a microglial marker, I question the assertion that its expression is "exclusive" to microglia and that microglial Hexb is solely responsible for normal neuronal function. Neurons must express at least low levels of Hexb to prevent GM2 ganglioside buildup, as shown by the absence of GM2 accumulation in cultured neurons. In the context of Sandhoff disease, it is plausible that Hexb-sufficient microglia or macrophages could supply the enzyme to deficient neurons to prevent GM2 ganglioside buildup. Clarification on these points would be helpful. Ultimately, definitive evidence that microglia are the sole source of Hexb in the CNS would require a conditional knockout of Hexb specifically in microglia.

We thank the reviewer for this comment. We agree that the expression of *Hexb* "exclusively" in microglia and how it potentially serves as a source of Hex β for normal CNS function is an important topic. We agree that a model that allows for cell type-specific knockout of *Hexb* is needed to fully address this concern. Unfortunately, a *Hexb* floxed mouse is not currently available—we are in the process of procuring one, but it will require validation and considerable time to evaluate (more time than permitted for this resubmission). It should be noted that previous studies provide strong evidence of the microglial specificity of *Hexb* (Hickman et al., 2013, Masuda et al., 2020; see Reviewer 2's remarks to the authors about the previous establishment of exclusive *Hexb* expression in microglia). In agreement with this, we show for the first time using spatial transcriptomics

that *Hexb* transcripts are only detected within myeloid cells (see Figure 1i). Moreover, our data shows that myeloid-derived Hex β is sufficient to alleviate neuronal phenotypes in Sandhoff disease (SD), as the source for *Hexb*/Hex β in our model is from transplanted myeloid cells. However, we acknowledge the lack of certainty in the case of neuronal production of functional enzyme under homeostatic conditions and have therefore tempered our language to address the reviewer's concern (Lines 92, 115, 738-739, 795). Regarding the reviewer's comment about cultured neurons lacking GM2 accumulation, we would like to point out that neurons within the *Hexb*^{-/-} mouse brain also lack GM2 accumulation until several months of age; substrate buildup requires sustained deprivation of Hex β protein *in vivo*. We would also like to note to this reviewer that, as a pilot experiment, we cultured primary cortical neurons from neonatal *Hexb*^{-/-} mice and did not observe abnormalities during that period.

2. A key conclusion of the manuscript is that the combined bone marrow transplant (BMT) and CSF1 receptor inhibition (CSF1Ri) approach is significantly more effective than BMT alone and may enable "long-term substrate reduction." Previous studies on BMT in the Sandhoff disease model have shown a lifespan of eight months, doubling their normal lifespan. However, the current study includes a lifespan analysis only up to four months. Extending this experiment with humane endpoints would provide stronger evidence on whether the combined treatment offers a more substantial improvement than BMT alone.

We thank the reviewer for this comment and agree that extending this experiment would provide strong evidence of long-term substrate reduction in BMT + CSF1Ri treated mice. However, the selected timepoint (16 weeks of age) was chosen to allow for direct age-matched comparison between treatment groups and the untreated control *Hexb*^{-/-} animals. It should be noted that while previous publications reported the maximum lifespan of *Hexb*^{-/-} mice to be around 19 weeks of age, we and others observed that these mice began dying off as early as 13 weeks of age. It is also important to note that we show sustained substrate reduction up to 16 weeks, which is just a few weeks shy of the normal lifespan of a *Hexb*^{-/-} mouse. Sixteen weeks was chosen as a humane endpoint in this study due to the well-documented rapidly progressive decline in motor function leading to paralysis and death in controls; a lifespan/survival study which included the untreated controls would have therefore been of major ethical concern, and further aging of only the treatment groups would have rendered us unable to directly compare pathology to controls. While we agree that a follow up study showing the duration of survival following treatment would be of great interest, we believe this is outside the scope of the current manuscript due to time constraints. Regarding the reviewer's comments about BMT treatment alone, it is important to note that the BMT + CSF1Ri treatment led to the complete or near-complete clearance of several pathological hallmarks that were not resolved within the BMT group, including the newly added GM2 ganglioside data (see response to the comment immediately below + the new Figure 5 panels a-c). We believe that these significant improvements in disease phenotype and pathology are sufficient to demonstrate the superiority of the BMT + CSF1Ri treatment over BMT alone.

3. GM2 ganglioside accumulation in neurons is the biochemical hallmark of Sandhoff disease. If Hexb enzyme is transferred from wild-type microglia to enzyme-deficient neurons, it would be expected to reduce GM2 ganglioside burden. The authors used PAS staining to show a reduction in lysosomal storage, but PAS staining is not specific to GM2 ganglioside and instead reacts with various carbohydrate substrates. Demonstrating a significant reduction of CNS GM2 levels in BMT + CSF1Ri mice compared to BMT-only mice would provide strong evidence for the central hypothesis of the study: that enzyme supplied by microglia or macrophages is transferred to neurons and reduces GM2 ganglioside storage in Sandhoff disease.

We thank the reviewer for this excellent suggestion. We have now performed additional experiments to directly quantify GM2 ganglioside burden in mouse brain homogenates from our experimental groups via mass spectrometry. The new text and data can be found in lines 525-544 and Figure 5a-c. The results show a statistically significant reduction in GM2 ganglioside content in comparison to *Hexb*^{-/-} controls as a result of the combined BMT + CSF1Ri approach, which is not achieved with BMT alone. In combination with the previously shown PAS data, which also stains other substrates of the Hex β enzyme as the reviewer notes, we believe this provides strong evidence of enzyme transfer and reduction/clearance of pathological burden.

Reviewer #2 (Remarks to the Author):

The current manuscript by Tsourmas and colleagues is conceptionally interesting, describing the effect of a complete microglial exchange/replacement in the mouse model of Hexb-deficiency that mimics human Sandhoff disease to some extent.

Despite the fact that the idea of microglial replacement by bone marrow reconstitution is not new at all and the combination with prior microglia depletion with as also already introduced by the Bo Peng and other groups before (Zhang et al. *Neurosci Bull* 2023, Xu Z et al. *STAR Protoc* 2021, Xu Z et al. *Cell Rep* 2020) some aspects of this study are interesting.

The novelty is based on the fact that the authors are able to rescue the lethal phenotype associated with Hexb-deficiency in mice. However, despite being interesting, the mechanisms are largely unclear and further extensive experiments are clearly needed to substantiate their findings.

In general, the manuscript is quite overloaded, very dense and difficult to read with already 175 references. I would therefore strongly suggest to improve reading and understanding by focusing on the main points of the paper.

- In detail, authors should put emphasis on their findings in figures 2-6 highlighting bone marrow chimeras and their phenotypes and add more supportive experimental data. In contrast, figure 1 is interesting but mainly shows that a) Hexb is microglia-specific (that was already nicely shown by Hickman et al. *Nat Neurosci* 2013) and b) that Hexb microglia show a “DAM-like” signature that is somehow expected.

In order to improve readability as suggested by the reviewer, we have added and edited the text associated with Figure 1 (lines 118-229). While we agree that there are aspects of Figure 1 that align with previous reports, this is the first ever presentation of a spatial transcriptomic dataset in Sandhoff disease mice. In this figure, we detect transcriptomic alterations in various cell types, including but not limited to microglia, demonstrating the widespread effect of loss of *Hexb*. We think it is especially important that it establishes a number of transcriptional changes in neurons associated with apoptosis and cellular stress due to loss of *Hexb*, which we later show are reversed with BMT + CSF1Ri. Additionally, the identification of region-specific gene expression patterns that closely align with regional vulnerabilities identified in human patients is a novel utilization of this technology.

We have also added more supportive experimental data in the form of GM2 ganglioside burden assessment via MALDI mass spectrometry (Figure 5a-c), showing that BMT + CSF1Ri significantly reduces pathological accumulation of the most common SD pathological hallmark, and immunohistochemical assessment of GFAP+ astrocyte volume (Figure 5n-o) showing reduction in astrocyte activation in BMT + CSF1Ri-treated mice in comparison to BMT. Both measures demonstrate the efficacy of microglial replacement in improving pathological manifestations of SD.

- Similarly, figure 7 does not add so much to the microglia replacement findings and could be omitted.

We thank the reviewer for this comment, but we respectfully disagree and believe that Figure 7 represents not only another novel aspect of the manuscript, but also an important one. In this section of the study, we aim to understand the mechanism(s) by which Hex β derived from myeloid cells alleviates the neuronal phenotype by exploring if Hex β is released by microglia and taken up by neurons. Instead of removing this aspect of the manuscript, we have performed additional *in vitro* experiments using primary hippocampal neurons to assess the mechanism of neuronal Hex β protein uptake. Through these experiments, we have identified that neuronal uptake of Hex β is likely mediated by the mannose 6-phosphate receptor, which is also associated with the pathway by which lysosomal enzyme secretion occurs, and clathrin-mediated endocytosis (See lines 689-714, 720-727, 795-824 in the text, and Figure 7h-k).

- One main weakness of the study is the experimental design. The authors used whole bone marrow that is a highly complex mixture of several myeloid cell types (HSC, MDSC, monocytes, CMP etc.) that potentially all can enter the CNS and prevent disease. For the translation value of this study it is highly desirable to identify by adoptive transfer study the nature of the myeloid subset that improves disease.

We thank the reviewer for identifying this issue. In an elegant recent study, Shibuya et. al (2022) performed an adoptive transfer experiment (performed similarly to the reviewer's suggestion) and profiled infiltrating myeloid cells following BMT + CSF1Ri in a similar paradigm (the authors utilized busulfan rather than whole body irradiation for myeloablation). The authors found that hematopoietic stem cells (HSCs) have the highest capacity to give rise to the cells that replace microglia under this paradigm. While we had previously cited this study in our manuscript, we have provided additional information in our manuscript (lines 309-313) to provide more context on this topic.

In addition to these text additions, we also spent a significant amount of time dedicated to addressing this concern during the revision process. We considered it a high priority (as the reviewer suggested) to identify the therapeutic population of interest which was able to reconstitute the missing enzyme in our *Hexb*^{-/-} CNS and identify whether engrafted cells retain an HSC signature after enduring engraftment in the brain. Thus, to address the reviewer comment regarding the nature of the engrafted population, we added a new cohort of mice that underwent BMT + CSF1Ri treatment and used flow cytometry analysis to confirm the cellular identity of the engrafted cells (see lines 320-339). To avoid unnecessary duplication of the aforementioned study by Shibuya et al., brain myeloid cells were collected six weeks after engraftment into the brain to not only maintain similarity to the previously generated *Hexb* chimeras in this study, which already had multi-week cell engraftment and potential cell differentiation prior to disease onset, but also given our knowledge of the timing and dynamics of myeloid cell replacement (i.e., it usually takes 21-28 days for full brain reconstitution, and that early timepoints may yield insufficient cell numbers for flow analysis [Hohsfield et al. 2020, *J Neuroinflamm*]; and that in some cases the monocyte to microglial-like transition takes up to 62 days [Wang et al. 2024, *Cell Reports*]). Specifically, we generated 20 new chimeric mice using CAG-EGFP and CX3CR1-GFP-CCR2-RFP donors and assessed the identity of the donor-derived cells in the brain via flow cytometry (including markers for CD34, Sca-1, ckit, CD11b, Ly6C, Ly6G, CD45, CCR2, etc.) and histology, respectively. Our additional flow cytometry experiment demonstrated that ~95% of GFP⁺ cells (i.e., engrafted bone marrow-derived cells) were CD45^{hi}Cd11b⁺Sca-1⁻ckit⁻Ly6G⁻Ly6C⁻CCR2⁻ (Extended Data Figure 6b-h). These data provide evidence that the engrafted bone marrow derived cells resemble macrophages and/or non-classical (Ly6C^{lo}) monocytes rather than microglia (CD45^{lo}Cd11b⁺), HSCs (ckit⁺Sca-1⁺), classical monocytes or M-MDSCs (CCR2⁺ or Ly6C⁺) common myeloid progenitors/granulocyte myeloid progenitors (CD45^{hi}Cd11b⁺ckit⁺Sca-1⁻), HSCs (ckit⁺Sca-1⁺), or PMN-MDSCs/neutrophils (CD45^{hi}Cd11b⁺Ly6C⁺Ly6G⁺CCR2⁻) (Extended Data Figure 6b-h). We also show that irradiation and transplant of WT mice with CX3CR1-GFP/CCR2-RFP bone marrow results in 75% engraftment of all myeloid cells with CX3CR1-GFP⁺ cells (Extended Data Figure 6i-l, see lines 340-347). In addition, to further validate the gene expression signature of engrafted cells, we conducted an additional single cell spatial transcriptomics experiment on the previously generated control, BMT, and BMT + CSF1Ri brains (collected between 7-9 weeks recovery) using a new custom-created probe set that included a probe for *Gfp*, allowing us to evaluate cell specific gene expression in GFP or bone marrow derived cells. In comparison to microglia in WT controls (i.e., Myeloid 1, which express canonical microglial genes, including *Sall1*, *P2ry12*, *Siglech*, and *Tmem119*), *Gfp*⁺ engrafted cells (i.e., Myeloid 2) are enriched in several genes that we identified as monocyte signature genes (Hohsfield et al. 2020), including *Apobec1*, *Lyz2*, *Mrc1*, *Lilrb4*, and *Msr1*, as well as genes expressed by macrophages and disease-associated myeloid cells (*Mrc1*, *Msr1*, *Cd68*, *Trem2*, *Tyrobp*, *C1qa/b/c*, *Ctss*, *Apoe*, *Cx3cr1*, *Itgam* [CD11b], *Ptprc* [CD45], including genes associated with lysosomal/phagocytic activity (e.g., *Lyz1/Lyz2*, *Ctsb*, *Ctsd*, *Ctsz*) and macrophage immune modulation and tissue repair (e.g., *Tgfb1*, *Tgfb1*) (Extended Data Figure 7e). A previous study has shown that non-parenchymal macrophages are also enriched for *Mrc1* and *Msr1* (Utz et al. 2020, *Cell*). Results are described in detail in the manuscript (Lines 348-368, Extended Data Figures 6-7. Thus, taken together, our data provides evidence that the engrafted bone marrow-derived cells found in *Hexb*^{-/-} BMT + CSF1Ri brains are macrophages and/or non-classical monocytes.

We appreciate the reviewer's comment regarding translational relevance, as translation is very important to us. HSC transplant is the first line of treatment for other lysosomal storage disorders such as Gaucher's disease and Hurler's syndrome but falls short in conditions with a heavy CNS burden due to the difficulty in directing donor-derived cells to the brain with traditional BMT approaches. A major goal of this manuscript was to improve upon existing BMT paradigms and to test a translationally relevant treatment for SD and other CNS

lysosomal storage disorders. Given that infusion of a specific myeloid subtype into a human SD patient without myeloablative preconditioning and an HSC transplant would result in very poor engraftment of that cell and any of its progeny in the brain (as it would be competing with the endogenous host myeloid cells), we do not agree that an adoptive transfer experiment is the most translationally relevant approach. We agree that identifying specific cell type(s) is important, and adoptive transfer offers an attractive approach to understand this; however, we have serious concerns about the survivability of mice if myeloablation is done without subsequent adequate HSC or bone marrow reconstitution. We also have concerns regarding the survivability of other myeloid populations with adoptive transfer, especially for non-progenitor populations such as monocytes, which have a short lifespan. Shibuya et. al (2022) showed that adoptive transfer of specific cell subsets (i.e., HSCs, CMPs, GMPs) could be done, but the authors followed these transfers with administration of unlabeled whole bone marrow “rescue cells” to increase survivability, which we believe would compromise our ability to accurately assess the origin of engrafted cells due to competition. The authors showed that specific cell subsets were outcompeted by whole bone marrow, except for HSCs.

- Another weakness is the lack of mechanistic explanation how *Hexb*-deficiency leads to microglial activation. This main point was not addressed in this manuscript at all. Fig.7 falls short in providing some mechanisms in this direction.

We thank for the reviewer for the comment – mechanisms of microglial activation are of interest, but they were not the focus of this study. In Figure 1h, we show that myeloid cells in *Hexb*^{-/-} mice express *Apoe*, *Ctsd*, *Ctsb*, *Lyz2*, and *Tyrobp*, which are known disease-associated microglia signature genes. In Figure 2f-m, we also show that microglia in *Hexb*^{-/-} mice exhibit shorter processes, fewer branches, and larger cells bodies, which are all morphologically consistent with an activated microglial phenotype. We would suspect that the observed microglial activation is a response to the accumulating GM2 gangliosides and other glycolipid species within neurons, which progressively drives neuronal dysregulation and eventual mass neuronal apoptosis—as microglia are very reactive to changes in neuronal health—as well as the deleterious deprivation of functional Hexβ. This point has been considered and discussed at length in previous studies and review articles related to Sandhoff and Tay-Sachs diseases: see Wada et al., 2000 and Ogawa et al., 2017 in particular. With microglial activation previously identified as a key pathological hallmark of Sandhoff disease, we chose to focus not on mechanisms of activation but instead whether replacement of *Hexb*^{-/-} myeloid cells with *Hexb*-sufficient cells could resolve disease phenotypes including myeloid cell activation. To address the reviewer’s concern, we have added commentary on this matter to the Discussion (Lines 755-767). Of note, we are also in the process of transplanting *Hexb*^{-/-} BM into WT mice and subsequently performing microglial replacement as described in the current manuscript, which will inform us if loss of *Hexb* itself induces microglial reactivity. However, the inclusion of these experiments and data analysis is not feasible for the current manuscript.

In sum, the authors should address the main points below to strengthen the validity of their findings and to improve the focus of the manuscript.

Major points:

1. Figure 1: This figure is difficult to digest and does not help the overall story, because
 - a) all CNS cells are explored by biased (!) spatial transcriptomics approach leading to the description of endless neuronal subsets and leaving only very few left CNS myeloid cells,

We apologize to the reviewer if Figure 1 was difficult to digest. To address the reviewer’s concern, we have made changes to the titles in Figure 1 to make the figure more digestible. If the reviewer has other suggestions for improved digestibility in Figure 1, we would be more than happy to make those modifications.

However, regarding reviewer’s comment about the inclusion of Figure 1 in this manuscript, we believe there is high value in understanding the distinct single cell gene expression changes in *Hexb*^{-/-} mice, before exploring the therapeutic benefits of myeloid cell replacement in this disease model. This is the first study that has explored the *Hexb* deficient brain at a single cell level with spatial resolution.

In response to the reviewer's comment about a "biased spatial transcriptomics approach," we would like to highlight that all technologies have advantages and disadvantages. We recognize that our CosMx spatial transcriptomics approach does have limited gene coverage, allowing currently for the exploration of 1000 genes. However, we would like to remind the reviewer that this is cutting edge technology. We would also like to highlight that despite this gene coverage, the data still offers value in exploring single cells at a spatial level. In this study, we were able to capture 196,533 cells in the brain (with a mean transcript count of 800 transcripts per cell and identify 39 distinct cell type clusters). This technology also offers the advantage of true single cell resolution compared to spot-based sequencing transcriptomics approaches. Spots can lie on a group of cells and so deconvolution is necessary to identify which cells are represented by the spot, adding nuance to the correct and adequate identification of single cells. Moreover, our data provides robust and thorough datasets (which we have included as open source data files for others to explore) to evaluate changes in cell populations and widespread changes in their gene expression profiles. Also of note, we have *not* selected the genes/proteins within our panels and have instead used predetermined panels designed for maximal coverage of all cell types and states in the brain. Thus, while the technique is "biased," we would argue that "targeted" better represents the reality. This approach is also far more effective in recovering myeloid cells (compared snRNA-seq or scRNA-seq approaches, which typically capture 1-2% of microglia from all CNS cells), as we recover ~99% of myeloid cells within each section. We recognize that there are fewer myeloid cell *clusters* than neuronal clusters, but this represents the actual reality of the homogeneous nature of microglia/CNS myeloid cells. This is a considerable advantage over snRNA-seq and scRNA-seq, which are known to induce transcriptional changes in myeloid cells during the isolation process. Language has been added to the manuscript regarding this point (see lines 129-133).

b) the nature of the described myeloid cells remains totally unclear (DCs or monocytes or microglia etc.),

We thank the reviewer for this comment and have added text to the manuscript describing our assessment of the nature of the myeloid cells in Figure 1 (see lines 147-162). The Myeloid 1 subcluster expresses genes associated with a homeostatic microglial signature (e.g., *Csf1r*, *Hexb*, *Pr2y12*, *Cx3cr1*, *Tmem119*, *Sall1*), leading us to believe that their identity is most likely to be microglia. The top enriched genes in the Myeloid 2 subcluster, which is found primarily in *Hexb*^{-/-} brains, include genes associated with MHCII antigen presentation and cell activation or disease-associated markers (e.g., *H2-Aa*, *Cd74*, *H2-Ab1*, *Lyz1/2*, *Ptpnc1*, *Ctss*, *Itgax*, *Axl*, *ApoE*, and *Cst7*).

c) no protein validation was performed and finally

Extensive spatial proteomic data which strongly corroborates our spatial transcriptomic findings in Figure 1 and Figure 3 can be found in Figure 4.

d) *Hexb* specificity was already shown by others before (Hickman et al. Nat Neurosc 2013). In sum, I would suggest to remove to figure to the suppl. section or to omit it. Further, analyses on 3 mice each is not sufficient. More mice are definitely needed.

In addition to our above comments regarding the utility of Figure 1, we would like to clarify that the n of 3 listed for the spatial transcriptomics experiments is a smaller subset taken from the larger groups used in the remaining figures, which are composed of 10-11 mice per group for an n of >60. Three brain sections (randomly assigned) from each group were then partially allocated for the spatial transcriptomics experiments. The constraints of the current spatial transcriptomics setup are such that a maximum of no more than 6 brains can be imaged per slide, and only 2 slides can be run concurrently, leading to a maximum n of 12 per experiment. For an experiment involving 4 groups, or in the case that we are only able to run one slide due to resource constraints, this means that we must adhere to an n of 3 per group. Additional constraints are introduced by the cost of each run (~\$12,000) and the time—each run takes ~2 weeks to complete, with much of that time in the instrument itself. We would also like to note that an n of 3-4 per group is very common in the

literature for spatial transcriptomic experiments, and the priority is often placed upon the number of cells captured—in our case, we have captured over 190,000 cells with a high mean transcript per cell value of ~800, representing the very upper end of captured transcripts produced to date using these imaging modalities.

2. As already mentioned above the *Hexb*^{-/-} disease rescue by wt bone marrow transfer is exciting but the nature of the disease modulating myeloid cell remains completely unclear. Due to the highly complex mixture of the bone marrow compartment several myeloid cells can be the candidates. Thus, the cell of action needs to be identified (HSC versus MDSC versus monocytes versus CMP etc.).

We thank the reviewer for this comment. We agree that the rescue of *Hexb*^{-/-} is extremely exciting and of high therapeutic value. Thus, we considered it a high priority to identify the therapeutic population of interest which was able to reconstitute the missing enzyme in our *Hexb*^{-/-} CNS. As mentioned in our comment above, we have added new flow cytometry, reporter mice, and spatial transcriptomics experiments to address this (see new Extended Data Figures 6 and 7). As mentioned above, Shibuya et. al (2022) performed an adoptive transfer experiment and profiled infiltrating myeloid cells following BMT + CSF1Ri in a similar paradigm to our experiment. The authors found that hematopoietic stem cells (HSCs) have the highest capacity to give rise to the cells that replace microglia under this paradigm. Thus, we focused on identifying the cell that actually persists in the *Hexb*^{-/-} CNS to rescue disease phenotypes, as described above in detail.

3. Figure 7: This figure starts to explain the crosstalk between *Hexb*, microglia and neurons. It is somehow a start for a better understanding underlying pathophysiology but fails to explain WHY loss of a microglial enzyme leads to massive neurodegeneration seen in *Hexb* mice. I would therefore suggest to remove this immature part from the manuscript.

The rationale for the neurodegeneration observed following the loss of *Hexb* has been a frequent topic of discussion in various other manuscripts and review articles pertaining to SD and other LSDs, which have largely concluded that neuronal apoptosis is likely the result of the accumulation of Hex β substrates that cannot be broken down, including GM2 gangliosides and other glycolipids. Building upon this, recent elegant work demonstrates that neuronal cell death in response to the accumulation of lysosomal substrates is the result of neuron-intrinsic cGAS–STING signalling (Wang et al. 2024, Nature Cell Biology). We discuss several of these relevant findings in paragraph 3 of the Discussion section. As such, our intention was not to assess the causes of neurodegeneration, but rather to understand the mechanisms by which enzyme transfer may occur, namely the potential mechanism by which microglia release Hex β enzyme and how neurons facilitate its uptake. We acknowledge that the initial version of Figure 7 represented an incomplete mechanistic understanding, and we have subsequently carried out additional investigation of the potential pathways of neuronal uptake which have now been added as Figure 7h-k. We suggest that the mannose 6-phosphate pathway is a likely means of Hex β secretion by myeloid cells, and mannose-6-phosphate receptors on the cell surface, in turn, as the likely mediator of Hex β uptake by neurons. These data align well with existing literature regarding lysosomal enzyme sorting, secretion, and uptake, including a study on the heteromeric isoform of the β -Hexosaminidase enzyme (HexA) which shows it is taken up by the mannose-6-phosphate receptor.

Minor points:

1. The generation of bone marrow chimeras for the delivery of GFP bone marrow was already described by the Priller group in 2021 Nat Med and therefore this pioneering study should be cited.

We apologize for this oversight and thank the reviewer for drawing this to our attention. We have added a citation of this critical 2001 study by the Priller group.

2. Line 81: The term “neuroinflammation” is wrongly used because only conditions with infiltration of hematogenic cells such as lymphocytes, neutrophils etc. can be considered as true neuroinflammation (see also the white consensus paper on this topic Paolicelli et al. Neuron 2022).

The line has been updated, and we thank the reviewer for this correction.

Reviewer #3 (Remarks to the Author):

This manuscript presents a comprehensive study on the potential of microglial replacement therapy for treating Sandhoff disease (SD) using a combined bone marrow transplant (BMT) and colony stimulating factor 1 receptor inhibition (CSF1Ri) approach in a mouse model. The authors demonstrate that this treatment strategy leads to significant improvements in disease pathology, including reversal of gene expression changes, clearance of substrate accumulation, and normalization of lysosomal phenotypes in neurons. Below are some major and minor comments:

1. Sample Size for spatial analysis (Figure 1a as an example): The sample size of three mice per group is notably small for a mouse study. While acknowledging the challenges of in vivo experiments, this sample size is insufficient to provide robust and generalizable results. Current standards typically require larger groups, even for human studies, to ensure statistical validity and reproducibility. Consider increasing the number of mice per group to strengthen the study's conclusions.

We would like to clarify that the n of 3 listed for the spatial transcriptomics experiments is a smaller subset taken from the larger groups used in the remaining figures, which are composed of 10-11 mice per group for an n of >60. Three brain sections (randomly assigned) from each group were then partially allocated for the spatial transcriptomics experiments. The constraints of the current spatial transcriptomics setup are such that a maximum of no more than 6 brains can be imaged per slide, and only 2 slides can be run concurrently, leading to a maximum n of 12 per experiment. For an experiment involving 4 groups, or in the case that we are only able to run one slide due to resource constraints, this means that we must adhere to an n of 3 per group. Additional constraints are introduced by the cost of each run (~\$12,000) and the time—each run takes ~2 weeks to complete, with much of that time in the instrument itself. We would also like to note that an n of 3-4 per group is very common in the literature for spatial transcriptomic experiments, and the priority is often placed upon the number of cells captured—in our case, we have captured over 190,000 cells with a high mean transcript per cell value of ~800, representing the very upper end of captured transcripts produced to date using these imaging modalities.

2. Deconvolution of spatial transcriptomics data to single-cell resolution is particularly challenging, especially in disease settings where microglial populations can fluctuate significantly. Given that reference maps like the Allen Brain Atlas are based on healthy samples, it is unclear how single-cell resolution was confidently achieved in this study. Please provide a detailed explanation of the methods and validations used to ensure accuracy in applying the Allen map to a diseased context.

Unlike spot-based transcriptomics, this is true single cell spatial transcriptomics and therefore requires no deconvolution. It utilizes Multiplexed Error-Robust Fluorescence in Situ Hybridization (MERFISH), which relies on single cell segmentation based on a number of fluorescent probes which allow for highly accurate (~98%) segmentation. We understand that this is a relatively new technology and have therefore provided Extended Data Figure 1 to demonstrate extensive examples of the efficiency of segmentation, even in densely packed regions such as the dentate gyrus.

3. Long-term effects: The study focuses on a specific time point (16 weeks). It would be valuable to assess the long-term effects of the treatment, including survival data and whether the improvements are sustained over time.

As mentioned in the response to Reviewer 1, the selected timepoint (16 weeks of age) was chosen to allow for direct comparison between treatment groups and the untreated control *Hexb*^{-/-} animals. This time point is just a few weeks prior to the average age of death in untreated *Hexb*^{-/-} mice and was thus designated as the humane endpoint in this study. We note that the manuscript does include some incidental survival differences in each group—3 mice in the untreated *Hexb*^{-/-} control group, 4 mice in the *Hexb*^{-/-} BMT group, and 1 mouse in the *Hexb*^{-/-} BMT + CSF1Ri group either died or reached the previously designated humane endpoint prior to 16 weeks. Like the reviewer, we are also interested in the extended survival of the BMT + CSF1Ri-treated animals. Of note, we can report that *Hexb*^{-/-} being treated with a similar paradigm (i.e., microglial replacement

with Hexb-sufficient cells) for a separate study have reached the age of 25 weeks with no behavioral abnormalities.

4. Dose-response relationship: Investigate whether different doses of CSF1Ri or varying durations of treatment affect the efficacy of the therapy.

In the current study, we demonstrate that the overall rate of engraftment is correlated with the degree of improvement, with animals receiving BMT only showing very limited engraftment and subsequent limited improvement in behavioral and pathological manifestations. Our dosages of CSF1Ri to induce broad peripheral cell infiltration were selected based on prior studies, many from our lab, which established the doses required to cause sufficient microglial depletion and, later, to reliably induce high rates of peripheral cell infiltration following irradiation—see Elmore et al., 2014, Cronk et al., 2018, Hohsfield et al., 2020, and Hohsfield et al., 2021 for background on the effects of different dosages in this context. From these previous studies, we can assume that lower doses would be unlikely to yield sufficient microglial depletion to facilitate peripheral infiltration to the degree that is necessary for the observed rescue of Sandhoff disease pathologies

5. Behavioral assessments: While the authors present data on motor function using the Rotarod test, including additional behavioral tests to assess cognitive function and other neurological symptoms would provide a more comprehensive view of the treatment's effects.

Though we agree that it would be ideal to also assess cognitive function as this is also affected in human Sandhoff disease patients, *Hexb*^{-/-} mice exhibit severe motor decline starting from week 13 which progresses to near paralysis and extremely limited movement by weeks 15 and 16. Unfortunately, cognitive assessment is not possible in animals with such severely limited mobility under any of the paradigms our lab has previously employed, and thus no meaningful comparisons of the cognitive performance of treated animals with untreated controls would be possible.

6. Mechanistic Clarity on Hexb Secretion: While the study shows Hexb secretion from microglia and uptake by neurons, the exact mechanisms of lysosomal trafficking and uptake remain speculative. For example, how Hexb integrates into neuronal lysosomes needs further exploration, particularly under pathological conditions.

We thank the reviewer for this suggestion and have performed additional *in vitro* experiments using cultured hippocampal neurons to assess the mechanism(s) of neuronal Hex β uptake in response. Through these experiments, we have determined that neuronal uptake is most likely to be mediated by the mannose-6-phosphate receptor and facilitated by clathrin-mediated endocytosis. Most lysosomal enzymes are transported from the endoplasmic reticulum to the late endosome via the mannose-6-phosphate (M6P) pathway after being tagged with an M6P group. M6P can then be recognized by one of two M6P receptors. Importantly, a large percentage (~40%) of enzyme escapes from this pathway and is secreted into the extracellular space; from here, M6P receptors on the cell surface of neighboring cells can facilitate uptake. As this pathway is common to many lysosomal enzymes, including Hex β , we identified it as a likely candidate for the uptake of Hex β into neurons and found that treatment with the allosteric M6P receptor inhibitor IGF-II sharply reduces the number of neurons with internalized Hex β . For the updated information, see the new panels in Figure 7k-h and Extended Data Figure 15, and lines 689-714, 720-727, and 795-824 in the text.

7. Interpretation of Inflammatory Role: While inflammation is not the main focus, the study hints at its interplay with Hexb secretion. However, the manuscript could benefit from discussing whether inflammatory conditions alter Hexb function or neuronal lysosomal health beyond secretion dynamics.

The rationale for the neurodegeneration observed following the loss of Hexb has been a frequent topic of discussion in various other manuscripts and review articles pertaining to SD and other LSDs, which have largely concluded that neuronal apoptosis is likely the result of the accumulation of Hex β substrates that cannot be broken down, including GM2 gangliosides and other glycolipids. Building upon this, recent elegant work demonstrates that neuronal cell death in response to the accumulation of lysosomal substrates is the result of neuron-intrinsic cGAS–STING signalling (Wang et al. 2024, Nature Cell Biology). We discuss several of these relevant findings in paragraph 3 of the Discussion section. To summarize here, previous investigations

have shown that GM2 ganglioside, which accumulates abundantly in various cell types in SD, activate microglia via protein kinase C and NADPH oxidase *in vitro*. Deletion of neuroinflammatory factors such as tumor necrosis factor- α (TNF- α), activating immune receptor (Fc γ), and macrophage-inflammatory protein 1 α (MIP-1 α /CCL3) reduce neurodegeneration and slightly extend the lifespan of *Hexb*^{-/-} mice, but these inflammation-related interventions did not reduce pathological neuronal GM2 ganglioside burden in *Hexb*^{-/-} mice. We have added additional points to the discussion regarding secretion dynamics and potential mechanisms such as the M6P pathway/receptor (see lines 794-824). However, because restoration of Hex β by microglial replacement appears to drastically reduce inflammation in the case of both myeloid cells and astrocytes, it is difficult to speculate how inflammation might affect the function of Hex β , at least in our model—the observed inflammation appears to be a direct consequence of systemic Hex β deprivation which is resolved upon its restoration along with lysosomal abnormalities within neurons. Additionally, since *Hexb*^{-/-} mice lack any functional Hex β enzyme, inflammation in these mice without any sort of enzyme restoration will not affect secretion of Hex β .

8. Sex differences: Is there a sex difference? Analyze whether there are any sex-specific differences in treatment efficacy or disease progression. Or make a statement in that regard.

We thank the reviewer for noting this omission; groups were composed of an equal number of males and females and data were initially analyzed with sexes separated to detect any sex differences; however, no sex differences outside of males having heavier weights than females were observed in any of the behavioral outcomes, and there were no differences detected in any of the immunohistochemical, biochemical, or transcriptomic assessments. A comment regarding this observation has been added to the discussion in lines 848-850.

9. Controls and Comparisons: The manuscript does not explicitly address whether the replacement efficacy of BMT + CSF1Ri differs by region. It would strengthen the conclusions if regional efficacy were correlated with therapeutic outcomes (e.g., motor performance vs. corpus callosum infiltration

This analysis was previously performed and can be observed in Figure 2e and Extended Data Figure 5d-f; it is addressed in lines 282-296 in the text. Briefly, we observed that the degree of infiltration in the corpus callosum was positively correlated with the maintenance of motor performance: increased infiltration was associated with improved performance. No significant correlation was observed with motor performance and infiltration rates in other assessed brain regions. After careful observation of GFP+ cell deposition in sagittal sections from each BMT-recipient brain, we did not observe differences in regional replacement efficacy with CSF1Ri, and variability in infiltration rates did not appear to have any regional specificity.

Minor Concerns

- Figures demonstrating spatial transcriptomics and proteomics results are not included in the snippets. Ensure that the visualizations effectively highlight cell-specific and region-specific changes and that axis labels and legends are clear.

Figures 1, 3, and 4 and Extended Data Figures 1-4 and 7-14 all pertain to spatial transcriptomics and proteomics and address cell-specific changes and region-specific changes. If the reviewer could provide clarity on “the snippets” they are referring to, we would be happy to include the results where necessary.

- Provide additional quantitative comparisons for the therapeutic effects of BMT vs. BMT + CSF1Ri across different brain regions.

The panels in the spatial transcriptomics figures pertaining to DEG score, and particularly the spatial heatmaps demonstrating regional heterogeneity in Figures 1 and 3, show a quantitative assessment of the brain regions most affected by the loss of *Hexb* (Figure 1g) and, in turn, the regions most improved with BMT + CSF1Ri (Figure 3g) which are not improved with BMT alone. The aforementioned correlation analysis also pertains to this comment. Though the confocal representative images in the manuscript figures mainly show the cortex, it is worth noting that other brain regions were also imaged and quantified for every stain, neuronal phenotypes,

PAS+ staining, LAMP1+ staining, and myeloid cell activation metrics were largely consistent across regions; therefore, these additional images/quantifications were not shown for the sake of brevity.

- Include a more in-depth discussion on how this work bridges gaps in SD pathology, specifically the non-cell-autonomous roles of microglia.

Paragraph 3 of the discussion has been updated and expanded with regards to this point.

- Expand on the molecular pathways of Hexb secretion and neuronal uptake, particularly the role of P2X7 signaling under inflammatory conditions.

We thank the reviewer for this comment, and we have carried out additional investigation into the potential pathways of neuronal uptake during the revision process. These results have now been added to Figure 7k-h and are discussed in the context of our previous findings regarding secretion from microglia in lines 689-714, 720-727, and 795-824 in the text.

Dear Reviewers,

We would like to first thank the reviewers for their time and thoughtful consideration of the manuscript. Please see our point-by-point response to the reviewers below for details on how we have addressed their concerns. All changes to the manuscript are indicated with blue text.

Regarding Reviewer #2's concerns, we respectfully disagree with their comments and expectations of the revised manuscript – we have added our reasons below. While the reviewer proposes valid experiments, we believe these experiments are beyond the scope of our proof of principle manuscript. Moreover, the concerns raised, and experiments proposed by Reviewer #2 would either require an additional 10-12 months of experiments at the minimum, which we consider to be long-term experiments that are therefore outside the scope of the current manuscript, and/or are not currently possible due to technical limitations. We also do not believe that our manuscript should be required to overcome all potential translational issues in a single study – the current state of this new research is not at a stage to translate to humans. We also find it unfair to be held responsible for the lack of reproduction of another study utilizing bone marrow transplants by another group in a different disease with a different paradigm.

Despite this, we hope that you find the revised manuscript suitable for publication in *Nature Communications*. We will continue to progress this work for a follow-up manuscript, which will include experiments detailed by Reviewer #2. We have put a great deal of work into this manuscript and the resubmission, and we believe it is a promising line of research for Sandhoff's disease (SD) that could lead to meaningful interventions down the road.

Sincerely,

Kim Green

REVIEWER COMMENTS

Reviewer #1 (Remarks to the Author):

The authors have responded satisfactorily to my comments. The manuscript presents novel findings that support a significant role for microglia in Sandhoff disease, a disease in which neuronal cell death dominates.

However, I would like to note that, contrary to the authors' statement "neurons within the Hexb^{-/-} mouse brain also lack GM2 accumulation until several months of age; substrate buildup requires sustained deprivation of Hexβ protein in vivo.", GM2

ganglioside can, in fact, be detected in the fetal brain of Sandhoff mice, (<https://doi.org/10.1016/j.neures.2019.07.004>).

We thank the Reviewer for this clarification.

Reviewer #2 (Remarks to the Author):

As in the first round of reviews this referee acknowledges especially the novelty of this story, namely that bone marrow transfer in microglia-depleted Hexb-deficient recipients is able to stop clinical symptoms.

Despite some general improvements of the study, two major but essential points from my first round of reviewing still remain open. I will describe below in depth why it is so important to clarify them before publication of this interesting topic.

Major points:

1. One still remaining weakness of the study is the usage of whole bone marrow to rescue Hexb-deficient brains. As I wrote before, the bone marrow contains a highly heterogeneous mixture of myeloid cells (monocytes, HSC, MDP, CMP etc.) that ALL have the potential to replace microglia cells in the brain. Even short-living monocytes have just very recently shown by the S.Jung group to be potential microglia precursors for aged microglia (PMID: 40279248).

Why is important to identify by short-term adoptive transfer experiments (that will last max. three months) the precise nature of the bone marrow myeloid subset that improves disease? Here is the reason: A similar bone marrow rescue has been shown in Mecp2-deficient mice with the overall claim that bone marrow-derived phagocytes could rescue the microglia phenotype in Mecp2 mice (Derecki NC et al. Nature 2012, PMID: 22425995). Based on this report that included an insufficient characterization of the transferred bone marrow subsets, clinical trials of bone marrow transfer for patients with Rett-Syndrome (Mecp2 deficiency) have been initiated (MT2013-31:Allogeneic Hematopoietic Cell Transplantation for Inherited Metabolic Disorders, Sever Osteoporosis and Males with Rett Syndrome Following Conditioning with Busulfan (Therapeutic Drug Monitoring), Fludarabine +/- ATG. University of Minnesota; Minneapolis, MN USA) with unsuccessful results. Only later, the primary data from the initial Nature 2012 paper were shown to be non-reproducible (Wang J et al. Nature 2015, PMID: 25993969). Taken together, the easy-to-do adoptive transfer experiments are simply essential for translational reasons. Otherwise, there is a significant risk of mis- or overinterpretations of the data that could be fatal for future patient's curative approaches. The newly added flow data on Extended Data Figure 6 are not sufficient because they didn't used individually sorted and transferred cells. I simply do not agree with their given arguments.

We appreciate the reviewer's perspective, but disagree with the importance and feasibility of the proposed experiment to the current study for the following reasons:

- 1) It is not reasonable to expect us to overcome all translational issues in a single study/manuscript. We have demonstrated that bone marrow transplants + CSF1R inhibition are able to prevent SD pathology – this is proof of principle. Whether or not this approach will be feasible in humans in its current form is unknown, but BM transplants have already been utilized in SD and Tay Sachs disease – the further use of CSF1R inhibitors to facilitate brain engraftment is suggested by our data to be highly beneficial, but there are many issues to overcome first before progressing into humans. Making the jump to adoptive transfer in this manuscript is premature, and unnecessary, as the reviewers' rationale above is based on a completely different disease, and unproven in SD.
- 2) Further, we find it unlikely that adoptive transfer would result in sufficient engraftment of therapeutic cells into the brain under the conditions we are describing, as the influx of exogenous Hex β -sufficient cells into the brain would also be competing with the Hex β -null host bone marrow derived cells. Our paradigm requires both myeloablative preconditioning (lethal irradiation) and treatment with CSF1R inhibitors, and without both components, we cannot achieve sufficient CNS engraftment of donor populations. These experiments would not only be long term in nature, but also require significant optimization to complete, and we do not think that it would be effective. In our current study, we correlate motor rescue with influx of peripheral cells, suggesting that high numbers of donor cells are necessary to achieve rescue.
- 3) Another group has already identified the cell types that are capable of entering the brain following myeloablation, utilizing adoptive transfer of specific cell populations, supportive HSC administration, and CSF1R inhibitor treatment (Shibuya et al., 2022). These experiments demonstrate that only HSCs are capable of producing cells which engraft the brain under this paradigm. As such, we do not feel that repeating the experiment in our hands would justify animal loss or follow the core principles guiding responsible conduct in animal research (Replacement, Reduction, and Refinement). Instead, we have provided a complementary experiment in our prior resubmission.
- 4) We find it extremely unfair for the reviewer to hold another group's non-reproducible study against us. We do not understand how the reviewer can come to the conclusion that we now need adoptive transfer experiments here as a study utilizing bone marrow transplants from 15 years ago was not reproducible in a completely different disease etc. We are a different group studying a different disease and working with a different paradigm, with, I would argue, an excellent track record in reproducibility. Additionally, bone marrow transplants are used in

an incredibly broad spectrum of diseases in the clinic. We provide more information on this point below.

To address the reviewer's comment regarding clinical translation, we have added language on these points to the discussion in lines 872-881. We agree that there is a risk for overinterpretation of findings and have mentioned the need for further study more explicitly. The final paragraph of the discussion now reads as follows:

“Future advances in the safety and tolerability of BMT and optimization of microglial replacement will make this approach more widely applicable in patients. Our current approach involves the transplantation of recipient mice with a heterogeneous mixture of whole bone marrow donor cells and while we hypothesize based on prior research that the donor-derived cells engrafted in the CNS are derived from HSCs³³, further experimentation is required to identify the specific cell population which gives rise to the therapeutic CNS-engrafted cells. We also note that our endpoint of 16 weeks of age, while demonstrating robust correction of the SD CNS at a disease-relevant time point, is not sufficient to assess the duration of correction with microglial replacement. A long-term survival study would be required to determine if the CNS remains normalized across the lifespan. For these reasons, we caution against overinterpretation of the current results prior to further optimization. Overall, however, our approach harnesses a commonly utilized clinical practice in BMT and the innate properties of myeloid cells to deliver Hexβ to correct the SD CNS, with BMDMs replacing microglia as the putative cellular source of Hexβ. Further research and refinement of this approach to mitigate the present limitations will improve its viability and enhance a strategy that could be applied in other neurodegenerative LSDs and a litany of additional CNS conditions in the future.

RE: Non-reproducibility of bone marrow transplants in Rett syndrome: In addition to our comments above, we see no indication that the failure to reproduce the results of Derecki et al. can be attributed to the researchers' use of whole bone marrow. In fact, both the original manuscript and the follow up by Wang et al. utilized whole bone marrow for transplantation. There is no indication that identification of a specific cell type for transplant would have led to rescue in Rett syndrome. In fact, it is unsurprising that bone marrow transplant without microglial depletion was ineffective, as the limitations of this approach for CNS disorders have been well described: following traditional bone marrow transplant, there is a marked lack of bone marrow-derived cells in the brain. Existing bone marrow transplant paradigms have thus been insufficient in treating Sandhoff disease, Tay Sachs disease, and many other disorders with a neurological component regardless of the nature of the transferred cell. Additionally, other groups who have recently shown the effectiveness of microglial replacement with a myeloablative conditioning + CSF1R inhibitor paradigm in other disease contexts have used whole bone marrow for transplants¹⁻⁴.

We would also like to mention that while bone marrow transplant/HSC transplant was not effective in Rett syndrome in clinical trials, the disease pathogenesis of Rett syndrome bears very little similarity to Sandhoff disease, as it is a neurodevelopmental disorder rather than a neurodegenerative lysosomal storage disorder. By contrast, HSC transplant

is already the standard treatment for many lysosomal storage disorders with similar disease manifestations to Sandhoff disease, including Gaucher disease, Hurler syndrome, metachromatic leukodystrophy, and globoid cell leukodystrophy⁵.

2. Along this line, this study still lacks on a too early survival time point after bone marrow transfer (16 weeks of age). It is also essential to look for long-term survival including clinical tests, neuropathology, GM2 accumulations, engraftment of donor-derived cells in the CNS and rescue of neuronal numbers as a minimum. Here, I would ask for the 28 weeks' time point (7 months) for a robust experimental group of mice. Again, these later time points are key for future translation aspects in Sandhoff patients. I don't agree that this would induce "major ethical concerns" when control ko animals will not be treated.

We respectfully disagree with the reviewer regarding the importance of these data for the current proof of principle study and point out that they would take us well over a year to perform given the aforementioned breeding constraints and would not necessarily add anything to the conclusions we have already made.

Firstly, we chose an endpoint (4 months) at which mice begin to show mortality (on average, death occurs at 4.5 months) and have extensive pathology. We have demonstrated that we prevent or slow both. Why a 7-month timepoint specifically is now necessary is unclear. Our results are very clear, and even if our paradigm does not prevent lethality by 7 months, we are sure that parents of affected children would still be enthusiastic about a treatment that slows disease or prolongs life. We would like to reiterate to the reviewer that we are not setting out to cure SD in humans with a single study – this is a proof of principle study and provides a starting point for translation to humans. We do not claim to "cure" the disease but rather state what we observe within our paradigm, which is a lack of disease progression at a relevant age for this disease phenotype, when death begins to occur in untreated controls.

Secondly, we also disagree that these experiments would not present ethical issues. At 16 weeks, a widely used and disease-relevant time point in SD mice (that only live to 4.5 months at best), we already observed some lethality in all groups as stated in the original manuscript: "Additionally, four *Hexb*^{-/-} BMT mice died prematurely or required humane euthanasia at or before week 16, in comparison to three mice in the *Hexb*^{-/-} control group and only one mouse in the *Hexb*^{-/-} BMT + CSF1Ri group." Thus, any extensions beyond this point become a survival study by nature which have significant ethical concerns associated. For these reasons, we believe our IACUC would have an issue with a survival study as requested by the reviewer. Additionally, in a survival study, recovering the brains and bodies of mice becomes extremely challenging unless we are there the moment they die and would harshly limit our ability to assess the endpoints mentioned by the reviewer.

We do acknowledge that we are unable to assess the long-term implications of our treatment and whether the CNS remains normalized over time. Therefore, we have added this point to the discussion as mentioned above (Lines 876-878): “We also note that our endpoint of 16 weeks of age, while demonstrating robust correction of the SD CNS at a disease-relevant time point, is not sufficient to assess the duration of correction with microglial replacement. A long-term survival study would be required to determine if the CNS remains normalized across the lifespan.”

3. A clear mechanistic proof of microglia-derived Hexb delivery into the lysosomes of Hexb-deficient neurons in the putative rescued bone marrow chimeras is still lacking. This should be done *in situ*, namely in the brain sections of recipient mice.

We agree with this comment and have attempted to address it. However, as stated in our prior resubmission/response to reviewers, no antibodies against Hex β exist that allow for detection of Hex β *in situ*. In pursuit of this, we have screened several antibodies, searched the literature, and spoken with others in the field, but these attempts ultimately resulted in no success. To our knowledge, no other study has been able to identify an antibody against Hex β . The proposed experiment would therefore require a genetically modified *Hexb* mouse or virus to test this, which is beyond the timeframe that we can devote to this particular manuscript. We have added this caveat to the discussion. Please see lines 824-827: “These *in vitro* experiments provide insight into the potential mechanism(s) by which myeloid cell Hex β release plays a role in neuronal function, and, by extension, how neuronal lysosomal abnormalities may be corrected following BMT + CSF1Ri in *Hexb*^{-/-} mice. Future studies are necessary to confirm the mechanism of enzyme transfer *in vivo*.”

References:

1. Colella, P. *et al.* CNS-wide repopulation by hematopoietic-derived microglia-like cells corrects progranulin deficiency in mice. *Nat. Commun.* **15**, 5654 (2024).
2. Mader, M. M.-D. *et al.* Myeloid cell replacement is neuroprotective in chronic experimental autoimmune encephalomyelitis. *Nat. Neurosci.* **27**, 901–912 (2024).
3. Shibuya, Y. *et al.* Treatment of a genetic brain disease by CNS-wide microglia replacement. *Sci. Transl. Med.* **14**, eabl9945 (2022).
4. Yoo, Y., Neumayer, G., Shibuya, Y., Mader, M. M.-D. & Wernig, M. A cell therapy approach to restore microglial Trem2 function in a mouse model of Alzheimer’s disease. *Cell Stem Cell* **30**, 1043-1053.e6 (2023).
5. Tan, E. Y., Boelens, J. J., Jones, S. A. & Wynn, R. F. Hematopoietic Stem Cell Transplantation in Inborn Errors of Metabolism. *Front Pediatr* **7**, 433 (2019).